# Towards Understanding Hierarchical Learning: Benefits of Neural Representations

**Minshuo Chen**[*]     **Yu Bai**[†]     **Jason D. Lee**[◇]     **Tuo Zhao**[*]     **Huan Wang**[†]
**Caiming Xiong**[†]     **Richard Socher**[†]

[*]Georgia Tech     [†]Salesforce Research     [◇]Princeton University
{mchen393, tourzhao}@gatech.edu     jasonlee@princeton.edu
{yu.bai, huan.wang, cxiong, rsocher}@salesforce.com

## Abstract

Deep neural networks can empirically perform efficient hierarchical learning, in which the layers learn useful representations of the data. However, how they make use of the intermediate representations are not explained by recent theories that relate them to "shallow learners" such as kernels. In this work, we demonstrate that intermediate *neural representations* add more flexibility to neural networks and can be advantageous over raw inputs. We consider a fixed, randomly initialized neural network as a representation function fed into another trainable network. When the trainable network is the quadratic Taylor model of a wide two-layer network, we show that neural representation can achieve improved sample complexities compared with the raw input: For learning a low-rank degree-$p$ polynomial ($p \geq 4$) in $d$ dimension, neural representation requires only $\widetilde{O}(d^{\lceil p/2 \rceil})$ samples, while the best-known sample complexity upper bound for the raw input is $\widetilde{O}(d^{p-1})$. We contrast our result with a lower bound showing that neural representations do not improve over the raw input (in the infinite width limit), when the trainable network is instead a neural tangent kernel. Our results characterize when neural representations are beneficial, and may provide a new perspective on why depth is important in deep learning.

## 1 Introduction

Deep neural networks have been empirically observed to be more powerful than their shallow counterparts on a variety of machine learning tasks [38]. For example, on the ImageNet classification task, a 152-layer residual network can achieve 8%-10% better top-1 accuracy than a shallower 18-layer ResNet [30]. A widely held belief on why depth helps is that deep neural networks are able to perform efficient *hierarchical learning*, in which the layers learn representations that are increasingly useful for the present task. Such a hierarchical learning ability has been further leveraged in transfer learning. For example, [28] and [19] show that by combining with additional task-specific layers, the bottom layers of pre-trained neural networks for image classification and language modeling can be naturally transferred to other related tasks and achieve significantly improved performance.

Despite significant empirical evidence, we are in the lack of practical theory for understanding the hierarchical learning abilities of deep neural networks. Classical approximation theory has established a line of "depth separation" results which show that deep networks are able to approximate certain functions with much fewer parameters than shallow networks [18, 49, 24, 54, 12]. These work often manipulates the network parameters in potentially pathological ways, and it is unclear whether the resulting networks can be efficiently found through gradient-based optimization. A more recent line of work shows that overparametrized deep networks can be provably optimized and generalize as well as the so-called Neural Tangent Kernels (NTKs) [35, 21, 22, 3, 4, 7]. However, these results

do not take the hierarchical structure of the neural networks into account, and cannot justify any advantage of deep architectures. More recently, [33] show that some NTK models of deep networks are actually degenerate, and their generalization performance are no better than those associated with shallow networks.

In this paper, we provide a new persepctive for understanding hierarchical learning through studying intermediate *neural representations*—that is, feeding fixed, randomly initialized neural networks as a representation function (feature map) into another trainable model. The prototypical model we consider is a wide two-layer neural network taking a representation function $\mathbf{h}$ as the input, that is,

$$f_{\mathbf{W}}(\mathbf{x}) := \frac{1}{\sqrt{m}} \sum_{r=1}^{m} a_r \phi(\mathbf{w}_r^\top \mathbf{h}(\mathbf{x})), \tag{1}$$

where $\mathbf{x} \in \mathbb{R}^d$ is the feature, $\mathbf{h} : \mathbb{R}^d \to \mathbb{R}^D$ is a data-independent representation function that is held fixed during learning, and $\mathbf{W} = [\mathbf{w}_1, \dots, \mathbf{w}_m]^\top \in \mathbb{R}^{m \times D}$ is the weight matrix to be learned from the data. For example, when $\mathbf{h}(\mathbf{x}) = \sigma(\mathbf{V}\mathbf{x} + \mathbf{b})$ is another one-hidden-layer network (i.e. neural representations), the model $f$ is a three-layer network in which we only learn the weight matrix $\mathbf{W}$. Studying this model will reveal how the lower-level representation affects learning in a three-layer network, a previously missing yet important aspect of hierarchical learning.

To demonstrate the importance of the representation function $\mathbf{h}$, we investigate the *sample complexity* for learning certain target functions using model (1). This is a fine-grained measure of the power of $\mathbf{h}$ compared with other notions such as approximation ability. Indeed, we expect $f_{\mathbf{W}}$ to be able to approximate any "regular" (e.g. Lipschitz) function of $\mathbf{x}$, whenever we use a non-degenerate $\mathbf{h}$ and a sufficiently large width $m$. However, different choices of $\mathbf{h}$ can result in different ways (for the trainable two-layer network) to approximate the same target function, thereby leading to different sample complexity guarantees. We will specifically focus on understanding when learning with the neural representation $\mathbf{h}(\mathbf{x}) = \sigma(\mathbf{V}\mathbf{x} + \mathbf{b})$ is more sample efficient than learning with the raw input $\mathbf{h}(\mathbf{x}) = \mathbf{x}$, which is a sensible baseline for capturing the benefits of representations.

As the optimization and generalization properties of a general two-layer network can be rather elusive, we consider more *optimization aware* versions of the prototype (1)—we replace the trainable two-layer network in $f_{\mathbf{W}}$ by tractable alternatives such as its **linearized model** [21] (also known as "lazy training" in [15]) or **quadratic Taylor model** [8]:

(NTK-$\mathbf{h}$) $$f_{\mathbf{W}}^L(\mathbf{x}) = \frac{1}{\sqrt{m}} \sum_{r=1}^{m} a_r \phi'(\mathbf{w}_{0,r}^\top \mathbf{h}(\mathbf{x}))(\mathbf{w}_r^\top \mathbf{h}(\mathbf{x})),$$

(Quad-$\mathbf{h}$) $$f_{\mathbf{W}}^Q(\mathbf{x}) = \frac{1}{2\sqrt{m}} \sum_{r=1}^{m} a_r \phi''(\mathbf{w}_{0,r}^\top \mathbf{h}(\mathbf{x}))(\mathbf{w}_r^\top \mathbf{h}(\mathbf{x}))^2.$$

When $\mathbf{h}$ is the raw input (NTK-Raw, Quad-Raw), these are models with concrete convergence and generalization guarantees, and can approximate the training of the full two-layer network in appropriate infinite-width limits (e.g. [21, 7, 4, 39, 8]). However, for learning with other representation functions, these models are less understood. The goal of this paper is to provide a quantitative understanding of these models, in particular when $\mathbf{h}$ is a one-hidden-layer neural network (NTK-Neural, Quad-Neural), in terms of their convergence, generalization, and sample complexities of learning.

The contributions of this paper are summarized as follows:

• We show that the Quad-$\mathbf{h}$ model has a benign optimization landscape, and prove generalization error bounds with a precise dependence on the norm of the features and weight matrices, as well as the conditioning of the empirical covariance matrix of the features (Section 3).

• We study sample complexities of learning when the representation is chosen as a one-hidden-layer neural network (Quad-Neural model, Section 4). For achieving a small excess risk against a low-rank degree-$p$ polynomial, we show that the Quad-Neural model requires $\widetilde{O}(d^{\lceil p/2 \rceil})$ samples. When $p$ is large, this is significantly better than the best known $\widetilde{O}(d^{p-1})$ upper bound for the Quad-Raw model, demonstrating the benefits of neural representations.

• When the trainable network is instead a linearized model (or an NTK), we present a lower bound showing that neural representations are provably *not* beneficial: in a certain infinite-width limit, the NTK-Neural model requires at least $\Omega(d^p)$ samples for learning a degree-$p$ polynomial (Section 5).

Since $O(d^p)$ samples also suffice for learning with the NTK-Raw model, this shows that neural representations are not beneficial when fed into a linearized neural network.

**Additional paper organization**  We present the problem setup and algorithms in Section 2, review related work in Section 6, and provide conclusions as well as a broader impact statement in Section 7.

**Notations**  We use bold lower-case letters to denote vectors, e.g., $\mathbf{x} \in \mathbb{R}^d$, and bold upper-case letters to denote matrices, e.g., $\mathbf{W} \in \mathbb{R}^{d_1 \times d_2}$. Given a matrix $\mathbf{W} \in \mathbb{R}^{d_1 \times d_2}$, we let $\|\mathbf{W}\|_{\mathrm{op}}$ denote its operator norm, and $\|\mathbf{W}\|_{2,4}$ denote its $(2,4)$-norm defined as $\|\mathbf{W}\|_{2,4}^4 = \sum_{i=1}^{d_1} \|\mathbf{W}_{i,:}\|_2^4$, where $\mathbf{W}_{i,:} \in \mathbb{R}^{d_1}$ is the $i$-th row of $\mathbf{W}$. Given a function $f(\mathbf{x})$ defined on domain $\mathcal{X}$ with a probability measure $\mathcal{D}$, the $L_2$ norm is defined as $\|f\|_{L_2}^2 = \int_{\mathcal{X}} f^2(\mathbf{x}) \mathcal{D}(d\mathbf{x})$.

## 2 Preliminaries

**Problem setup**  We consider the standard supervised learning task, in which we receive $n$ i.i.d. training samples $S_n = \{(\mathbf{x}_i, y_i)\}_{i=1}^n$ from some data distribution $\mathcal{D}$, where $\mathbf{x} \in \mathcal{X}$ is the input and $y \in \mathcal{Y}$ is the label. In this paper, we assume that $\mathcal{X} = \mathbb{S}^{d-1} \subset \mathbb{R}^d$ (the unit sphere) so that inputs have unit norm $\|\mathbf{x}\|_2 = 1$. Our goal is to find a predictor $f : \mathcal{X} \mapsto \mathbb{R}$ such that the population risk

$$\mathcal{R}(f) := \mathbb{E}_{(\mathbf{x},y) \sim \mathcal{D}}[\ell(f(\mathbf{x}), y)]$$

is low, where $\ell : \mathbb{R} \times \mathcal{Y} \to \mathbb{R}$ is a loss function. We assume that $\ell(\cdot, y)$ is convex, twice differentiable with the first and second derivatives bounded by 1, and satisfies $|\ell(0, y)| \le 1$ for any $y \in \mathcal{Y}$. These assumptions are standard and are satisfied by commonly used loss functions such as the logistic loss and soft hinge loss.

Given dataset $S_n$, we define the empirical risk of a predictor $f$ as

$$\widehat{\mathcal{R}}(f) := \frac{1}{n} \sum_{i=1}^n \ell(f(\mathbf{x}), y).$$

**Model, regularization, and representation**  We consider the case where $f$ is either the linearized or the quadratic Taylor model of a wide two-layer network that takes a fixed representation function as the input:

(NTK-$\mathbf{h}$)  $$f_{\mathbf{W}}^L(\mathbf{x}) = \frac{1}{\sqrt{m}} \sum_{r=1}^m a_r \phi'(\mathbf{w}_{0,r}^\top \mathbf{h}(\mathbf{x}))(\mathbf{w}_r^\top \mathbf{h}(\mathbf{x})),$$

(Quad-$\mathbf{h}$)  $$f_{\mathbf{W}}^Q(\mathbf{x}) = \frac{1}{2\sqrt{m}} \sum_{r=1}^m a_r \phi''(\mathbf{w}_{0,r}^\top \mathbf{h}(\mathbf{x}))(\mathbf{w}_r^\top \mathbf{h}(\mathbf{x}))^2,$$

where $\mathbf{h} : \mathbb{R}^d \to \mathbb{R}^D$ is a fixed representation function, $\mathbf{w}_{0,r} \overset{\mathrm{iid}}{\sim} \mathsf{N}(\mathbf{0}, \mathbf{I}_D)$ and $a_r \overset{\mathrm{iid}}{\sim} \mathrm{Unif}(\{\pm 1\})$ are randomly initialized and held fixed during the training, $\mathbf{W} = [\mathbf{w}_1, \dots, \mathbf{w}_m]^\top \in \mathbb{R}^{m \times D}$ is the trainable weight matrix[1], and $\phi : \mathbb{R} \to \mathbb{R}$ is a nonlinear activation. These models are taken as proxies for a full two-layer network of the form $\frac{1}{\sqrt{m}} \mathbf{a}^\top \phi((\mathbf{W}_0 + \mathbf{W})\mathbf{h}(\mathbf{x}))$, so as to enable better understandings of their optimization.

For the Quad-$\mathbf{h}$ model, we add a regularizer to the risk so as to encourage $\mathbf{W}$ to have low norm. We use the regularizer $\|\mathbf{W}\|_{2,4}^4 = \sum_{r=1}^m \|\mathbf{w}_r\|_2^4$, and consider minimizing the regularized empirical risk

$$\widehat{\mathcal{R}}_\lambda(f_{\mathbf{W}}^Q) := \widehat{\mathcal{R}}(f_{\mathbf{W}}^Q) + \lambda \|\mathbf{W}\|_{2,4}^4 = \frac{1}{n} \sum_{i=1}^n \ell(f_{\mathbf{W}}^Q(\mathbf{x}_i), y_i) + \lambda \|\mathbf{W}\|_{2,4}^4. \qquad (2)$$

In the majority of this paper, we will focus on the case where $\mathbf{h}(\mathbf{x})$ is a fixed, randomly initialized neural network with one hidden layer of the form $\sigma(\mathbf{V}\mathbf{x} + \mathbf{b})$, with certain pre-processing steps when necessary. However, before we make the concrete choices, we think of $\mathbf{h}$ as a general function that maps the raw input space $\mathbb{R}^d$ into a feature space $\mathbb{R}^D$ without any additional assumptions.

**Connection to a three-layer model**   It is worth noticing that when $\mathbf{h}$ is indeed a neural network, say $\mathbf{h}(\mathbf{x}) = \sigma(\mathbf{V}\mathbf{x})$ (omitting bias for simplicity), our NTK-$\mathbf{h}$ and Quad-$\mathbf{h}$ models are closely related to the Taylor expansion of a *three-layer network*

$$\widetilde{f}_{\mathbf{W},\mathbf{V}}(\mathbf{x}) = \frac{1}{\sqrt{m}}\mathbf{a}^\top \phi((\mathbf{W}_0 + \mathbf{W})\sigma(\mathbf{V}\mathbf{x})).$$

Indeed, the {NTK-$\mathbf{h}$, Quad-$\mathbf{h}$} models correspond to the {linear, quadratic} Taylor expansion of the above network over $\mathbf{W}$, and is thus a part of the full Taylor expansion of the three-layer network. By studying these Taylor models, we gain understandings about how deep networks use its intermediate representation functions, which is lacking in existing work on Taylorized models.

## 3   Quadratic model with representations

We begin by studying the (non-convex) optimization landscape as well as the generalization properties of the model (Quad-$\mathbf{h}$), providing insights on what can be a good representation $\mathbf{h}$ for such a model.

**Base case of $\mathbf{h}(\mathbf{x}) = \mathbf{x}$: a brief review**   When $\mathbf{h}(\mathbf{x}) = \mathbf{x}$ is the raw input, model (Quad-$\mathbf{h}$) becomes

(Quad-Raw) $$f^Q_{\mathbf{W}}(\mathbf{x}) = \frac{1}{2\sqrt{m}}\sum_{r=1}^m a_r \phi''(\mathbf{w}_{0,r}^\top \mathbf{x})(\mathbf{w}_r^\top \mathbf{x})^2,$$

which is the quadratic Taylor model of a wide two-layer neural network. This model is analyzed by Bai and Lee [8] who show that (1) the (regularized) risk $\widehat{\mathcal{R}}_\lambda(f_{\mathbf{W}})$ enjoys a nice optimization landscape despite being non-convex, and (2) the generalization gap of the model $f^Q_{\mathbf{W}}$ is controlled by $\|\mathbf{W}\|_{2,4}$ as well as $\|\frac{1}{n}\sum_{i\in[n]}\mathbf{x}_i\mathbf{x}_i^\top\|_{\mathrm{op}}$. Building on these results, [8] show that learning low-rank polynomials with (Quad-Raw) achieves a better sample complexity than with the NTK. Besides the theoretical investigation, [9] empirically show that (Quad-Raw) model also approximates the training trajectories of standard neural networks better than the linearized model.

**General case**   We analyze optimization landscape and establish generalization guarantees when $\mathbf{h}$ is a general representation function, extending the results in [8]. We make the following assumption:

**Assumption 1** (Bounded representation and activation). There exists a constant $B_h$ such that $\|\mathbf{h}(\mathbf{x})\|_2 \leq B_h$ almost surely for $(\mathbf{x}, y) \sim \mathcal{D}$. The activation $\phi''$ is uniformly bounded: $\sup_{t\in\mathbb{R}} |\phi''(t)| \leq C$ for some absolute constant $C$.

**Theorem 1** (Optimization landscape and generalization of Quad-$\mathbf{h}$). Suppose Assumption 1 holds.

(1) (Optimization) Given any $\epsilon > 0$, $\tau = \Theta(1)$, and some radius $B_{w,\star} > 0$, suppose the width $m \geq \widetilde{O}(B_h^4 B_{w,\star}^4 \epsilon^{-1})$ and we choose a proper regularization coefficient $\lambda > 0$. Then any second-order stationary point [2] (SOSP) $\widehat{\mathbf{W}}$ of the regularized risk $\widehat{\mathcal{R}}_\lambda(f^Q_{\mathbf{W}})$ satisfies $\|\widehat{\mathbf{W}}\|_{2,4} \leq O(B_{w,\star})$, and achieves

$$\widehat{\mathcal{R}}_\lambda(f^Q_{\widehat{\mathbf{W}}}) \leq (1+\tau)\min_{\|\mathbf{W}\|_{2,4}\leq B_{w,\star}} \widehat{\mathcal{R}}(f^Q_{\mathbf{W}}) + \epsilon.$$

(2) (Generalization) For any radius $B_w > 0$, we have with high probability (over $(\mathbf{a}, \mathbf{W}_0)$) that

$$\mathbb{E}_{(\mathbf{x}_i, y_i)}\left[\sup_{\|\mathbf{W}\|_{2,4}\leq B_w} \left|\mathcal{R}(f^Q_{\mathbf{W}}) - \widehat{\mathcal{R}}(f^Q_{\mathbf{W}})\right|\right] \leq \widetilde{O}\left(\frac{B_h^2 B_w^2 M_{h,\mathrm{op}}}{\sqrt{n}} + \frac{1}{\sqrt{n}}\right),$$

where $M_{h,\mathrm{op}}^2 = B_h^{-2}\mathbb{E}_{\mathbf{x}}\left[\|\frac{1}{n}\sum_{i=1}^n \mathbf{h}(\mathbf{x}_i)\mathbf{h}(\mathbf{x}_i)^\top\|_{\mathrm{op}}\right]$.

**Efficient optimization; role of feature isotropicity** Theorem 1 has two main implications: (1) With a sufficiently large width, any SOSP of the regularized risk $\widehat{\mathcal{R}}(f_{\mathbf{W}}^Q)$ achieves risk close to the optimum in a certain norm ball, and has controlled norm itself. Therefore, escaping-saddle type algorithms such as noisy SGD [36, 40] that can efficiently find SOSPs can also efficiently find these near global minima. (2) The generalization gap is controlled by $M_{h,\mathrm{op}}$, which involves the operator norm of $\frac{1}{n}\sum_{i=1}^n \mathbf{h}(\mathbf{x}_i)\mathbf{h}(\mathbf{x}_i)^\top$. It is thus beneficial if our representation $\mathbf{h}(\mathbf{x})$ is (approximately) *isotropic*, so that $M_{h,\mathrm{op}} \asymp O(1/\sqrt{D})$, which is much lower than its naive upper bound 1. This will be a key insight for designing our neural representations in Section 4. The proof of Theorem 1 can be found in Appendix A.

## 4 Learning with neural representations

We now develop theories for learning with neural representations, where we choose $\mathbf{h}$ to be a wide one-hidden-layer neural network.

### 4.1 Neural representations

We consider a fixed, randomly initialized one-hidden-layer neural network:

$$\mathbf{g}(\mathbf{x}) = \sigma(\mathbf{V}\mathbf{x} + \mathbf{b}) = \left[\sigma(\mathbf{v}_1^\top \mathbf{x} + b_1), \dots, \sigma(\mathbf{v}_D^\top \mathbf{x} + b_D)\right]^\top \in \mathbb{R}^D, \tag{3}$$

where $\mathbf{v}_i \overset{\mathrm{iid}}{\sim} \mathsf{N}(0, \mathbf{I}_d)$ and $b_i \overset{\mathrm{iid}}{\sim} \mathsf{N}(0, 1)$ are the weights. Throughout this section we will use the indicator activation $\sigma(t) = \mathbb{1}\{t \geq 0\}$. We will also choose $\phi(t) = \mathrm{relu}(t)^2/2$ so that $\phi''(t) = \mathbb{1}\{t \geq 0\}$ as well.[3]

We define the representation function $\mathbf{h}(\mathbf{x})$ as the *whitened* version of $\mathbf{g}(\mathbf{x})$:

$$\mathbf{h}(\mathbf{x}) = \widehat{\mathbf{\Sigma}}^{-1/2}\mathbf{g}(\mathbf{x}), \quad \text{where} \quad \widehat{\mathbf{\Sigma}} = \frac{1}{n_0}\sum_{i=1}^{n_0} \mathbf{g}(\widetilde{\mathbf{x}}_i)\mathbf{g}(\widetilde{\mathbf{x}}_i)^\top. \tag{4}$$

Above, $\widehat{\mathbf{\Sigma}}$ is an estimator of the population covariance matrix[4] $\mathbf{\Sigma} = \mathbb{E}_{\mathbf{x}}[\mathbf{g}(\mathbf{x})\mathbf{g}(\mathbf{x})^\top] \in \mathbb{R}^{D\times D}$, and $\{\widetilde{\mathbf{x}}_i\}_{i\in[n_0]} =: \widetilde{S}_{n_0}$ is an additional set of unlabeled training examples of size $n_0$ (or a split from the existing training data). Such a whitening step makes $\mathbf{h}(\mathbf{x})$ more isotropic than the original features $\mathbf{g}(\mathbf{x})$, which according to Theorem 1 item (2) reduces the sample complexity for achieving low test error. We will discuss this more in Section 4.2.

We summarize our overall learning algorithm (with the neural representation) in Algorithm 1.

---

**Algorithm 1** Learning with Neural Representations (`Quad-Neural` method)

---

**Input**: Labeled data $S_n$, unlabeled data $\widetilde{S}_{n_0}$, initializations $\mathbf{V} \in \mathbb{R}^{D\times d}$, $\mathbf{b} \in \mathbb{R}^D$, $\mathbf{W}_0 \in \mathbb{R}^{m\times D}$, parameters $(\lambda, \epsilon)$.
**Step 1:** Construct model $f_{\mathbf{W}}^Q$ as

(Quad-Neural) $\qquad f_{\mathbf{W}}^Q(\mathbf{x}) = \frac{1}{2\sqrt{m}}\sum_{r=1}^m a_r \phi''(\mathbf{w}_{0,r}^\top \mathbf{h}(\mathbf{x}))(\mathbf{w}_r^\top \mathbf{h}(\mathbf{x}))^2,$

where $\mathbf{h}(\mathbf{x}) = \widehat{\mathbf{\Sigma}}^{-1/2}\mathbf{g}(\mathbf{x})$ is the neural representation (4) (using $\widetilde{S}_{n_0}$ to estimate the covariance).
**Step 2:** Find a second-order stationary point $\widehat{\mathbf{W}}$ of the regularized empirical risk (on the data $S_n$):

$$\widehat{\mathcal{R}}_\lambda(f_{\mathbf{W}}^Q) = \frac{1}{n}\sum_{i=1}^n \ell(f_{\mathbf{W}}^Q(\mathbf{x}_i), y_i) + \lambda\|\mathbf{W}\|_{2,4}^4.$$

## 4.2 Learning low-rank polynomials with neural representations

We now study the sample complexity of Algorithm 1 to achieve low excess test risk compared with the best *low-rank degree-p polynomial*, that is, sum of polynomials of the form $(\boldsymbol{\beta}^\top \mathbf{x})^p$. This setting has been considered in a variety of prior work on learning polynomials [47, 13] as well as analyses of wide neural networks [7, 8].

We need the following additional assumption on the random features.

**Assumption 2** (Lower Bounded Covariance). For any $k$ and $D \leq O(d^k)$, with high probability over $\mathbf{V}, \mathbf{b}$ (as $d \to \infty$), we have the minimum eigenvalue $\lambda_{\min}(\boldsymbol{\Sigma}) \geq \lambda_k$ for some constant $\lambda_k > 0$ that only depends on $k$ but not $d$, where $\boldsymbol{\Sigma} = \mathbb{E}_{\mathbf{x}}[\sigma(\mathbf{V}\mathbf{x} + \mathbf{b})\,\sigma(\mathbf{V}\mathbf{x} + \mathbf{b})^\top]$.

Assumption 2 states the features $\{\sigma(\mathbf{v}_i^\top \mathbf{x} + b_i)\}$ to be not too correlated, which roughly requires the distribution of $\mathbf{x}$ to span all directions in $\mathbb{R}^d$. For example, when $\mathbf{x} \sim \mathrm{Unif}(\mathbb{S}^{d-1})$ (and with our choice of $\sigma(t) = \mathbb{1}\{t \geq 0\}$), we show that this assumption is satisfied with

$$\lambda_k = \Theta\left(\min_{\deg(q)\leq k} \mathbb{E}_{z\sim\mathsf{N}(0,1)}[(\sigma(z) - q(z))^2]\right) \asymp k^{-1/2},$$

where $q(z)$ denotes a polynomial in $z$ and its degree is denoted as $\deg(q)$. For general distributions of $\mathbf{x}$, we show Assumption 2 still holds under certain moment conditions on the distribution of $\mathbf{x}$ (see the formal statement and proof of both results in Appendix B).

**Sample complexity for learning polynomials**  We focus on low-rank polynomials of the form

$$f_\star(\mathbf{x}) = \sum_{s=1}^{r_\star} \alpha_s(\boldsymbol{\beta}_s^\top \mathbf{x})^{p_s}, \quad \text{where} \quad |\alpha_s| \leq 1, \;\; \left\|(\boldsymbol{\beta}_s^\top \mathbf{x})^{p_s}\right\|_{L_2} \leq 1, \; p_s \leq p \;\text{ for all } s. \tag{5}$$

We state our main result for the Quad-Neural model to achieve low excess risk over such functions.

**Theorem 2** (Sample complexity of learning with Quad-Neural). Suppose Assumption 2 holds, and there exists some $f_\star$ of the form (5) that achieves low risk: $\mathcal{R}(f_\star) \leq \mathsf{OPT}$. Then for any $\epsilon, \delta \in (0,1)$ and $\tau = \Theta(1)$, choosing

$$D = \Theta\left(\mathrm{poly}(r_\star, p) \sum_s \|\boldsymbol{\beta}_s\|_2^{2\lceil p_s/2\rceil} \epsilon^{-2}\delta^{-1}\right), \quad m \geq \widetilde{O}\big(\mathrm{poly}(r_\star, D)\epsilon^{-2}\delta^{-1}\big), \tag{6}$$

$n_0 = \widetilde{O}(D\delta^{-2})$, and a proper $\lambda > 0$, Algorithm 1 achieves the following guarantee: with probability at least $1 - \delta$ over the randomness of data and initialization, any second-order stationary point $\widehat{\mathbf{W}}$ of $\widehat{\mathcal{R}}_\lambda(f_{\mathbf{W}}^Q)$ satisfies

$$\mathcal{R}(f_{\widehat{\mathbf{W}}}^Q) \leq (1+\tau)\mathsf{OPT} + \underbrace{\epsilon}_{\substack{\text{approx.,}\\ \text{requires large } D}} + \underbrace{\widetilde{O}\left(\sqrt{\frac{\mathrm{poly}(r_\star, p, \delta^{-1})\lambda_{\lceil p/2\rceil}^{-1}\epsilon^{-2}\sum_{s=1}^{r_\star}\|\boldsymbol{\beta}_s\|_2^{2\lceil p_s/2\rceil}}{n}}\right)}_{\text{generalization, requires large } n \text{ (given } \epsilon)}.$$

In particular, for any $\epsilon > 0$, we can achieve $\mathcal{R}(f_{\widehat{\mathbf{W}}}^Q) \leq (1+\tau)\mathsf{OPT} + 2\epsilon$ with sample complexity

$$n_0 + n \leq \widetilde{O}\left(\mathrm{poly}(r_\star, p, \lambda_{\lceil p/2\rceil}^{-1}, \epsilon^{-1}, \delta^{-1}) \sum_{s=1}^{r_\star} \|\boldsymbol{\beta}_s\|_2^{2\lceil p_s/2\rceil}\right). \tag{7}$$

According to Theorem 2, Quad-Neural can learn polynomials of any degree by doing the following: (1) Choose a sufficiently large $D$, so that the neural representations are expressive enough; (2) Choose a large width $m$ in the quadratic model so as to enable a nice optimization landscape, where such $m$ only appears logarithmically in generalization error (Theorem 1).

**Improved dimension dependence over `Quad-Raw` and `NTK-Raw`** We parse the sample complexity bound in Theorem 2 in the following important case: $\mathbf{x}$ is relatively uniform (e.g. $\mathrm{Unif}(\mathbb{S}^{d-1})$), $\|\boldsymbol{\beta}_s\|_2 = O(\sqrt{d})$, and the data is noiseless and realized by $f_\star$ (so that $\mathsf{OPT} = 0$). In this case we have $\left\|(\boldsymbol{\beta}_s^\top \mathbf{x})^{p_s}\right\|_{L_2} = O(1)$, and Assumption 2 holds with $\lambda_{\lceil p/2 \rceil} \geq p^{-O(1)}$. Thus, when only highlighting the $d$ dependence[5], the sample complexity required to achieve $\epsilon$ test risk with the `Quad-Neural` is (reading from (18))

$$N_{\texttt{quad-neural}} = \widetilde{O}\Big(d^{\lceil p/2 \rceil}\Big).$$

In comparison, the sample complexity for learning with the `Quad-Raw` (quadratic neural network with the raw input) is

$$N_{\texttt{quad-raw}} = \widetilde{O}\big(d^{p-1}\big)$$

(see, e.g. [8, Thm 7]). Therefore, Theorem 2 shows that neural representations can significantly improve the sample complexity over the raw input, when fed into a quadratic Taylor model.

**Overview of techniques** At a high level, the improved sample complexity achieved in Theorem 2 is due to the flexibility of the neural representation: the `Quad-h` model can express polynomials *hierarchically*, using weight matrices with much smaller norms than that of a shallow learner such as the `Quad-Raw` model. This lower norm in turn translates to a better generalization bound (according to Theorem 1) and an improved sample complexity. We sketch the main arguments here, and leave the complete proof to Appendix D.

(1) **Expressing functions using hierarchical structure**: We prove the existence of some $\mathbf{W}^* \in \mathbb{R}^{m \times D}$ such that $f_{\mathbf{W}^*}^Q \approx f_\star$ by showing the following: (1) As soon as $D \geq \widetilde{O}(d^k)$, $\mathbf{h}(\mathbf{x})$ can linearly express certain degree-$k$ polynomials as "bases"; (2) For large $m$, the top quadratic taylor model can further express degree $\lceil p_s/k \rceil$ polynomials of the bases, thereby expressing $f_\star$. This is an explicit way of utilizing the hierarchical structure of the model. We note that our proof used $k = \lceil p/2 \rceil$, but the argument can be generalized to other $k$ as well.

(2) **Making representations isotropic**: We used a whitened version of a one-hidden-layer network as our representation function $\mathbf{h}$ (cf. (4)). The whitening operation does not affect the expressivity argument in part (1) above, but helps improve the conditioning of the feature covariance matrix (cf. the quantity $M_{h,\mathrm{op}}$ in Theorem 1). Applying whitening, we obtain nearly isotropic features: $\mathbb{E}_{\mathbf{x}}[\mathbf{h}(\mathbf{x})\mathbf{h}(\mathbf{x})^\top] \approx \mathbf{I}_D$, which is key to the sample complexity gain over the `Quad-Raw` model as discussed above. We note that well-trained deep networks with BatchNorm may have been implicitly performing such whitening operations in practice [46]. We also remark that the whitening step in Algorithm 1 may be replaced with using unwhitened representations with a *data-dependent* regularizer, e.g., $\sum_{r=1}^m \|\widehat{\boldsymbol{\Sigma}}^{1/2}\mathbf{w}_r\|_2^4$, which achieves similar sample complexity guarantees (see Appendix D).

## 5 NTK with neural representations: a lower bound

In this section, we show that neural representations may not be beneficial over raw inputs when the trainable network is a *linearized* neural network through presenting a sample complexity lower bound for this method in the infinite width limit.

More concretely, we consider `NTK-Neural`, which learns a model $f_{\mathbf{W}}^L$ of the form

$$(\texttt{NTK-Neural}) \quad f_{\mathbf{W}}^L(\mathbf{x}) := \frac{1}{\sqrt{m}} \sum_{r=1}^m a_r \phi'\Big(\mathbf{w}_{0,r}^\top \mathbf{g}(\mathbf{x})/\sqrt{D}\Big)\Big(\mathbf{w}_r^\top \mathbf{g}(\mathbf{x})/\sqrt{D}\Big),$$

where $\mathbf{g}(\mathbf{x}) := [\sigma(\mathbf{v}_1^\top \mathbf{x} + b_1), \dots, \sigma(\mathbf{v}_D^\top \mathbf{x} + b_D)]^\top \in \mathbb{R}^D$ are the neural random features (same as in (3)), and the $1/\sqrt{D}$ factor rescales $\mathbf{g}(\mathbf{x})$ to $O(1)$ norm on average.

**Infinite-width limit: a kernel predictor**    Model (`NTK-Neural`) is linear model with parameter $\mathbf{W}$, and can be viewed as a kernel predictor with a (finite-dimensional kernel) $H_{m,D} : \mathbb{S}^{d-1} \times \mathbb{S}^{d-1} \to \mathbb{R}$. In the infinite-width limit of $D, m \to \infty$, we have $H_{m,D} \to H_\infty$, where

$$H_\infty(\mathbf{x}, \mathbf{x}') := \mathbb{E}_{(u,v) \sim \mathsf{N}(\mathbf{0}, \mathbf{\Sigma}(\mathbf{x}, \mathbf{x}'))}[\phi'(u)\phi'(v)] \cdot \Sigma_{12}(\mathbf{x}, \mathbf{x}'), \text{ and}$$

$$\mathbf{\Sigma}(\mathbf{x}, \mathbf{x}') = \begin{pmatrix} \mathbb{E}_{\mathbf{v},b}[\sigma(\mathbf{v}^\top \mathbf{x} + b)^2] & \mathbb{E}_{\mathbf{v},b}[\sigma(\mathbf{v}^\top \mathbf{x} + b)\sigma(\mathbf{v}^\top \mathbf{x}' + b)] \\ \mathbb{E}_{\mathbf{v},b}[\sigma(\mathbf{v}^\top \mathbf{x} + b)\sigma(\mathbf{v}^\top \mathbf{x}' + b)] & \mathbb{E}_{\mathbf{v},b}[\sigma(\mathbf{v}^\top \mathbf{x}' + b)^2] \end{pmatrix},$$

(see e.g. [35, 20] for the derivation). Motivated by this, we consider kernel predictors of the form

$$\widehat{f}_\lambda = \underset{f}{\operatorname{argmin}} \sum_{i=1}^n \ell(f(\mathbf{x}_i), y_i) + \lambda \|f\|_{H_\infty}^2 \tag{8}$$

as a proxy for (`NTK-Neural`), where $\|\cdot\|_{H_\infty}^2$ denotes the RKHS (Reproducing Kernel Hilbert Space) norm associated with kernel $H_\infty$. This set of predictors is a reliable proxy for the (`NTK-Neural`) method: for example, taking $\lambda \to 0_+$, it recovers the solution found by gradient descent (with a small stepsize) on the top layer of a wide three-layer network [20].

We now present a lower bound for the predictor $\widehat{f}_\lambda$, adapted from [27, Theorem 3].

**Theorem 3** (Lower bound for `NTK-Neural`). Suppose the input distribution is $\mathbf{x} \sim \operatorname{Unif}(\mathbb{S}^{d-1})$, and $y_\star = f_\star(\mathbf{x})$ where $f_\star \in L_2(\operatorname{Unif}(\mathbb{S}^{d-1}))$ consists of polynomials of degree at least $p$[6]. Assume the sample size $n \le O(d^{p-\delta})$ for some $\delta > 0$. Then for any fixed $\epsilon \in (0,1)$, as $d \to \infty$, the predictor $\widehat{f}_\lambda$ defined in (8) suffers from the following lower bound with high probability (over $\{(\mathbf{x}_i, y_i)\}$):

$$\mathbb{E}_{\mathbf{x}}\left[\inf_{\lambda > 0}(\widehat{f}_\lambda(\mathbf{x}) - f_\star(\mathbf{x}))^2\right] \ge (1-\epsilon)\mathbb{E}_{\mathbf{x}}[f_\star(\mathbf{x})]^2,$$

that is, any predictor of the form (8) will not perform much better than the trivial zero predictor.

**No improvement over `NTK-Raw`; benefits of neural representations**    Theorem 3 shows that the infinite width version (8) of the `NTK-Neural` method requires roughly at least $\Omega(d^p)$ samples in order to learn any degree-$p$ polynomial up to a non-trivial accuracy (in squared error). Crucially, this lower bound implies that `NTK-Neural` *does not* improve over `NTK-Raw` (i.e. NTK with the raw input) in the infinite width limit—the infinite width `NTK-Raw` already achieves sample complexity *upper bound* of $O(d^p)$ for learning a degree-$p$ polynomial $y = f_\star(\mathbf{x})$ when $\mathbf{x} \sim \operatorname{Unif}(\mathbb{S}^{d-1})$ [27]. This is in stark contrast with our Theorem 2 which shows that `Quad-Neural` improves over `Quad-Raw`, suggesting that neural representations are perhaps only beneficial when fed into a sufficiently complex model.

## 6    Related work

**Approximation theory and depth separation.** Extensive efforts have been made on the expressivity of neural networks and the benefits of increased depth. Two separate focuses were pursued: 1) Universal approximation theory for approximating dense function classes, e.g., Sobolev and squared integrable functions [17, 31, 10, 34, 25, 16, 41, 44, 43]; 2) depth separation theory demonstrating the benefits of increased depth on expressing certain structured functions, e.g., saw-tooth functions [29, 18, 48, 49, 24]. More recently, the recent work [54] merged the two focuses by studying unbounded-depth ReLU networks for approximating Sobolev functions. In all these work, the network parameters are constructed in potentially weird ways, and it is unclear whether such networks can be efficiently found using gradient-based optimization.

**Neural tangent kernels and beyond**    A growing body of recent work show the connection between gradient descent on the full network and the Neural Tangent Kernel (NTK) [35], from which one can prove concrete results about neural network training [42, 21, 20, 3, 56] and generalization [7, 4, 11]. Despite such connections, these results only show that neural networks are as powerful as shallow learners such as kernels. The gap between such shallow learners and the full neural network has been established in theory by [52, 1, 55, 26, 53, 23] and observed in practice [6, 39, 14]. Higher-order expansions of the {network, training dynamics} such as Taylorized Training [8, 9] and the Neural

Tangent Hierarchy [32] have been recently proposed towards closing this gap. Finally, recent work by Allen-Zhu and Li [2] shows that there exists a class of polynomials that can be efficiently learned by a deep network but not any "non-hiearchical" learners such as kernel methods or neural tangent kernels, thereby sheding light on how representations are learned hierarchically.

**Learning low-rank polynomials in high dimension** In [47] and [45], the authors propose a tensor unfolding algorithm to estimate a rank $k$ order $p$ tensor with $(d)^{p/2}k$ samples. Under Gaussian input data, [13] propose a Grassmanian manifold optimization algorithm with spectral initialization to estimate a polynomial over $k$-dimensional subspace of variables of degree $p$ with $O_{k,p}(d\log^d p)$ samples, where $O_{k,p}$ suppresses unknown (super)-exponential dependence on $k$ and $p$. However, these methods explicitly use knowledge about the data distribution. Neural networks can often learn polynomials in distribution-free ways. [5, 7] show that wide two-layer networks that simulate an NTK require $\widetilde{O}(d^p)$ samples to learn a degree-$p$ polynomial. [27] show that $\Omega(d^p)$ samples is also asymptotically necessary for any rotationally invariant kernel. [8] show that a randomized wide two-layer network requires $\widetilde{O}(d^{p-1})$ samples instead by coupling it with the quadratic Taylor model. Our algorithm belongs to this class of distribution-free methods, but achieve an improved sample complexity when the distribution satisfies a mild condition.

## 7 Conclusion

This paper provides theoretical results on the benefits of neural representations in deep learning. We show that using a neural network as a representation function can achieve improved sample complexity over the raw input in a neural quadratic model, and also show such a gain is not present if the model is instead linearized. We believe these results provide new understandings to hiearchical learning in deep neural networks. For future work, it would be of interest to study whether deeper representation functions are even more beneficial than shallower ones, or what happens when the representation is fine-tuned together with the trainable network.

## Acknowledgment

We thank the anonymous reviewers for the suggestions. We thank Song Mei for the discussions about the concentration of long-tailed covariance matrices. JDL acknowledges support of the ARO under MURI Award W911NF-11-1-0303, the Sloan Research Fellowship, and NSF CCF 2002272.

## Broader impact

This paper extensively contributes to the theoretical frontier of deep learning. We do not foresee direct ethical or societal consequences. Instead, our theoretical finding reduces the gap between the theory and practice, and is in sharp contrast to existing theories, which cannot show any advantage of deep networks over the shallow ones. In viewing of a notably increasing trend towards establishing a quantitative framework using deep neural networks in diverse areas, e.g., computational social science, this paper will provide an important theoretical guideline for practitioners.

## Footnotes

[1] Our parameterization decouples the weight matrix in a standard two-layer network into two parts: the initialization $\mathbf{W}_0 \in \mathbb{R}^{m \times D}$ that is held fixed during training, and the "weight movement matrix" $\mathbf{W} \in \mathbb{R}^{m \times D}$ that can be thought of as initialized at $\mathbf{0}$.

[2]$\mathbf{W}$ is a second-order stationary point (SOSP) of a twice-differentiable loss $L(\mathbf{W})$ if $\nabla L(\mathbf{W}) = \mathbf{0}$ and $\nabla^2 L(\mathbf{W}) \succeq \mathbf{0}$.

[3]We can use a non-smooth $\sigma$ since $(\mathbf{V}, \mathbf{b})$ are not trained. Our results can be extended to the situation where $\sigma$ or $\phi''$ is the relu activation as well.

[4]Strictly speaking, $\mathbf{\Sigma}$ is the second moment matrix of $\mathbf{g}(\mathbf{x})$.

[5]For example, in the high-dimensional setting when $\epsilon = \Theta(1)$ and $d$ is large [27].

[6]That is, $\|\mathsf{P}_{<p}f_\star\|_{L_2} = 0$, where $\mathsf{P}_{<p}$ denotes the $L_2$ projection onto the space of degree $< p$ polynomials.

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
