[Supplementary Material]

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

# Supplementary Materials for "Towards Understanding Hierarchical Learning: Benefits of Neural Representations"

## A   Proofs for Section 3

### A.1   Proof of Optimization in Theorem 1

We first derive the gradient and Hessian of empirical risk $\widehat{\mathcal{R}}(f_{\mathbf{W}}^Q)$, which will be used throughout the rest of the proof. For a better presentation, we denote $\langle \cdot, \cdot \rangle$ as inner product and

$$f_{\mathbf{W}}^Q(\mathbf{x}) = \frac{1}{2\sqrt{m}} \left\langle \mathbf{x}\mathbf{x}^\top, \mathbf{W}\mathbf{D}(\mathbf{x})\mathbf{W}^\top \right\rangle \text{ for } \mathbf{D}_{rr}(\mathbf{x}) = a_r \phi''(\mathbf{w}_{0,r}^\top \mathbf{h}(\mathbf{x})).$$

We compute the gradient and Hessian of $\widehat{\mathcal{R}}(f_{\mathbf{W}}^Q)$ along a given direction $\mathbf{W}_\star$.

$$\nabla_{\mathbf{W}} \widehat{\mathcal{R}}(f_{\mathbf{W}}^Q) = \frac{2}{n} \sum_{i=1}^n \ell'(f_{\mathbf{W}}^Q(\mathbf{x}_i), y_i) \frac{1}{2\sqrt{m}} \mathbf{x}_i \mathbf{x}_i^\top \mathbf{W}\mathbf{D}(\mathbf{x}_i) \quad \text{and}$$

$$\nabla_{\mathbf{W}}^2 \widehat{\mathcal{R}}(f_{\mathbf{W}}^Q)[\mathbf{W}_\star, \mathbf{W}_\star] = \frac{2}{n} \sum_{i=1}^n \ell'(f_{\mathbf{W}}^Q(\mathbf{x}_i), y_i) \cdot \underbrace{\frac{1}{2\sqrt{m}} \left\langle \mathbf{x}_i \mathbf{x}_i^\top, \mathbf{W}_\star \mathbf{D}(\mathbf{x}_i) \mathbf{W}_\star^\top \right\rangle}_{f_{\mathbf{W}_\star}^Q(\mathbf{x}_i)}$$

$$+ \frac{4}{n} \sum_{i=1}^n \ell''(f_{\mathbf{W}}^Q(\mathbf{x}_i), y_i) \cdot \left( \underbrace{\frac{1}{2\sqrt{m}} \left\langle \mathbf{x}_i \mathbf{x}_i^\top, \mathbf{W}\mathbf{D}(\mathbf{x}_i) \mathbf{W}_\star^\top \right\rangle}_{\widetilde{y}_i} \right)^2$$

$$= \underbrace{\frac{2}{n} \sum_{i=1}^n \ell'(f_{\mathbf{W}}^Q(\mathbf{x}_i), y_i) f_{\mathbf{W}_\star}^Q(\mathbf{x}_i)}_{\text{I}} + \underbrace{\frac{4}{n} \sum_{i=1}^n \ell''(f_{\mathbf{W}}^Q(\mathbf{x}_i), y_i) \widetilde{y}_i^2}_{\text{II}}.$$

We denote $\widehat{\mathcal{D}}$ as the empirical data distribution, and bound I and II separately.

$$\text{I} = 2\mathbb{E}_{\widehat{\mathcal{D}}}[\ell'(f_{\mathbf{W}}^Q(\mathbf{x}), y) f_{\mathbf{W}_\star}^Q(\mathbf{x})]$$

$$= 2\mathbb{E}_{\widehat{\mathcal{D}}}[\ell'(f_{\mathbf{W}}^Q(\mathbf{x}), y) f_{\mathbf{W}}^Q(\mathbf{x})] + 2\mathbb{E}_{\widehat{\mathcal{D}}}[\ell'(f_{\mathbf{W}}^Q(\mathbf{x}), y)(f_{\mathbf{W}_\star}^Q(\mathbf{x}) - f_{\mathbf{W}}^Q(\mathbf{x}))]$$

$$\overset{(i)}{\leq} \left\langle \nabla \widehat{\mathcal{R}}(f_{\mathbf{W}}^Q), \mathbf{W} \right\rangle + 2\mathbb{E}_{\widehat{\mathcal{D}}}[\ell(f_{\mathbf{W}_\star}^Q(\mathbf{x}), y) - \ell(f_{\mathbf{W}}^Q(\mathbf{x}), y)]$$

$$= \left\langle \nabla \widehat{\mathcal{R}}(f_{\mathbf{W}}^Q), \mathbf{W} \right\rangle - 2(\widehat{\mathcal{R}}(f_{\mathbf{W}}^Q) - \widehat{\mathcal{R}}(f_{\mathbf{W}^*}^Q)),$$

where (i) follows directly by computing $\left\langle \nabla \widehat{\mathcal{R}}(f_{\mathbf{W}}^Q), \mathbf{W} \right\rangle$ and the convexity of $\ell$. For II, with $\ell'' \leq 1$, we have

$$\text{II} \leq \frac{4}{n} \sum_{i=1}^n \sum_{r \leq m} \widetilde{y}_i^2 = \mathbb{E}_{\widehat{\mathcal{D}}} \left[ \frac{2}{m} \sum_{r \leq m} \phi''(\mathbf{w}_{0,r}^\top \mathbf{h}(\mathbf{x}))^2 (\mathbf{w}_r^\top \mathbf{h}(\mathbf{x}))^2 (\mathbf{w}_{\star,r}^\top \mathbf{h}(\mathbf{x}))^2 \right]$$

$$\leq C^2 \mathbb{E}_{\widehat{\mathcal{D}}} \left[ \frac{1}{m} \sum_{r \leq m} (\mathbf{w}_r^\top \mathbf{h}(\mathbf{x}))^2 (\mathbf{w}_{\star,r}^\top \mathbf{h}(\mathbf{x}))^2 \right]$$

$$\leq \frac{1}{m} C^2 B_h^4 \sum_{r \leq m} \|\mathbf{w}_r\|_2^2 \|\mathbf{w}_{\star,r}\|_2^2$$

$$\leq m^{-1} C^2 B_h^4 \|\mathbf{W}\|_{2,4}^2 \|\mathbf{W}_\star\|_{2,4}^2,$$

where the last step used Cauchy-Schwarz on $\{\|\mathbf{w}_r\|_2\}$ and $\{\|\mathbf{w}_{\star,r}\|_2\}$, and the constant $C$ is the uniform upper bound on $\phi''$. Putting terms I and II together, we have

$$\nabla^2_{\mathbf{W}}\widehat{\mathcal{R}}(f^Q_{\mathbf{W}})[\mathbf{W}_\star, \mathbf{W}_\star] \leq \left\langle \nabla\widehat{\mathcal{R}}(f^Q_{\mathbf{W}}), \mathbf{W} \right\rangle - 2(\widehat{\mathcal{R}}(f^Q_{\mathbf{W}}) - \widehat{\mathcal{R}}(f^Q_{\mathbf{W}^*})) + m^{-1}C^2 B^4_h \|\mathbf{W}\|^2_{2,4} \|\mathbf{W}_\star\|^2_{2,4}. \tag{9}$$

*Proof of Theorem 1, Optimization Part.* We denote $\mathbf{W}^* = \mathrm{argmin}_{\|\mathbf{W}\|_{2,4} \leq B_{w,\star}} \widehat{\mathcal{R}}(f^Q_{\mathbf{W}})$ and let its risk $\widehat{\mathcal{R}}(f^Q_{\mathbf{W}^*}) = M$. We begin by choosing the regularization strength as

$$\lambda = \lambda_0 B^{-4}_{w,\star},$$

where $\lambda_0$ is a constant to be determined.

We argue that any second order stationary point $\widehat{\mathbf{W}}$ has to satisfy $\|\widehat{\mathbf{W}}\|_{2,4} = O(B_{w,\star})$. We have for any $\mathbf{W}$ that

$$\begin{aligned}
\left\langle \nabla\widehat{\mathcal{R}}(f^Q_{\mathbf{W}}), \mathbf{W} \right\rangle &= \mathbb{E}_{\widehat{\mathcal{D}}}\left[ \ell'(f^Q_{\mathbf{W}}(\mathbf{x}), y) \cdot 2f^Q_{\mathbf{W}}(\mathbf{x}) \right] \\
&= 2\mathbb{E}_{\widehat{\mathcal{D}}}\left[ \ell'(f^Q_{\mathbf{W}}(\mathbf{x}), y) \cdot (f^Q_{\mathbf{W}}(\mathbf{x}) - f^Q_0(\mathbf{x})) \right] \\
&\overset{(i)}{\geq} 2(\widehat{\mathcal{R}}(f^Q_{\mathbf{W}}) - \widehat{\mathcal{R}}(f^Q_0)) \\
&\overset{(ii)}{\geq} -2,
\end{aligned}$$

where (i) uses convexity of $\ell$ and (ii) uses the assumption that $\ell(0, y) \leq 1$ for all $y \in \mathcal{Y}$.

Combining with the fact that $\left\langle \nabla_{\mathbf{W}}(\|\mathbf{W}\|^4_{2,4}), \mathbf{W} \right\rangle = 4\|\mathbf{W}\|^4_{2,4}$, we have simultaneously for all $\mathbf{W}$ that

$$\begin{aligned}
\left\langle \nabla\widehat{\mathcal{R}}_\lambda(f^Q_{\mathbf{W}}), \mathbf{W} \right\rangle &\geq \left\langle \nabla_{\mathbf{W}}(\lambda\|\mathbf{W}\|^4_{2,4}), \mathbf{W} \right\rangle + \left\langle \nabla_{\mathbf{W}}\widehat{\mathcal{R}}(f^Q_{\mathbf{W}}), \mathbf{W} \right\rangle \\
&\geq 4\lambda\|\mathbf{W}\|^4_{2,4} - 2.
\end{aligned}$$

Therefore we see that any stationary point $\mathbf{W}$ has to satisfy

$$\|\mathbf{W}\|_{2,4} \leq (2\lambda)^{-1/4}.$$

Choosing

$$\lambda_0 = \frac{1}{36}(2\tau M + \epsilon),$$

we get $36\lambda B^4_{w,\star} = 2\tau M + \epsilon$. The Hessian of $\widehat{\mathcal{R}}_\lambda(f^Q_{\mathbf{W}})$ along direction $\mathbf{W}_\star$ is

$$\begin{aligned}
\nabla^2_{\mathbf{W}}\widehat{\mathcal{R}}_\lambda(f^Q_{\mathbf{W}})[\mathbf{W}_\star, \mathbf{W}_\star] &= \nabla^2_{\mathbf{W}}\widehat{\mathcal{R}}(f^Q_{\mathbf{W}})[\mathbf{W}_\star, \mathbf{W}_\star] + \lambda\nabla^2_{\mathbf{W}}\|\mathbf{W}\|^4_{2,4}[\mathbf{W}_\star, \mathbf{W}_\star] \\
&= \nabla^2_{\mathbf{W}}\widehat{\mathcal{R}}(f^Q_{\mathbf{W}})[\mathbf{W}_\star, \mathbf{W}_\star] + 4\lambda\sum_{r \leq m}\|\mathbf{w}_r\|^2_2\|\mathbf{w}_{\star,r}\|^2_2 + 2\langle\mathbf{w}_r, \mathbf{w}_{\star,r}\rangle^2 \\
&\leq \left\langle \nabla\widehat{\mathcal{R}}(f^Q_{\mathbf{W}}), \mathbf{W} \right\rangle - 2(\widehat{\mathcal{R}}(f^Q_{\mathbf{W}}) - M) + m^{-1}C^2 B^4_h\|\mathbf{W}\|^2_{2,4}\|\mathbf{W}_\star\|^2_{2,4} \\
&\quad + 12\lambda\|\mathbf{W}\|^2_{2,4}\|\mathbf{W}_\star\|^2_{2,4} \\
&\overset{(i)}{\leq} \left\langle \nabla\widehat{\mathcal{R}}(f^Q_{\mathbf{W}}), \mathbf{W} \right\rangle - 2(\widehat{\mathcal{R}}(f^Q_{\mathbf{W}}) - M) + m^{-1}C^2 B^4_h\|\mathbf{W}\|^2_{2,4}\|\mathbf{W}_\star\|^2_{2,4} \\
&\quad + \lambda\|\mathbf{W}\|^4_{2,4} + 36\lambda\|\mathbf{W}_\star\|^4_{2,4} \\
&\leq \left\langle \nabla\widehat{\mathcal{R}}_\lambda(f^Q_{\mathbf{W}}), \mathbf{W} \right\rangle - 2(\widehat{\mathcal{R}}_\lambda(f^Q_{\mathbf{W}}) - M) + m^{-1}C^2 B^4_h\|\mathbf{W}\|^2_{2,4}\|\mathbf{W}_\star\|^2_{2,4} \\
&\quad - \lambda\|\mathbf{W}\|^4_{2,4} + 36\lambda\|\mathbf{W}_\star\|^4_{2,4}.
\end{aligned}$$

We used the fact $12ab \leq a^2 + 36b^2$. For a second order-stationary point $\widehat{\mathbf{W}}$ of $\widehat{\mathcal{R}}_\lambda(f^Q_{\mathbf{W}})$, its gradient vanishes and the Hessian is positive definite. Therefore, we have

$$0 \leq -2(\widehat{\mathcal{R}}_\lambda(f^Q_{\widehat{\mathbf{W}}}) - M) + m^{-1}C^2 B^4_h\left\|\widehat{\mathbf{W}}\right\|^2_{2,4}\|\mathbf{W}_\star\|^2_{2,4} - \lambda\left\|\widehat{\mathbf{W}}\right\|^4_{2,4} + 36\lambda\|\mathbf{W}_\star\|^4_{2,4}.$$

We choose $m = \epsilon^{-1}(2\lambda_0)^{-1/2}C^2 B_h^4 B_{w,\star}^4 \geq \epsilon^{-1}C^2 B_h^4 \|\widehat{\mathbf{W}}\|_{2,4}^2 \|\mathbf{W}_\star\|_{2,4}^2$ and the above inequality implies

$$2(\widehat{\mathcal{R}}_\lambda(f_{\widehat{\mathbf{W}}}^Q) - M) \leq 2\tau M + \epsilon + \epsilon$$
$$\implies \widehat{\mathcal{R}}_\lambda(f_{\widehat{\mathbf{W}}}^Q) \leq (1+\tau)M + \epsilon.$$

The proof is complete. $\qquad\qquad\qquad\qquad\qquad\qquad\qquad\qquad\qquad\qquad\qquad\qquad\qquad\square$

### A.2  Proof of Generalization in Theorem 1

*Proof of Theorem 1, Generalization Part.* Using symmetrization, we have

$$\mathbb{E}_{(\mathbf{x}_i,y_i)}\left[\sup_{\|\mathbf{W}\|_{2,4}\leq B_w} \left|\mathcal{R}(f_{\mathbf{W}}^Q) - \widehat{\mathcal{R}}(f_{\mathbf{W}}^Q)\right|\right] \leq 2\mathbb{E}_{(\mathbf{x}_i,y_i),\boldsymbol{\xi}}\left[\sup_{\|\mathbf{W}\|_{2,4}\leq B_w} \left|\frac{1}{n}\sum_{i=1}^n \xi_i \ell(f_{\mathbf{W}}^Q(\mathbf{x}_i),y_i)\right|\right],$$

where $\xi$ is i.i.d. Rademacher random variables. The above Rademacher complexity can be bounded using the contraction theorem [51, Chapter 5]:

$$\mathbb{E}_{(\mathbf{x}_i,y_i),\boldsymbol{\xi}}\left[\left|\sup_{\|\mathbf{W}\|_{2,4}\leq B_w} \frac{1}{n}\sum_{i=1}^n \xi_i \ell(y_i, f_{\mathbf{W}}^Q(\mathbf{x}_i))\right|\right]$$

$$= \mathbb{E}_{(\mathbf{x}_i,y_i),\boldsymbol{\xi}}\left[\sup_{\|\mathbf{W}\|_{2,4}\leq B_w} \max\left\{\frac{1}{n}\sum_{i=1}^n \xi_i \ell(y_i, f_{\mathbf{W}}^Q(\mathbf{x}_i)), -\frac{1}{n}\sum_{i=1}^n \xi_i \ell(y_i, f_{\mathbf{W}}^Q(\mathbf{x}_i))\right\}\right]$$

$$\leq \mathbb{E}_{(\mathbf{x}_i,y_i),\boldsymbol{\xi}}\left[\sup_{\|\mathbf{W}\|_{2,4}\leq B_w} \frac{1}{n}\sum_{i=1}^n \xi_i \ell(y_i, f_{\mathbf{W}}^Q(\mathbf{x}_i)) + \sup_{\|\mathbf{W}\|_{2,4}\leq B_w} \frac{1}{n}\sum_{i=1}^n -\xi_i \ell(y_i, f_{\mathbf{W}}^Q(\mathbf{x}_i))\right]$$

$$\leq 4\mathbb{E}_{(\mathbf{x}_i,y_i),\boldsymbol{\xi}}\left[\sup_{\|\mathbf{W}\|_{2,4}\leq B_w} \frac{1}{n}\sum_{i=1}^n \xi_i f_{\mathbf{W}}^Q(\mathbf{x}_i)\right] + 2\mathbb{E}_{(\mathbf{x}_i,y_i),\boldsymbol{\xi}}\left[\frac{1}{n}\sum_{i=1}^n \xi_i \ell(0, y_i)\right]$$

$$\leq 4\mathbb{E}_{\mathbf{x}_i,\boldsymbol{\xi}}\left[\sup_{\|\mathbf{W}\|_{2,4}\leq B_w} \frac{1}{\sqrt{m}}\sum_{r\leq m}\left\langle \frac{1}{n}\sum_{i=1}^n \xi_i a_r \phi''(\mathbf{w}_{0,r}^\top \mathbf{h}(\mathbf{x}_i))\mathbf{h}(\mathbf{x}_i)\mathbf{h}(\mathbf{x}_i)^\top, \mathbf{w}_r\mathbf{w}_r^\top\right\rangle\right] + \frac{2}{\sqrt{n}}$$

$$\leq 4\mathbb{E}_{\mathbf{x}_i,\boldsymbol{\xi}}\left[\sup_{\|\mathbf{W}\|_{2,4}\leq B_w} \max_{r\in[m]}\left\|\frac{1}{n}\sum_{i=1}^n \xi_i \phi''(\mathbf{w}_{0,r}^\top \mathbf{h}(\mathbf{x}_i))\mathbf{h}(\mathbf{x}_i)\mathbf{h}(\mathbf{x}_i)^\top\right\|_{\mathrm{op}} \cdot \frac{1}{\sqrt{m}}\sum_{r\leq m}\|\mathbf{w}_r\mathbf{w}_r^\top\|_*\right] + \frac{2}{\sqrt{n}}$$

$$\leq 4\mathbb{E}_{\mathbf{x}_i,\boldsymbol{\xi}}\left[\max_{r\in[m]}\left\|\frac{1}{n}\sum_{i=1}^n \xi_i \phi''(\mathbf{w}_{0,r}^\top \mathbf{h}(\mathbf{x})_i)\mathbf{h}(\mathbf{x}_i)\mathbf{h}(\mathbf{x}_i^\top)\right\|_{\mathrm{op}}\right] \cdot \underbrace{\sup_{\|\mathbf{W}\|_{2,4}\leq B_w} \frac{1}{\sqrt{m}}\sum_{r\leq m}\|\mathbf{w}_r\|_2^2}_{\leq B_w^2} + \frac{2}{\sqrt{n}},$$

where the last step used the power mean (or Cauchy-Schwarz) inequality on $\{\|\mathbf{w}_r\|_2\}$ and $\|\cdot\|_*$ denotes the matrix nuclear norm (sum of singular values). Now it only remains to bound the expected max operator norm above. We apply the matrix concentration lemma Bai and Lee [8, Lemma 8] to

deduce that

$$\mathbb{E}_{\mathbf{x}_i,\boldsymbol{\xi}}\left[\max_{r\in[m]}\left\|\frac{1}{n}\sum_{i=1}^n \xi_i\phi''(\mathbf{w}_{0,r}^\top\mathbf{h}(\mathbf{x}_i))\mathbf{h}(\mathbf{x}_i)\mathbf{h}(\mathbf{x}_i)^\top\right\|_{\mathrm{op}}\right]$$

$$\leq 4\sqrt{\log(2Dm)}\cdot\mathbb{E}_{\mathbf{x}_i}\left[\sqrt{\max_{r\in[m]}\left\|\frac{1}{n^2}\sum_{i=1}^n \phi''(\mathbf{w}_{0,r}^\top\mathbf{h}(\mathbf{x}_i))^2\left\|\mathbf{x}_i\right\|_2^2\mathbf{h}(\mathbf{x}_i)\mathbf{h}(\mathbf{x}_i)^\top\right\|_{\mathrm{op}}}\right]$$

$$\leq 4B_h\sqrt{\frac{\log(2Dm)}{n}}\cdot\mathbb{E}_{\mathbf{x}_i}\left[\sqrt{\max_{r,i}\phi''(\mathbf{w}_{0,r}^\top\mathbf{h}(\mathbf{x}_i))^2\cdot\left\|\frac{1}{n}\sum_{i=1}^n\mathbf{h}(\mathbf{x}_i)\mathbf{h}(\mathbf{x}_i)^\top\right\|_{\mathrm{op}}}\right]$$

$$\leq 4B_h\sqrt{\frac{\log(2Dm)}{n}}\left(\mathbb{E}_{\mathbf{x}_i}\left[\max_{r,i}\phi''(\mathbf{w}_{0,r}^\top\mathbf{h}(\mathbf{x}_i))^2\right]\cdot\mathbb{E}_{\mathbf{x}_i}\left[\left\|\frac{1}{n}\sum_{i=1}^n\mathbf{h}(\mathbf{x}_i)\mathbf{h}(\mathbf{x}_i)^\top\right\|_{\mathrm{op}}\right]\right)^{1/2}$$

$$\leq 4C^2 B_h\sqrt{\frac{\log(2Dm)}{n}}\mathbb{E}_{\mathbf{x}_i}\left[\left\|\frac{1}{n}\sum_{i=1}^n\mathbf{h}(\mathbf{x}_i)\mathbf{h}(\mathbf{x}_i)^\top\right\|_{\mathrm{op}}\right]^{1/2}.$$

Combining all the ingredients and substituting $M_{h,\mathrm{op}} = B_h^{-1}\mathbb{E}_{\mathbf{x}_i}\left[\left\|\frac{1}{n}\sum_{i=1}^n\mathbf{h}(\mathbf{x}_i)\mathbf{h}(\mathbf{x}_i)^\top\right\|_{\mathrm{op}}\right]^{1/2}$, the generalization error is bounded by

$$\mathbb{E}_{(\mathbf{x}_i,y_i)}\left[\sup_{\|\mathbf{W}\|_{2,4}\leq B_w}\left|\mathcal{R}(f_{\mathbf{W}}^Q)-\widehat{\mathcal{R}}(f_{\mathbf{W}}^Q)\right|\right]\leq\widetilde{O}\left(\frac{B_h^2 B_w^2 M_{h,\mathrm{op}}}{\sqrt{n}}\sqrt{\log(Dm)}+\frac{1}{\sqrt{n}}\right).$$

$\square$

# B   Results on Feature Covariance

## B.1   Technical tool

We first present a Lemma for relating the covariance of nonlinear random features to the covariance of certain polynomial bases, adapted from [27, Proposition 2].

**Lemma 1** (Covariance through polynomials)**.** Let $\mathbf{v}_i \overset{\mathrm{iid}}{\sim} \mathrm{Unif}(\mathbb{S}^{d-1})$ be random unit vectors for $i\in[D]$ and $\mathbf{V}=[\mathbf{v}_1,\ldots,\mathbf{v}_D]^\top\in\mathbb{R}^{D\times d}$.

(a) For any $k\geq 0$, suppose $D\leq O(d^{k+1-\delta})$ for some $\delta>0$, then we have with high probability as $d\to\infty$ that

$$\lambda_{\min}\left((\mathbf{V}\mathbf{V}^\top)^{\odot(k+1)}\right)\geq\frac{1}{2},$$

where $(\cdot)^{\odot k}$ is the Hadamard product: $(\mathbf{A}^{\odot k})_{ij}=\mathbf{A}_{ij}^k$.

(b) In the same setting as above, let $\mathbf{x}\sim\mathrm{Unif}(\mathbb{S}^{d-1})$ and define $\boldsymbol{\Sigma}\in\mathbb{R}^{D\times D}$ with

$$\boldsymbol{\Sigma}_{ij}=\mathbb{E}_{\mathbf{x}}\left[\mathbb{1}\left\{\mathbf{v}_i^\top\mathbf{x}\geq 0\right\}\mathbb{1}\left\{\mathbf{v}_j^\top\mathbf{x}\geq 0\right\}\right],$$

then we have

$$\lambda_{\min}(\boldsymbol{\Sigma})\geq\frac{1}{2}\left\|\mathsf{P}_{\geq k+1}\sigma_d\right\|_{L_2}^2,$$

where $\sigma_d:\mathrm{Unif}(\mathbb{S}^{d-1}(\sqrt{d}))\to\mathbb{R}$ is defined as $\sigma_d(\mathbf{x}):=\mathbb{1}\{x_1\geq 0\}$, and $\mathsf{P}_{\geq k+1}$ denotes the projection onto degree $\geq(k+1)$ polynomials under the base measure $\mathrm{Unif}(\mathbb{S}^{d-1}(\sqrt{d}))$.

## B.2 Lower bound on population covariance

We first present a lower bound when $\mathbf{x} \sim \mathrm{Unif}(\mathbb{S}^{d-1})$ is uniform on the sphere, and when the features are the biasless indicator random features.

**Lemma 2** (Lower bound of population covariance). *Let $\mathbf{x} \sim \mathrm{Unif}(\mathbb{S}^{d-1})$ and suppose we sample $\mathbf{v}_i \overset{\mathrm{iid}}{\sim} \mathsf{N}(\mathbf{0}, \mathbf{I}_d)$ for $1 \leq i \leq D$, where $D \leq O(d^K)$. Let $\boldsymbol{\Sigma} \in \mathbb{R}^{D \times D}$ be the (population) covariance matrix of the random features $\left\{ \mathbb{1}\left\{ \mathbf{v}_i^\top \mathbf{x} \geq 0 \right\} \right\}_{i \in [D]}$, that is,*

$$\boldsymbol{\Sigma}_{ij} := \mathbb{E}_{\mathbf{x} \sim \mathrm{Unif}(\mathbb{S}^{d-1})}\left[ \mathbb{1}\left\{ \mathbf{v}_i^\top \mathbf{x} \geq 0 \right\} \mathbb{1}\left\{ \mathbf{v}_j^\top \mathbf{x} \geq 0 \right\} \right],$$

*then we have $\lambda_{\min}(\boldsymbol{\Sigma}) \geq c > 0$ with high probability as $d \to \infty$, where $c = c_K$ is a constant that depends on $K$ (and the indicator activation) but not $d$.*

*Proof.* Let $\widetilde{\mathbf{v}}_i := \mathbf{v}_i / \|\mathbf{v}_i\|_2$ denote the normalized version of $\mathbf{v}_i$, then $\widetilde{\mathbf{v}}_i \overset{\mathrm{iid}}{\sim} \mathrm{Unif}(\mathbb{S}^{d-1})$ due to the spherical symmetry of $\mathsf{N}(\mathbf{0}, \mathbf{I}_d)$. Further using the positive homogeneity of $t \mapsto \mathbb{1}\left\{ t \geq 0 \right\}$ yields that

$$\begin{aligned}
\boldsymbol{\Sigma}_{ij} &= \mathbb{E}_{\mathbf{x} \sim \mathrm{Unif}(\mathbb{S}^{d-1})}\left[ \mathbb{1}\left\{ \widetilde{\mathbf{v}}_i^\top \mathbf{x} \geq 0 \right\} \mathbb{1}\left\{ \widetilde{\mathbf{v}}_j^\top \mathbf{x} \geq 0 \right\} \right] \\
&= \mathbb{E}_{\mathbf{x} \sim \mathrm{Unif}(\mathbb{S}^{d-1}(\sqrt{d}))}\left[ \mathbb{1}\left\{ \widetilde{\mathbf{v}}_i^\top \mathbf{x} \geq 0 \right\} \mathbb{1}\left\{ \widetilde{\mathbf{v}}_j^\top \mathbf{x} \geq 0 \right\} \right].
\end{aligned}$$

This falls into the setting of Lemma 1(b), applying which implies that with high probability (as $d \to \infty$) we have

$$\lambda_{\min}(\boldsymbol{\Sigma}) \geq \frac{1}{2} \left\| \mathsf{P}_{\geq K+1} \sigma_d \right\|_{L_2}^2,$$

where $\sigma_d : \mathbb{S}^{d-1}(\sqrt{d}) \to \mathbb{R}$ is defined as $\sigma_d(\mathbf{x}) = \mathbb{1}\left\{ x_1 \geq 0 \right\}$, $\mathsf{P}_{\geq K+1}$ denotes the projection onto degree-$\geq K+1$ polynomials under the base measure $\mathrm{Unif}(\mathbb{S}^{d-1}(\sqrt{d}))$, and the $L_2$ norm is under the same base measure. For large enough $d$, as $x_1 | \mathbf{x} \sim \mathrm{Unif}(\mathbb{S}^{d-1}(\sqrt{d})) \Rightarrow \mathsf{N}(0,1) := \gamma$ (where $\Rightarrow$ denotes convergence in distribution), this is further lower bounded by

$$\frac{1}{4} \left\| \mathsf{P}_{\geq K+1} \mathbb{1}\left\{ \cdot \geq 0 \right\} \right\|_{L_2(\gamma)}^2 = c_K > 0 \tag{10}$$

as the indicator function is not a polynomial of any degree (so that its $L_2$ projection onto polynomials of degree $\leq K$ is not itself for any $K \geq 0$).

**Decay of Eigenvalue Lower Bounds with Uniform Data**.

We now provide a lower bound for the quantity $\left\| \mathsf{P}_{\geq K+1} \mathbb{1}\left\{ \cdot \geq 0 \right\} \right\|_{L_2(\gamma)}^2$, thereby giving a lower bound on $c_K$ defined in (10). Indeed, we have

$$\left\| \mathsf{P}_{\geq K+1} \mathbb{1}\left\{ \cdot \geq 0 \right\} \right\|_{L_2(\gamma)}^2 = \sum_{j=K+1}^{\infty} \widehat{\sigma}_j^2,$$

where

$$\mathbb{1}\left\{ z \geq 0 \right\} \overset{L_2(\gamma)}{=} \sum_{j=0}^{\infty} \widehat{\sigma}_j h_j(z)$$

is the Hermite decomposition of $\mathbb{1}\left\{ z \geq 0 \right\}$. By [37], we know that

$$\widehat{\sigma}_0 = \frac{1}{2}, \quad \widehat{\sigma}_{2i+1} = (-1)^i \sqrt{\frac{1}{2\pi(2i+1)!}} \frac{(2i)!}{2^i i!}, \quad \widehat{\sigma}_{2i+2} = 0 \text{ for all } i \geq 0.$$

We now calculate the decay of $\widehat{\sigma}_{2i+1}^2$. By Stirling's formula, we have

$$\widehat{\sigma}_{2i+1}^2 = \frac{1}{2\pi(2i+1)!} \cdot \frac{(2i)!^2}{2^{2i}(i!)^2} \asymp \frac{1}{2\pi(2i+1)} \cdot \frac{\sqrt{2\pi \cdot 2i}(2i/e)^{2i}}{2^{2i} \cdot 2\pi i \cdot (i/e)^{2i}} \asymp Ci^{-3/2}$$

for some absolute constant $C > 0$. This means that for all $i \geq 0$ we have $\widehat{\sigma}_{2i+1}^2 \geq Ci^{-3/2}$ for some (other) absolute constant $C > 0$, which gives

$$\sum_{j=K+1}^{\infty} \widehat{\sigma}_j^2 \geq \sum_{i : 2i+1 \geq K+1}^{\infty} Ci^{-3/2} \geq CK^{-1/2}.$$

Therefore we have $c_K \geq \Omega(K^{-1/2})$ for all $K$. $\qquad\square$

**Covariance Lower Bounds for Non-uniform Data**.

We further show that Assumption 2 can hold fairly generally when $\mathbf{x}$ is no longer uniform on $\mathbb{S}^{d-1}$. Recall we choose $D \leq O(d^K)$ for some constant $K$.

We begin with assuming there exists a positive definite matrix $\mathbf{S} \in \mathbb{R}^{d \times d}$ such that $\mathbf{x}$ is equal in distribution as $\mathbf{S}^{1/2}\mathbf{z}/\left\|\mathbf{S}^{1/2}\mathbf{z}\right\|_2$, where $\mathbf{z} \sim \mathsf{N}(\mathbf{0}, \mathbf{I}_d)$. In other words, $\mathbf{x}$ is distributed as a rescaled version of a $d$-dimensional Gaussian with *arbitrary covariance*, a fairly expressive set of distributions which can model the case where $\mathbf{x}$ is far from uniform over the sphere. We show in this case that Assumption 2 holds, whenever $\mathbf{S}$ has a bounded condition number (i.e. $\lambda_{\min}(\mathbf{S})/\lambda_{\max}(\mathbf{S}) \geq 1/\kappa$ where $\kappa > 0$ does not depend on $d$).

Indeed, we can deduce

$$\mathbb{1}\{\mathbf{v}_j^\top \mathbf{x} \geq 0\} = \mathbb{1}\left\{\mathbf{v}_j^\top \frac{\mathbf{S}^{1/2}\mathbf{z}}{\left\|\mathbf{S}^{1/2}\mathbf{z}\right\|_2} \geq 0\right\} = \mathbb{1}\left\{(\mathbf{S}^{1/2}\mathbf{v}_j)^\top \mathbf{z} \geq 0\right\} = \mathbb{1}\left\{\frac{\mathbf{S}^{1/2}\mathbf{v}_j^\top}{\left\|\mathbf{S}^{1/2}\mathbf{v}_j\right\|_2}\mathbf{z} \geq 0\right\}.$$

Here the equality denotes two random variables following the same distribution. We apply the Hermite decomposition of indicator function to decompose the covariance matrix $\boldsymbol{\Sigma}$:

$$\lambda_{\min}(\boldsymbol{\Sigma}) = \min_{\|\mathbf{u}\|_2=1} \mathbb{E}_\mathbf{x}\left[\mathbf{u}^\top \mathbf{g}(\mathbf{x})\mathbf{g}(\mathbf{x})^\top \mathbf{u}\right]$$

$$= \min_{\|\mathbf{u}\|_2=1} \mathbb{E}_\mathbf{x}\left[\sum_{i,j} \mathbb{1}\{\mathbf{v}_i^\top \mathbf{x} \geq 0\}\mathbb{1}\{\mathbf{v}_j^\top \mathbf{x} \geq 0\}u_i u_j\right]$$

$$= \min_{\|\mathbf{u}\|_2=1} \sum_{i,j} \left(\mathcal{T}_1 + \mathcal{T}_2\right) u_i u_j, \tag{11}$$

where $\mathcal{T}_1$ and $\mathcal{T}_2$ are given as follows,

$$\mathcal{T}_1 = \sum_{\ell=0}^\infty \widehat{\sigma}_\ell^2 \mathbb{E}_\mathbf{z}\left[h_\ell\left(\frac{\mathbf{S}^{1/2}\mathbf{v}_j^\top}{\left\|\mathbf{S}^{1/2}\mathbf{v}_j\right\|_2}\mathbf{z}\right)h_\ell\left(\frac{\mathbf{S}^{1/2}\mathbf{v}_j^\top}{\left\|\mathbf{S}^{1/2}\mathbf{v}_j\right\|_2}\mathbf{z}\right)\right]$$

$$= \sum_{\ell=0}^\infty \widehat{\sigma}_\ell^2 \left(\frac{\mathbf{v}_i^\top \mathbf{S}\mathbf{v}_j}{\left\|\mathbf{S}^{1/2}\mathbf{v}_j\right\|_2 \left\|\mathbf{S}^{1/2}\mathbf{v}_i\right\|_2}\right)^\ell,$$

$$\mathcal{T}_2 = \sum_{\ell \neq k} \widehat{\sigma}_\ell \widehat{\sigma}_k \mathbb{E}_\mathbf{z}[h_\ell(\mathbf{v}_i^\top \mathbf{z})h_k(\mathbf{v}_j^\top \mathbf{z})] = 0,$$

where $\widehat{\sigma}_\ell$ is the coefficient of Hermite decomposition of the indicator function, and $\mathcal{T}_2$ vanishes, due to the orthogonality of probabilistic Hermite polynomials. We proceed to bound the minimum singular value of $\boldsymbol{\Sigma}$:

$$\lambda_{\min}(\boldsymbol{\Sigma}) = \min_{\|\mathbf{u}\|_2=1} \sum_{i,j} \sum_{\ell=0}^\infty \widehat{\sigma}_\ell^2 \left(\frac{\mathbf{v}_i^\top \mathbf{S}\mathbf{v}_j}{\left\|\mathbf{S}^{1/2}\mathbf{v}_j\right\|_2 \left\|\mathbf{S}^{1/2}\mathbf{v}_i\right\|_2}\right)^\ell u_i u_j.$$

Note that in the above decomposition, $\boldsymbol{\Sigma}$ is the sum of an infinite series of positive semidefinite matrices. To show $\boldsymbol{\Sigma}$ has a lower bounded smallest singular value, it suffices to show that there exists a summand in the infinite series being positive definite. We confirm this by analyzing the $\ell = K + 1$ summand. In fact, we show the following matrix $\boldsymbol{\Sigma}_{K+1} \in \mathbb{R}^{D \times D}$ is positive definite and its smallest singular value is lower bounded by some constant independent of $d$.

$$[\boldsymbol{\Sigma}_{K+1}]_{ij} = \widehat{\sigma}_{K+1}^2 \left(\frac{\mathbf{v}_i^\top \mathbf{S}\mathbf{v}_j}{\left\|\mathbf{S}^{1/2}\mathbf{v}_j\right\|_2 \left\|\mathbf{S}^{1/2}\mathbf{v}_i\right\|_2}\right)^{K+1} \quad \text{for} \quad i, j = 1, \ldots, D.$$

We denote the normalized $\mathbf{v}_j$ as $\widetilde{\mathbf{v}}_j = \mathbf{v}_j / \left\|\mathbf{v}_j\right\|_2$, and derive a lower bound on the singular value of $\widetilde{\boldsymbol{\Sigma}}_{K+1}$ with

$$[\widetilde{\boldsymbol{\Sigma}}_{K+1}]_{ij} = (\widetilde{\mathbf{v}}_i^\top \mathbf{S}\widetilde{\mathbf{v}}_j)^{K+1}.$$

Using the tensor product notation, we rewrite $\widetilde{\boldsymbol{\Sigma}}_{K+1}$ as

$$\widetilde{\boldsymbol{\Sigma}}_{K+1} = \widetilde{\mathbf{V}}^{*(K+1)} \mathbf{S}^{\otimes(K+1)} \left(\widetilde{\mathbf{V}}^{*(K+1)}\right)^{\top},$$

where $\widetilde{\mathbf{V}} = [\widetilde{\mathbf{v}}_1, \ldots, \widetilde{\mathbf{v}}_D]^{\top} \in \mathbb{R}^{D \times d}$, $\widetilde{\mathbf{V}}^{*(K+1)} \in \mathbb{R}^{D \times d^{K+1}}$ is the Khatri-Rao product, and $\mathbf{S}^{\otimes(K+1)} \in \mathbb{R}^{d^{K+1} \times d^{K+1}}$ denotes the Kronecker product. Then we know

$$\widetilde{\boldsymbol{\Sigma}}_{K+1} \succeq \lambda_{\min}^{K+1}(\mathbf{S}) \widetilde{\mathbf{V}}^{*(K+1)} \left(\widetilde{\mathbf{V}}^{*(K+1)}\right)^{\top}.$$

Moreover, using Lemma 1(a), we have $\lambda_{\min}\left(\widetilde{\mathbf{V}}^{*(K+1)} \left(\widetilde{\mathbf{V}}^{*(K+1)}\right)^{\top}\right) \geq 1/2$ as we picked $D \leq O(d^K)$. Substituting into $\boldsymbol{\Sigma}_{K+1}$, we have

$$\lambda_{\min}(\boldsymbol{\Sigma}_{K+1}) \geq \frac{1}{2}\widehat{\sigma}_{K+1}^2 \lambda_{\min}^{K+1}(\mathbf{S}) \left(\frac{\|\mathbf{v}_i\|_2 \|\mathbf{v}_j\|_2}{\left\|\mathbf{S}^{1/2}\mathbf{v}_i\right\|_2 \left\|\mathbf{S}^{1/2}\mathbf{v}_j\right\|_2}\right)^{K+1}$$

$$\geq \frac{1}{2}\widehat{\sigma}_{K+1}^2 \underbrace{\lambda_{\min}^{K+1}(\mathbf{S})\lambda_{\max}^{-(K+1)}(\mathbf{S})}_{\kappa^{-(K+1)}}.$$

Therefore the smallest singular value of $\boldsymbol{\Sigma}$ is lower bounded by $\Omega(\widehat{\sigma}_{K+1}^2 \kappa^{-(K+1)})$, which is a constant only depending on $K$ but not $d$. This finishes the proof. □

We remark that Assumption 2 can hold much more generally than rescaled Gaussian distributions, provided the following set of conditions holds: There exists some random variable $\mathbf{z} \in \mathbb{R}^d$ such that $\mathbf{x}$ is equal in distribution to $\mathbf{z}/\|\mathbf{z}\|_2$, a set of univariate polynomials $\{h_k^{(i)} : \mathbb{R} \to \mathbb{R}\}$ for $i = 1, \ldots, D$ and $k \geq 0$ where each $h_k^{(i)}$ is a degree-$k$ polynomial "assigned" to $\mathbf{v}_i$, and corresponding coefficients $\widehat{\sigma}_k^{(i)} \in \mathbb{R}$, such that

$$\mathbb{1}\left\{\mathbf{v}_i^{\top}\mathbf{z} \geq 0\right\} \overset{L_2}{=} \sum_{k=0}^{\infty} \widehat{\sigma}_k^{(i)} h_k^{(i)}(\mathbf{v}_i^{\top}\mathbf{z}).$$

Let $\boldsymbol{\Sigma}_k, \boldsymbol{\Sigma}_{k,>k}, \boldsymbol{\Sigma}_{>k} \in \mathbb{R}^{D \times D}$ be defined as

$$[\boldsymbol{\Sigma}_k]_{ij} := \mathbb{E}_{\mathbf{z}}\left[\widehat{\sigma}_k^{(i)}\widehat{\sigma}_k^{(j)} h_k^{(i)}(\widetilde{\mathbf{v}}_i^{\top}\mathbf{z})h_k^{(j)}(\widetilde{\mathbf{v}}_j^{\top}\mathbf{z})\right]$$

$$[\boldsymbol{\Sigma}_{k,>k}]_{ij} := \mathbb{E}_{\mathbf{z}}\left[\widehat{\sigma}_k^{(i)} h_k^{(i)}(\widetilde{\mathbf{v}}_i^{\top}\mathbf{z}) \cdot \sum_{\ell>k} \widehat{\sigma}_\ell^{(j)} h_\ell^{(j)}(\widetilde{\mathbf{v}}_j^{\top}\mathbf{z})\right]$$

$$[\boldsymbol{\Sigma}_{>k}]_{ij} := \mathbb{E}_{\mathbf{z}}\left[\sum_{\ell>k} \widehat{\sigma}_\ell^{(i)} h_\ell^{(i)}(\widetilde{\mathbf{v}}_i^{\top}\mathbf{z}) \cdot \sum_{\ell>k} \widehat{\sigma}_\ell^{(j)} h_\ell^{(j)}(\widetilde{\mathbf{v}}_j^{\top}\mathbf{z})\right],$$

where $\widetilde{\mathbf{v}}_i := \mathbf{v}_i/\|\mathbf{v}_i\|_2$. We assume we have

- The not-too-correlated condition: there exists some $\epsilon \in (0,1]$ such that for any $k \geq 0$, we have

$$\boldsymbol{\Sigma}_{\geq k} := \boldsymbol{\Sigma}_k + \boldsymbol{\Sigma}_{k,>k} + \boldsymbol{\Sigma}_{k,>k}^{\top} + \boldsymbol{\Sigma}_{>k} \succeq \epsilon(\boldsymbol{\Sigma}_k + \boldsymbol{\Sigma}_{>k}).$$

- For all large $d$ and $D \leq O(d^k)$, we have

$$\boldsymbol{\Sigma}_{k+1} \succeq c_{k+1}(\widetilde{\mathbf{V}}\widetilde{\mathbf{V}}^{\top})^{\odot(k+1)},$$

where $c_{k+1} > 0$ is a constant that depends on $k$ but not $d$.

In this case, we can deduce that for $D = O(d^K)$, Assumption 2 holds with $\lambda_K = \Omega(\epsilon^{K+2}c_{K+1})$. To see this, observe that we have the expansion $\mathbb{1}\{t \geq 0\} = \widehat{\sigma}_0^{(i)} h_0^{(i)}(t) + \sum_{\ell>0} \widehat{\sigma}_\ell^{(i)} h_\ell^{(i)}(t)$ for all $i = 1, \ldots, D$, and thus

$$\boldsymbol{\Sigma} = \boldsymbol{\Sigma}_0 + \boldsymbol{\Sigma}_{0,>0} + \boldsymbol{\Sigma}_{0,>0}^{\top} + \boldsymbol{\Sigma}_{>0,>0} \overset{(i)}{\succeq} \epsilon(\boldsymbol{\Sigma}_0 + \boldsymbol{\Sigma}_{>0}) \succeq \epsilon\boldsymbol{\Sigma}_{>0},$$

where $(i)$ applied the not-too-correlated condition. Repeating the above process for $K$ times leads to

$$\mathbf{\Sigma} \succeq \epsilon^{K+1} \mathbf{\Sigma}_{>K} \succeq \epsilon^{K+2}(\mathbf{\Sigma}_{K+1} + \mathbf{\Sigma}_{>(K+1)}) \succeq \epsilon^{K+2}\mathbf{\Sigma}_{K+1} \succeq \epsilon^{K+2}c_{K+1}(\widetilde{\mathbf{V}}\widetilde{\mathbf{V}}^{\top})^{\odot(K+1)}.$$

Combining with existing lower bound $\lambda_{\min}((\widetilde{\mathbf{V}}\widetilde{\mathbf{V}}^{\top})^{\odot(K+1)}) \geq 1/2$ (Lemma 1(a)), we see Assumption 2 holds with $\lambda_K = \Omega(\epsilon^{K+2}c_{K+1})$, a constant that depends on $K$ and independent of $d$.

We further note that the above two conditions are all satisfied by the rescaled Gaussian distributions: Choosing $h_k^{(i)} \equiv h_k$ (the $k$-th Hermite polynomial) for all $i = 1, \ldots, D$, the first condition holds with $\epsilon = 1$ since $\mathbf{\Sigma}_{k,>k} = \mathbf{0}$, and the second condition holds with $c_{k+1} = (\lambda_{\min}(\mathbf{S})/\lambda_{\max}(\mathbf{S}))^{k+1}$ (as shown earlier). Combining with the fact they only assume things about the moments of $\mathbf{x}$ (or $\mathbf{z}$; since $h_k^{(i)}$ are polynomials), we see that they are indeed moment-based assumptions that contain Gaussian distributions with arbitrary covariances, and thus can be fairly general.

We also remark that while we have verified Assumption 2 for random features without biases, our analyses can be straightforwardly generalized to the case with bias by looking at the augmented input $[\mathbf{x}^{\top}, 1]^{\top} \in \mathbb{R}^{d+1}$ and analyzing its distributions in similar fashions.

## B.3 Relative concentration of covariance estimator

**Lemma 3** (Relative concentration of covariance estimator). *Let $\{\mathbf{g}(\mathbf{x}_i) \in \mathbb{R}^D\}_{i=1}^n$ be i.i.d. random vectors such that $\|\mathbf{g}_1\|_2 \leq B_g$ almost surely and $\mathbb{E}[\mathbf{g}_1\mathbf{g}_1^{\top}] = \mathbf{\Sigma} \succeq \lambda_{\min}\mathbf{I}_D$. Let $\widehat{\mathbf{\Sigma}} := \frac{1}{n}\sum_{i=1}^n \mathbf{g}(\mathbf{x}_i)\mathbf{g}(\mathbf{x}_i)^{\top}$ denote the empirical covariance matrix of $\{\mathbf{g}(\mathbf{x}_i)\}$. For any $\epsilon \in (0, 1)$, as soon as $n \geq C\epsilon^{-2}\lambda_{\min}^{-1}B_g^2 \log(n \vee D)$, we have*

$$\mathbb{E}\left[\left\|\mathbf{\Sigma}^{-1/2}\widehat{\mathbf{\Sigma}}\mathbf{\Sigma}^{-1/2} - \mathbf{I}_D\right\|_{\mathrm{op}}\right] \leq \epsilon.$$

*Further, when $n \geq C\delta^{-2}\epsilon^{-2}\lambda_{\min}^{-1}B_g^2 \log(n \vee D)$ we have with probability at least $1 - \delta$ that*

$$\left\|\mathbf{\Sigma}^{-1/2}\widehat{\mathbf{\Sigma}}\mathbf{\Sigma}^{-1/2} - \mathbf{I}_D\right\|_{\mathrm{op}} \leq \epsilon,$$

*where $C > 0$ is a universal constant. On the same event, we have the relative concentration*

$$(1 - \epsilon)\mathbf{\Sigma} \preceq \widehat{\mathbf{\Sigma}} \preceq (1 + \epsilon)\mathbf{\Sigma}.$$

*Proof.* The first statement directly yields the second by the Markov inequality. To see how the second statement implies the third, we can left- and right- multiply the matrix inside by $\mathbf{\Sigma}^{1/2}\mathbf{v}$ for any $\mathbf{v} \in \mathbb{R}^D$ and get that

$$\left|(\mathbf{v}^{\top}\mathbf{\Sigma}^{1/2})\mathbf{\Sigma}^{-1/2}\widehat{\mathbf{\Sigma}}\mathbf{\Sigma}^{-1/2}(\mathbf{\Sigma}^{1/2}\mathbf{v}) - \mathbf{v}^{\top}\mathbf{\Sigma}\mathbf{v}\right| = \left|\mathbf{v}^{\top}\widehat{\mathbf{\Sigma}}\mathbf{v} - \mathbf{v}^{\top}\mathbf{\Sigma}\mathbf{v}\right| \leq \epsilon\mathbf{v}^{\top}\mathbf{\Sigma}\mathbf{v},$$

which implies that $(1 - \epsilon)\mathbf{\Sigma} \preceq \widehat{\mathbf{\Sigma}} \leq (1 + \epsilon)\mathbf{\Sigma}$.

We now prove the first statement, which builds on the following Rudelson's inequality for controlling expected deviation of heavy-tailed sample covariance matrices:

**Lemma 4** (Restatement of Theorem 5.45, [50]). *Let $\{\mathbf{a}_i \in \mathbb{R}^D\}_{i=1}^n$ be independent random vectors with $\mathbb{E}[\mathbf{a}_i\mathbf{a}_i^{\top}] = \mathbf{I}_D$. Let $\Gamma := \mathbb{E}[\max_{i \in [n]} \|\mathbf{a}_i\|_2^2]$. Then there exists a universal constant $C > 0$ such that letting $\delta := C\Gamma \log(n \vee D)/n$, we have*

$$\mathbb{E}\left[\left\|\frac{1}{n}\sum_{i=1}^n \mathbf{a}_i\mathbf{a}_i^{\top} - \mathbf{I}_D\right\|_{\mathrm{op}}\right] \leq \delta \vee \sqrt{\delta}.$$

We will apply Lemma 4 on the whitened random vectors $\mathbf{h}(\mathbf{x}_i) := \mathbf{\Sigma}^{-1/2}\mathbf{g}(\mathbf{x}_i)$ (Here we slightly abuse the notation to denote $\mathbf{h}(\mathbf{x})$ as the whitened feature using the population covariance matrix).

Clearly, $\mathbb{E}_{\mathbf{x}}[\mathbf{h}(\mathbf{x}_i)\mathbf{h}(\mathbf{x}_i)^\top] = \mathbb{E}_{\mathbf{x}}[\mathbf{\Sigma}^{-1/2}\mathbf{g}(\mathbf{x}_i)\mathbf{g}(\mathbf{x}_i)^\top\mathbf{\Sigma}^{-1/2}] = \mathbf{\Sigma}^{-1/2}\mathbf{\Sigma}\mathbf{\Sigma}^{-1/2} = \mathbf{I}_D$. Further, we have

$$\|\mathbf{h}(\mathbf{x}_i)\|_2^2 = \mathbf{g}(\mathbf{x}_i)^\top\mathbf{\Sigma}^{-1}\mathbf{g}(\mathbf{x}_i)^\top \leq \lambda_{\min}^{-1}\|\mathbf{g}(\mathbf{x}_i)\|_2^2 \leq \lambda_{\min}^{-1}B_g^2$$

almost surely, and thus $\Gamma := \mathbb{E}_{\mathbf{x}}[\max_{i \in [n]}\|\mathbf{h}(\mathbf{x}_i)\|_2^2] \leq \lambda_{\min}^{-1}B_g^2$. Therefore, $\{\mathbf{h}(\mathbf{x}_i)\}$ satisfy the conditions of Lemma 4, from which we obtain

$$\mathbb{E}_{\mathbf{x}}\left[\left\|\mathbf{\Sigma}^{-1/2}\widehat{\mathbf{\Sigma}}\mathbf{\Sigma}^{-1/2} - \mathbf{I}_D\right\|_{\mathrm{op}}\right]$$

$$= \mathbb{E}_{\mathbf{x}}\left[\left\|\frac{1}{n}\sum_{i=1}^{n}\mathbf{h}(\mathbf{x}_i)\mathbf{h}(\mathbf{x}_i)^\top - \mathbf{I}_D\right\|_{\mathrm{op}}\right] \leq \frac{C\lambda_{\min}^{-1}B_g^2\log(n\vee D)}{n} \vee \sqrt{\frac{C\lambda_{\min}^{-1}B_g^2\log(n\vee D)}{n}}.$$

Therefore, setting $n \geq C\epsilon^{-2}\lambda_{\min}^{-1}B_g^2\log(n\vee D)$, we get that

$$\mathbb{E}_{\mathbf{x}}\left[\left\|\mathbf{\Sigma}^{-1/2}\widehat{\mathbf{\Sigma}}\mathbf{\Sigma}^{-1/2} - \mathbf{I}_D\right\|_{\mathrm{op}}\right] \leq \epsilon.$$

This finishes the proof. $\qquad\square$

## C  Proof of Warm Up Example

We denote $\{Y_{\ell,j}^d(\boldsymbol{x})\}_j$ as collections of spherical harmonics of degree $\ell$ on unit sphere $\mathbb{S}^{d-1}$. Subscript $j$ indexes the harmonics, and $j = 1,\ldots,B(d,\ell)$ with $B(d,\ell) = \frac{2\ell+d-2}{\ell}\binom{\ell+d-3}{\ell-1}$. We omit the superscript $d$ when it is clear from the context. The spherical harmonics form an orthonormal basis of $V(d,\ell)$, where $V(d,\ell)$ denotes the span of restrictions of degree-$\ell$ homogeneous harmonic polynomials to the unit sphere $\mathbb{S}^{d-1}$. In particular, we have

$$\int_{\mathbb{S}^{d-1}} Y_{\ell,i}(\mathbf{x})Y_{k,j}(\mathbf{x})\tau_{d-1}d\mathbf{x} = \delta_{ij}\delta_{k\ell} \quad \text{with } \tau_{d-1} \text{ the uniform probability measure on } \mathbb{S}^{d-1}.$$

For a clear presentation, we consider a single nonzero component in $B(r_\star, p_{\max}, A)$ denoted by $\alpha(\boldsymbol{\beta}^\top\mathbf{x})^p$ with $\|\boldsymbol{\beta}\|_2 < \infty$. Fix $k = p/2$. We choose $h(\mathbf{x}) \in \mathbb{R}^{B(d,k)}$ by taking $\{Y_{k,i}(\mathbf{x})\}$ as entries with $j = 1,\ldots,B(d,k)$ for $k = 0,\ldots,p/2$. Since $\{Y_{k,i}(\mathbf{x})\}_{i=1}^{B(d,k)}$ is a basis of $V(d,k)$, there exists $\boldsymbol{\theta} \in \mathbb{R}^{B(d,k)}$ satisfying $\boldsymbol{\theta}^\top h(\mathbf{x}) = (\boldsymbol{\beta}^\top\mathbf{x})^k$ for any $\mathbf{x} \in \mathbb{S}^{d-1}$. Moreover, each entry of $\boldsymbol{\theta}$ is given by

$$\theta_i = \int_{\mathbb{S}^{d-1}} Y_{k,i}(\mathbf{x})(\boldsymbol{\beta}^\top\mathbf{x})^k\tau_{d-1}d\mathbf{x}.$$

We can also check that $\|\boldsymbol{\theta}\|_2 \leq \|\boldsymbol{\beta}\|_2^k$ by observing $\int_{\mathbb{S}^{d-1}}\boldsymbol{\theta}^\top h(\mathbf{x})h(\mathbf{x})^\top\boldsymbol{\theta}\tau_{d-1}d\mathbf{x} = \int_{\mathbb{S}^{d-1}}(\boldsymbol{\beta}^\top\mathbf{x})^{2k}\tau_{d-1}d\mathbf{x}$.

Given $\boldsymbol{\theta}$ in the previous paragraph, we further choose $a(\mathbf{w}_0) = 2\alpha$ and $\mathbf{w}^* = \boldsymbol{\theta}$. By the symmetry in $\mathbf{w}_0$, it yields

$$\mathbb{E}_{\boldsymbol{w}_0 \sim \mathsf{N}(\mathbf{0},\mathbf{I})}\left[a(\mathbf{w}_0)\mathbb{1}\{\mathbf{w}_0^\top h(\mathbf{x}) \geq 0\}((\mathbf{w}^*)^\top h(\mathbf{x}))^2\right] = \alpha(\boldsymbol{\beta}^\top\mathbf{x})^p.$$

We now pick some $m$ sufficiently large, and let $\mathbf{w}_r^* = \sqrt{2\alpha}m^{1/4}\mathbf{w}^*$ for $r = 1,\ldots,m$. We check that $\{\mathbf{w}_r^*\}_{r=1}^m$ can achieve a small regularized empirical risk. Indeed, we have

$$\frac{1}{n}\sum_{i=1}^{n}\ell(y_i, f_{\mathbf{w}^*}(\mathbf{x}_i)) \leq \frac{1}{n}\sum_{i=1}^{n}\left|f_{\mathbf{w}^*}(\mathbf{x}_i) - \alpha(\boldsymbol{\beta}^\top\mathbf{x}_i)^p\right|$$

$$\leq \sup_{\mathbf{x}\in\mathbb{S}^{d-1}}\left|f_{\mathbf{w}^*}(\mathbf{x}) - \mathbb{E}_{\boldsymbol{w}_0\sim\mathsf{N}(\mathbf{0},\mathbf{I})}\left[a(\mathbf{w}_0)\mathbb{1}\{\mathbf{w}_0^\top h(\mathbf{x}) \geq 0\}((\mathbf{w}^*)^\top h(\mathbf{x}))^2\right]\right|$$

$$\leq \sup_{\mathbf{x}} 2\alpha(\boldsymbol{\beta}^\top\mathbf{x})^p\left|\frac{1}{m}\sum_{r=1}^{m}\mathbb{1}\{\mathbf{w}_{0,r}^\top h(\mathbf{x}) \geq 0\} - \frac{1}{2}\right|.$$

By Markov's inequality, for any $t > 0$, we derive

$$\mathbb{P}\left(\left|\frac{1}{m}\sum_{r=1}^m \mathbb{1}\{\mathbf{w}_{0,r}^\top h(\mathbf{x}) \geq 0\} - \frac{1}{2}\right| \geq t\right) \leq \frac{\mathbb{E}\left[\frac{1}{m}\sum_{r=1}^m \mathbb{1}\{\mathbf{w}_{0,r}^\top h(\mathbf{x}) \geq 0\} - \frac{1}{2}\right]^2}{t^2}$$

$$= \frac{1}{4mt^2}.$$

On the other hand, the norm of $\{\mathbf{w}_r^*\}_{r=1}^m$ satisfies

$$\sum_{r=1}^m \|\mathbf{w}_r^*\|_2^4 = \sum_{r=1}^m \frac{4\alpha^2}{m}\|\mathbf{w}^*\|_2^4 = 4\alpha^2 \|\boldsymbol{\theta}\|_2^4 \leq 4\alpha^2 \|\boldsymbol{\beta}\|_2^p.$$

Therefore, for a fixed optimization accuracy $\epsilon > 0$, we choose $m = O(\epsilon^{-3})$ and $\lambda$ some constant. This yields $\mathcal{R}_{\ell,\lambda}(f_{\mathbf{w}^*}) \leq \epsilon$.

Considering any second order stationary point $\{\widehat{\mathbf{w}}_r\}_{r=1}^m$, Theorem **??** gives rise to the following generalization bound.

$$\mathbb{E}_{\mathbf{w}_0}\left[\mathcal{R}_\ell(f_{\widehat{\mathbf{w}}}) - \mathbb{E}_{\mathbf{x},y}[\mathcal{R}_\ell(f_{\widehat{\mathbf{w}}})]\right] \leq \widetilde{O}\left(\frac{B_{\mathbf{w}}^2 M_{h(\mathbf{x}),\mathrm{op}}}{\sqrt{n}} + \frac{1}{\sqrt{n}}\right),$$

It suffices to find $B_{\mathbf{w}}$ and $M_{h(\mathbf{x}),\mathrm{op}}$. We already checked $B_{\mathbf{w}}^4 = \sum_{r=1}^m \|\mathbf{w}_r^*\|_2^4 \leq 2\alpha \|\boldsymbol{\beta}\|_2^p = O(1)$. For $M_{h(\mathbf{x}),\mathrm{op}}$, we expand it as

$$M_{h(\mathbf{x}),\mathrm{op}}^2 = \mathbb{E}_{\mathbf{x}}\left[\left\|\frac{1}{n}\sum_{i=1}^n h(\mathbf{x}_i)h(\mathbf{x}_i)^\top h(\mathbf{x}_i)h(\mathbf{x}_i)^\top\right\|_{\mathrm{op}}\right].$$

By the orthogonality of spherical harmonics, we know

$$\mathbb{E}_{\mathbf{x}}\left[h(\mathbf{x})h(\mathbf{x})^\top h(\mathbf{x})h(\mathbf{x})^\top\right] = B(d,k)\mathbf{I}.$$

Hence, we have $M_{h(\mathbf{x}),\mathrm{op}}^2 = O(B(d,k)) = \widetilde{O}(d^k)$. Substituting into the generalization bound, we deduce

$$\mathbb{E}_{\mathbf{w}_0}\left[\mathcal{R}_\ell(f_{\widehat{\mathbf{w}}}) - \mathbb{E}_{\mathbf{x},y}[\mathcal{R}_\ell(f_{\widehat{\mathbf{w}}})]\right] \leq \widetilde{O}\left(\sqrt{\frac{d^k}{n}} + \frac{1}{\sqrt{n}}\right).$$

From a statistical recovery perspective, to achieve $\epsilon$ generalization gap, the sample complexity grows in the order of $d^{p/2}/\epsilon^2$.

To learn multiple polynomials in $B(r_\star, p_{\max}, A)$, we dilate $h(\mathbf{x})$ as concatenating spherical harmonics up to order $k = p_{\max}/2$, i.e., $h(\mathbf{x}) = [\{Y_{0,j}(\mathbf{x})\}_{j=1}^{B(d,0)}, \ldots, \{Y_{k,j}(\mathbf{x})\}_{j=1}^{B(d,k)}]^\top$. With a slight modification on the argument for learning a single polynomial, we can show that the sample complexity grows as $\widetilde{O}(d^{p_{\max}/2}/\epsilon^2)$.

# D   Proofs for Section 4

This section devotes to the proof of Theorem 2. The proof consists of two main parts: expressivity of neural representation (Sections D.1 and D.2) and generalization property of `Quad-Neural` (Section D.3). Besides, Section D.5 presents that using data dependent regularizer also achieves improved sample complexity.

## D.1   Expressivity of Neural Random Features

**Lemma 5.** For a given vector $\boldsymbol{\beta}$ and integer $k \geq 0$, we let $\mathbf{v} \sim \mathsf{N}(\mathbf{0}, \mathbf{I}_d)$ be a standard Gaussian vector and $b \sim \mathsf{N}(0,1)$ independent of $\mathbf{v}$. Then there exists $a(\mathbf{v}, b)$ such that $\mathbb{E}_{\mathbf{v},b}[a(\mathbf{v}, b)\mathbb{1}\{\mathbf{v}^\top \mathbf{x} + b \geq 0\}] = (\boldsymbol{\beta}^\top \mathbf{x})^k$ holds for any $\mathbf{x} \in \mathbb{S}^{d-1}$.

*Proof.* We denote by $H_j(x)$ the $j$-th probabilistic Hermite polynomial. We pick

$$a(\mathbf{v}, b) = \begin{cases} c_k H_k(\mathbf{v}^\top \boldsymbol{\beta}/\|\boldsymbol{\beta}\|_2)\mathbb{1}\{0 < -b < 1/(2k)\}, & \text{if } k \text{ is even} \\ c_k H_k(\mathbf{v}^\top \boldsymbol{\beta}/\|\boldsymbol{\beta}\|_2)\mathbb{1}\{|b| < 1/(2k)\}, & \text{if } k \text{ is odd} \end{cases},$$

where $c_k$ is a constant to be determined. For a fixed $\mathbf{x}$, we denote $z_1 = \mathbf{v}^\top \boldsymbol{\beta}/\|\boldsymbol{\beta}\|_2$ and $z_2 = \mathbf{v}^\top \mathbf{x}$. It is straightforward to check that $z_1, z_2$ is jointly Gaussian with zero mean and $\mathbb{E}[z_1 z_2] = \boldsymbol{\beta}^\top \mathbf{x}/\|\boldsymbol{\beta}\|_2$. We can now deduce that $z_1$ and $(\boldsymbol{\beta}^\top \mathbf{x}/\|\boldsymbol{\beta}\|_2)z_2 + \sqrt{1 - (\boldsymbol{\beta}^\top \mathbf{x}/\|\boldsymbol{\beta}\|_2)^2}z_3$ follow the same distribution, where $z_3$ is standard Gaussian independent of $z_1$ and $z_2$. For an even $k$, we can check

$$\begin{aligned}
&\mathbb{E}_{\mathbf{v},b}[a(\mathbf{v},b)\mathbb{1}\{\mathbf{v}^\top\mathbf{x}+b \geq 0\}] \\
=\ & c_k\mathbb{E}_{z_1,z_2,b}[H_k(z_1)\mathbb{1}\{z_2 + b \geq 0\}\mathbb{1}\{0 < -b < 1/(2k)\}] \\
=\ & c_k\mathbb{E}_{z_2,z_3,b}\left[H_k\left((\boldsymbol{\beta}^\top\mathbf{x}/\|\boldsymbol{\beta}\|_2)z_2 + \sqrt{1 - (\boldsymbol{\beta}^\top\mathbf{x}/\|\boldsymbol{\beta}\|_2)^2}z_3\right)\mathbb{1}\{z_2 + b \geq 0\}\mathbb{1}\{0 < -b < 1/(2k)\}\right] \\
=\ & c_k\mathbb{E}_b\mathbb{E}_{z_2,z_3}\left[H_k\left((\boldsymbol{\beta}^\top\mathbf{x}/\|\boldsymbol{\beta}\|_2)z_2 + \sqrt{1 - (\boldsymbol{\beta}^\top\mathbf{x}/\|\boldsymbol{\beta}\|_2)^2}z_3\right)\mathbb{1}\{z_2 \geq -b\}\mathbb{1}\{0 < -b < 1/(2k)\}\mid b\right] \\
\overset{(i)}{=}\ & c_k q_k (\boldsymbol{\beta}^\top\mathbf{x})^k\|\boldsymbol{\beta}\|_2^{-k},
\end{aligned}$$

where $q_k = \mathbb{E}_b\left[(k-1)!!\frac{\exp(-b^2/2)}{\sqrt{2\pi}}\mathbb{1}\{0 < -b < 1/(2k)\}\sum_{j=1,\text{ odd}}^{k-1}\frac{(-1)^{(k-1+j)/2}}{j!!}\binom{k/2-1}{(j-1)/2}b^j\right]$.
The equality $(i)$ invokes *Lemma A.6* in Allen-Zhu et al. [4]. Similarly, for an odd $k$, we have

$$\mathbb{E}_{\mathbf{v},b}[a(\mathbf{v},b)\mathbb{1}\{\mathbf{v}^\top\mathbf{x}+b \geq 0\}] = c_k q_k (\boldsymbol{\beta}^\top\mathbf{x})^k\|\boldsymbol{\beta}\|_2^{-k}$$

with $q_k = \mathbb{E}_b\left[(k-1)!!\frac{\exp(-b^2/2)}{\sqrt{2\pi}}\mathbb{1}\{|b| \leq 1/(2k)\}\sum_{j=1,\text{ even}}^{k-1}\frac{(-1)^{(k-1+j)/2}}{j!!}\binom{k/2-1}{(j-1)/2}b^j\right]$. Here we unify the notation to denote $q_k$ as the coefficient for both the even and odd $k$'s. Using *Claim C.1* in Allen-Zhu et al. [4], we can lower bound $p_k$ by $|p_k| \geq \frac{(k-1)!!}{200k^2}$. The proof is complete by choosing $c_k = 1/p_k$, and accordingly, $|c_k| \leq \frac{200k^2}{(k-1)!!}\|\boldsymbol{\beta}\|_2^k$. $\qquad\square$

**From Expectation to Finite Neuron Approximation**.

**Lemma 6.** For a given $\epsilon > 0$ and $\delta > 0$, we choose $D = 2\times 200^2 k^5 \|\boldsymbol{\beta}\|_2^{2k}/(\epsilon^2\delta)$ and independently generate $\mathbf{v}_j \sim \mathsf{N}(\mathbf{0}, \mathbf{I}_d)$ and $b_j \sim \mathsf{N}(0,1)$ for $j = 1, \ldots, D$. Then with probability at least $1 - \delta$, we have

$$\left\|\frac{1}{D}\sum_{j=1}^{D}a(\mathbf{v}_j, b_j)\mathbb{1}\{\mathbf{v}_j^\top\mathbf{x}+b_j \geq 0\} - (\boldsymbol{\beta}^\top\mathbf{x})^k\right\|_{L_2} \leq \epsilon.$$

*Proof.* The desired bound can be obtained by Chebyshev's inequality. We bound the second moment of the $L_2$ norm as

$$\begin{aligned}
&\mathbb{E}_{\mathbf{v},b}\left\|\frac{1}{D}\sum_{j=1}^{D}a(\mathbf{v}_j, b_j)\mathbb{1}\{\mathbf{v}_j^\top\mathbf{x}+b_j \geq 0\} - (\boldsymbol{\beta}^\top\mathbf{x})^k\right\|_{L_2}^2 \\
=\ & \mathbb{E}_{\mathbf{v},b}\mathbb{E}_{\mathbf{x}}\left[\frac{1}{D}\sum_{j=1}^{D}a(\mathbf{v}_j, b_j)\mathbb{1}\{\mathbf{v}_j^\top\mathbf{x}+b_j \geq 0\} - (\boldsymbol{\beta}^\top\mathbf{x})^k\right]^2 \\
=\ & \mathbb{E}_{\mathbf{x}}\mathbb{E}_{\mathbf{v},b}\left[\frac{1}{D}\sum_{j=1}^{D}\left(a(\mathbf{v}_j, b_j)\mathbb{1}\{\mathbf{v}_j^\top\mathbf{x}+b_j \geq 0\} - \mathbb{E}_{\mathbf{v}_j,b_j}[a(\mathbf{v}_j, b_j)\mathbb{1}\{\mathbf{v}_j^\top\mathbf{x}+b_j \geq 0\}]\right)\right]^2 \\
=\ & \frac{1}{D^2}\mathbb{E}_{\mathbf{x}}\left[\sum_{j=1}^{D}\mathbb{E}_{\mathbf{v}_j,b_j}\left[a(\mathbf{v}_j, b_j)\mathbb{1}\{\mathbf{v}_j^\top\mathbf{x}+b_j \geq 0\} - \mathbb{E}_{\mathbf{v}_j,b_j}[a(\mathbf{v}_j, b_j)\mathbb{1}\{\mathbf{v}_j^\top\mathbf{x}+b_j \geq 0\}]\right]^2\right] \\
=\ & \frac{1}{D}\mathbb{E}_{\mathbf{x},\mathbf{v},b}\left[a(\mathbf{v},b)\mathbb{1}\{\mathbf{v}^\top\mathbf{x}+b \geq 0\} - (\boldsymbol{\beta}^\top\mathbf{x})^k\right]^2.
\end{aligned}$$

Using Lemma 5, we have

$$\mathbb{E}_{\mathbf{x},\mathbf{v},b}\left[a(\mathbf{v},b)\mathbb{1}\{\mathbf{v}^\top\mathbf{x}+b\geq 0\}-(\boldsymbol{\beta}^\top\mathbf{x})^k\right]^2 = \mathbb{E}_{\mathbf{x},\mathbf{v},b}\left[a^2(\mathbf{v},b)\mathbb{1}\{\mathbf{v}^\top\mathbf{x}+b\geq 0\}-(\boldsymbol{\beta}^\top\mathbf{x})^{2k}\right]$$

$$\leq \mathbb{E}_{\mathbf{x},\mathbf{v},b}\left[c_k^2 H_k^2(\boldsymbol{\beta}^\top\mathbf{v}/\|\boldsymbol{\beta}\|_2)+(\boldsymbol{\beta}^\top\mathbf{x})^{2k}\right]$$

$$= c_k^2\sqrt{2\pi}k! + \mathbb{E}_{\mathbf{x}}\left[\boldsymbol{\beta}^\top\mathbf{x}\right]^{2k}$$

$$\leq \frac{200^2 k^4}{(k-1)!!(k-1)!!}k!\|\boldsymbol{\beta}\|_2^{2k}+\|\boldsymbol{\beta}\|_2^{2k}$$

$$\leq 2\times 200^2 k^5\|\boldsymbol{\beta}\|_2^{2k}.$$

The last inequality invokes the identity $\frac{k!}{((k-1)!!)^2}+1\leq \frac{k!}{(k-1)!}+1\leq 2\frac{k!}{(k-1)!}=2k$. Therefore, choosing $D=2\times 200^2 k^5\|\boldsymbol{\beta}\|_2^{2k}/(\epsilon^2\delta)$ gives rise to

$$\mathbb{P}\left(\left\|\frac{1}{D}\sum_{j=1}^D a(\mathbf{v}_j,b_j)\mathbb{1}\{\mathbf{v}_j^\top\mathbf{x}+b_j\geq 0\}-(\boldsymbol{\beta}^\top\mathbf{x})^k\right\|_{L_2}\geq\epsilon\right)$$

$$\leq \epsilon^{-2}\mathbb{E}_{\mathbf{x},b}\left[\left\|\frac{1}{D}\sum_{j=1}^D a(\mathbf{v}_j,b_j)\mathbb{1}\{\mathbf{v}_j^\top\mathbf{x}+b_j\geq 0\}-(\boldsymbol{\beta}^\top\mathbf{x})^k\right\|_{L_2}^2\right]$$

$$\leq \delta.$$

This completes the proof. $\qquad\square$

**From A Single Polynomial to A Sum of Polynomials**.

**Lemma 7.** Given a function $f(\mathbf{x})=\sum_{s=1}^{r_\star}(\boldsymbol{\beta}_s^\top\mathbf{x})^{k_s}$ defined on $\mathbf{x}\in\mathbb{S}^{d-1}$, and positive constants $\epsilon>0$ and $\delta>0$, we choose $D\geq\frac{2\times 200^2 r_\star^3\sum_{s=1}^{r_\star}k_s^5\|\boldsymbol{\beta}_s\|_2^{2k_s}}{\epsilon^2\delta}$, Then there exists scalar $a(\mathbf{v}_j,b_j)$ for $j=1,\ldots,D$, such that with probability at least $1-\delta$ over independently randomly sampled $\mathbf{v}_j\sim\mathsf{N}(\mathbf{0},\mathbf{I}_d)$ and $b_j\sim\mathsf{N}(0,1)$ for $j=1,\ldots,D$, we have

$$\left\|\frac{1}{D}\sum_{j=1}^D a(\mathbf{v}_j,b_j)\mathbb{1}\{\mathbf{v}_j^\top\mathbf{x}+b_j\geq 0\}-f(\mathbf{x})\right\|_{L_2}\leq\epsilon.$$

*Proof.* We apply Lemma 5 and Lemma 6 repeatedly for $r_\star$ times. Specifically, for each fixed $s\leq r_\star$, Lemma 5 implies that there exists $a_s(\mathbf{v},b)$ such that $\mathbb{E}_{\mathbf{v},b}[a_s(\mathbf{v},b)\mathbb{1}\{\mathbf{v}^\top\mathbf{x}+b\geq 0\}]=(\boldsymbol{\beta}_s^\top\mathbf{x})^{k_s}$. Then we choose $D_s\geq 2\times 200^2 k_s^5\|\boldsymbol{\beta}_s\|_2^{2k_s}r_\star^3/(\epsilon^2\delta)$ so that with probability at least $1-\delta/r_\star$, the following $L_2$ bound holds

$$\left\|\frac{1}{D_s}\sum_{j=1}^{D_s} a_s(\mathbf{v}_j,b_j)\mathbb{1}\{\mathbf{v}_j^\top\mathbf{x}+b_j\geq 0\}-(\boldsymbol{\beta}_s^\top\mathbf{x})^{k_s}\right\|_{L_2}\leq\epsilon/r_\star.$$

To this end, we set $D=\sum_{s=1}^{r_\star}D_s\geq\frac{2\times 200^2 r_\star^3\sum_{s=1}^{r_\star}k_s^5\|\boldsymbol{\beta}_s\|_2^{2k_s}}{\epsilon^2\delta}$ and define

$$\mathbf{g}(\mathbf{x})=[\mathbf{g}_1(\mathbf{x})^\top,\ldots,\mathbf{g}_{r_\star}(\mathbf{x})^\top]^\top$$

with the $j$-the element of $\mathbf{g}_s$ as $[\mathbf{g}_s(\mathbf{x})]_j=\mathbb{1}\{\mathbf{v}_j^\top\mathbf{x}+b_j\geq 0\}$ for $j=1,\ldots,D_s$. In other words, we construct a random feature vector $\mathbf{g}(\mathbf{x})\in\mathbb{R}^D$ by stacking all the random features for approximating the $k_s$-degree polynomial. Similar to $\mathbf{g}$, we denote $\mathbf{a}=[\mathbf{a}_1^\top,\ldots,\mathbf{a}_{r_\star}^\top]^\top$ with the $j$-th element of $\mathbf{a}_s$ as $[\mathbf{a}_s]_j=\frac{1}{D_s}a_s(\mathbf{v}_j,b_j)$ for $j=1,\ldots,D_s$. Then we can bound the $L_2$ distance between $f(\mathbf{x})$ and

$\mathbf{a}^\top \mathbf{g}(\mathbf{x})$:

$$\left\| \mathbf{a}^\top \mathbf{g}(\mathbf{x}) - f(\mathbf{x}) \right\|_{L_2} \leq \sum_{s=1}^{r_\star} \left\| \mathbf{a}_s^\top \mathbf{g}_s(\mathbf{x}) - (\boldsymbol{\beta}_s^\top \mathbf{x})^{k_s} \right\|_{L_2}$$

$$\leq \sum_{s=1}^{r_\star} \left\| \frac{1}{D_s} \sum_{j=1}^{D_s} a_s(\mathbf{v}_j, b_j) \, \mathbb{1}\{\mathbf{v}_j^\top \mathbf{x} + b_j \geq 0\} - (\boldsymbol{\beta}_s^\top \mathbf{x})^{k_s} \right\|_{L_2}$$

$$\leq \epsilon.$$

The above inequality holds with probability $1 - \delta$ by the union bound. We complete the proof. $\qquad \square$

Lemma 7 showcases how to express a sum of polynomials by stacking neural random features for approximating individual polynomials. This technique will be extensively used in the remaining proofs.

### D.2 Expressivity of `Quad-h`

We show `Quad-Neural` with neural representation $\mathbf{h}$ can approximate any function $f$ of the form

$$f(\mathbf{x}) = \sum_{s=1}^{r_\star} \alpha_s (\boldsymbol{\beta}_s^\top \mathbf{x})^{p_s}, \quad \text{where} \quad |\alpha_s| \leq 1, \; \left\| (\boldsymbol{\beta}_s^\top \mathbf{x})^{p_s} \right\|_{L_2} \leq 1, \; p_s \leq p \text{ for all } s. \tag{12}$$

To ease the presentation, we temporarily assume all the $p_s$ are even. We extend to odd-degree polynomials in 9. Recall we denote

$$\mathbf{g}(\mathbf{x}) = [\mathbf{g}_1(\mathbf{x})^\top, \ldots, \mathbf{g}_{r_\star}(\mathbf{x})^\top]^\top \quad \text{with } \mathbf{g}_s(\mathbf{x}) \text{ being a collection of random indicator functions.}$$

We whiten $\mathbf{g}(\mathbf{x})$ by the estimated covariance matrix $\widehat{\boldsymbol{\Sigma}}$ to obtain $\mathbf{h}(\mathbf{x}) = \widehat{\boldsymbol{\Sigma}}^{-1/2} \mathbf{g}(\mathbf{x})$. Note that $\mathbf{h}(\mathbf{x})$ is a $D$-dimensional vector. The approximation of `Quad-h` is stated in the following lemma.

**Lemma 8.** For a given $f$ in the form of (12) with all $p_s$ even, and for small constants $\epsilon > 0$ and $\delta > 0$, we choose $D \geq \frac{4 \times 50^2 r_\star^3 \sum_{s=1}^{r_\star} p_s^5 \|\boldsymbol{\beta}_s\|_2^{p_s}}{\epsilon^2 \delta}$, and $m \geq \frac{54 r_\star D (1 + \log \frac{8}{\delta})}{\epsilon^2} \log \frac{1}{\epsilon}$. Let $\mathbf{w}_{0,r} \overset{\text{iid}}{\sim} \mathsf{N}(\mathbf{0}, \mathbf{I}_D)$ and $a_r \overset{\text{iid}}{\sim} \text{Unif}(\{\pm 1\})$ for $r = 1, \ldots, m$, then there exist proper $\{\mathbf{w}_r^*\}$ such that with probability at least $1 - \delta$, we have

$$\left\| \frac{1}{2\sqrt{m}} \sum_{r=1}^{m} a_r \mathbb{1}\{\mathbf{w}_{0,r}^\top \mathbf{h}(\mathbf{x}) \geq 0\} \left( (\mathbf{w}_r^*)^\top \mathbf{h}(\mathbf{x}) \right)^2 - f(\mathbf{x}) \right\|_{L_2} \leq 7 r_\star \epsilon.$$

*Proof.* By definition, $f$ can be written as a sum of polynomials with leading coefficients $\alpha_s$. We partition $m$ neurons into two parts according to the sign of $a_r$. We will use the positive part to express those polynomials with positive coefficient $\alpha_s$, and negative part to express those with negative coefficients. We first show for sufficiently large $m$, the number of positive $a_r$'s exceeds $\frac{1}{3}m$ with high probability. This follows from the tail bound of i.i.d. binomial random variables. By the Hoeffding's inequality, we have

$$\mathbb{P}\left(\text{Number of positive } a_r \leq k\right) \leq \exp\left(-2m(1/2 - (k/m)^2)\right).$$

Letting $k = \frac{1}{3}m$ and setting $\mathbb{P}\left(\text{Number of positive } a_r \leq k\right) \leq \delta$, we have $m \geq 2 \log \frac{1}{\delta}$. We denote $\mathcal{I}_1 = \{1, \ldots, m/3\}$ and $\mathcal{I}_2 = \{m/3 + 1, \ldots, 2m/3\}$. Without loss of generality, we assume $a_r = 1$ for $r \in \mathcal{I}_1$.

The remaining proof is built upon Lemma 7. We choose $D = \frac{50^2 r_\star^3 \sum_{s=1}^{r_\star} p_s^5 \|\boldsymbol{\beta}_s\|_2^{p_s}}{\epsilon^2 \delta}$, so that with probability at least $1 - \delta$, there exists $\mathbf{a}$ with $\left\| \mathbf{a}^\top \mathbf{g}(\mathbf{x}) - \sum_{s=1}^{r_\star} (\boldsymbol{\beta}_s^\top \mathbf{x})^{p_s/2} \right\|_{L_2} \leq \epsilon$. We further partition $\mathcal{I}_1$ into $r_\star$ consecutive groups of equal size $m_0$, i.e., $r_\star m_0 = m/3$. Within a group, we aim to approximate $\alpha_s (\boldsymbol{\beta}_s^\top \mathbf{x})^{p_s}$ with $\alpha_s > 0$ for some fixed $s \leq r_\star$. Accordingly, we choose

$\mathbf{w}_r^{s,*} = 2\sqrt{\alpha_s}(3r_\star)^{1/4}m_0^{-1/4}\widehat{\boldsymbol{\Sigma}}^{1/2}[\mathbf{0}^\top, \ldots, \mathbf{a}_s^\top, \ldots, \mathbf{0}^\top]^\top$ for $r = 1, \ldots, m_0$. We have

$$\left\| \frac{1}{2\sqrt{m}} \sum_{r=1}^{m_0} a_{0,r}\mathbb{1}\{\mathbf{w}_{0,r}^\top\mathbf{h}(\mathbf{x}) \geq 0\} \left((\mathbf{w}_r^{s,*})^\top\mathbf{h}(\mathbf{x})\right)^2 - \alpha_s(\boldsymbol{\beta}_s^\top\mathbf{x})^{p_s} \right\|_{L_2}$$

$$= \left\| \frac{1}{2\sqrt{3r_\star m_0}} \sum_{r=1}^{m_0} 4\sqrt{3r_\star}\mathbb{1}\{\mathbf{w}_{0,r}^\top\mathbf{h}(\mathbf{x}) \geq 0\}\alpha_s m_0^{-1/2} \left(\mathbf{a}_s^\top\mathbf{g}_s\right)^2 - \alpha_s(\boldsymbol{\beta}_s^\top\mathbf{x})^{p_s} \right\|_{L_2}$$

$$= \left\| \frac{1}{m_0} \sum_{r=1}^{m_0} 2\alpha_s\mathbb{1}\{\mathbf{w}_{0,r}^\top\mathbf{h}(\mathbf{x}) \geq 0\} \left(\mathbf{a}_s^\top\mathbf{g}_s\right)^2 - \alpha_s(\boldsymbol{\beta}_s^\top\mathbf{x})^{p_s} \right\|_{L_2}. \tag{13}$$

We know $\mathbf{a}_s^\top\mathbf{g}_s$ well approximates $(\boldsymbol{\beta}_s^\top\mathbf{x})^{p_s/2}$. If $\sup_{\mathbf{x}\in\mathbb{S}^{d-1}} \frac{1}{m_0}\sum_{r=1}^{m_0} 2\mathbb{1}\{\mathbf{w}_{0,r}^\top\mathbf{h}(\mathbf{x}) \geq 0\}$ concentrates around 1, then the above $L_2$ norm can be bounded by $O(\epsilon)$. We substantiate this reasoning by the following claim:

**Claim 1.** With probability at least $1 - 2\delta$, we have

$$\sup_{\mathbf{x}\in\mathbb{S}^{d-1}} \left| \frac{1}{m_0} \sum_{r=1}^{m_0} 2\mathbb{1}\{\mathbf{w}_{0,r}^\top\mathbf{h}(\mathbf{x}) \geq 0\} - 1 \right| \leq 6\sqrt{\frac{D\log(3m_0)\left(1 + \log\frac{2}{\delta}\right)}{m_0}}.$$

The proof of the claim is deferred to Appendix D.4. Based on the claim, we are ready to finish proving (13). By the triangle inequality, we deduce

$$\left\| \frac{1}{m_0} \sum_{r=1}^{m_0} 2\alpha_s\mathbb{1}\{\mathbf{w}_{0,r}^\top\mathbf{h}(\mathbf{x}) \geq 0\} \left(\mathbf{a}_s^\top\mathbf{g}_s\right)^2 - \alpha_s(\boldsymbol{\beta}_s^\top\mathbf{x})^{p_s} \right\|_{L_2}$$

$$= \left\| \frac{1}{m_0} \sum_{r=1}^{m_0} 2\alpha_s\mathbb{1}\{\mathbf{w}_{0,r}^\top\mathbf{h}(\mathbf{x}) \geq 0\} \left(\mathbf{a}_s^\top\mathbf{g}_s\right)^2 - \alpha_s \left(\mathbf{a}_s^\top\mathbf{g}_s\right)^2 + \alpha_s \left(\mathbf{a}_s^\top\mathbf{g}_s\right)^2 - \alpha_s(\boldsymbol{\beta}_s^\top\mathbf{x})^{p_s} \right\|_{L_2}$$

$$\leq \left\| \frac{1}{m_0} \sum_{r=1}^{m_0} 2\alpha_s\mathbb{1}\{\mathbf{w}_{0,r}^\top\mathbf{h}(\mathbf{x}) \geq 0\} \left(\mathbf{a}_s^\top\mathbf{g}_s\right)^2 - \alpha_s \left(\mathbf{a}_s^\top\mathbf{g}_s\right)^2 \right\|_{L_2} + \left\| \alpha_s \left(\mathbf{a}_s^\top\mathbf{g}_s\right)^2 - \alpha_s(\boldsymbol{\beta}_s^\top\mathbf{x})^{p_s} \right\|_{L_2}$$

$$\leq \alpha_s \left\| \left(\mathbf{a}_s^\top\mathbf{g}_s\right)^2 \right\|_{L_2} \left\| \frac{1}{m_0} \sum_{r=1}^{m_0} 2\mathbb{1}\{\mathbf{w}_{0,r}^\top\mathbf{h}(\mathbf{x}) \geq 0\} - 1 \right\|_{L_2}$$

$$\quad + \alpha_s \left\| \mathbf{a}_s^\top\mathbf{g}_s + (\boldsymbol{\beta}_s^\top\mathbf{x})^{p_s/2} \right\|_{L_2} \left\| \mathbf{a}_s^\top\mathbf{g}_s - (\boldsymbol{\beta}_s^\top\mathbf{x})^{p_s/2} \right\|_{L_2}$$

$$\leq 6\alpha_s \left( \left\| (\boldsymbol{\beta}_s^\top\mathbf{x})^{p_s/2} \right\|_{L_2} + \epsilon \right) \sqrt{\frac{D\log(3m_0)\left(1 + \log\frac{2}{\delta}\right)}{m_0}} + \alpha_s\epsilon \left( 2 \left\| (\boldsymbol{\beta}_s^\top\mathbf{x})^{p_s/2} \right\|_{L_2} + \epsilon \right).$$

The above upper bound holds with probability no smaller than $1 - 3\delta$. Taking

$$m_0 = \frac{18D\left(1 + \log\frac{2}{\delta}\right)}{\epsilon^2} \log\frac{1}{\epsilon},$$

for a small $\epsilon < \left\| (\boldsymbol{\beta}_s^\top\mathbf{x})^{p_s/2} \right\|_{L_2}$, with probability at least $1 - 3\delta$, the following

$$\left\| \frac{1}{m_0} \sum_{r=1}^{m_0} a_r\mathbb{1}\{\mathbf{w}_{0,r}^\top\mathbf{h}(\mathbf{x}) \geq 0\} \left((\mathbf{w}_r^{s,*})^\top\mathbf{h}(\mathbf{x})\right)^2 - \alpha_s(\boldsymbol{\beta}_s^\top\mathbf{x})^{p_s} \right\|_{L_2} \leq 7\alpha_s\epsilon \left\| (\boldsymbol{\beta}_s^\top\mathbf{x})^{p_s/2} \right\|_{L_2}$$

holds true for the $s$-th group with $\alpha_s > 0$. When $\alpha_s < 0$, we simply set $\mathbf{w}_r^{s,\star} = \mathbf{0}$. As a result, in $\mathcal{I}_1$, we can express all the polynomial with a positive coefficient.

To express polynomials with negative coefficients, we use $\mathcal{I}_2$ analogously. By evenly partitioning $\mathcal{I}_2$ into $r_\star$ consecutive groups, for a fixed $s \leq r_\star$ and $\alpha_s < 0$, we choose $\mathbf{w}_r^{s,*} = 2\sqrt{|\alpha_s|}(3r_\star)^{1/4}m_0^{-1/4}\widehat{\boldsymbol{\Sigma}}^{1/2}[\mathbf{0}^\top, \ldots, \mathbf{a}_s^\top, \ldots, \mathbf{0}^\top]^\top$. Using exactly the same argument in $\mathcal{I}_1$, with probability at least $1 - 3\delta$, for $\alpha_s < 0$, we also have

$$\left\| \frac{1}{m_0} \sum_{r=1}^{m_0} a_r\mathbb{1}\{\mathbf{w}_{0,r}^\top\mathbf{h}(\mathbf{x}) \geq 0\} \left((\mathbf{w}_r^{s,*})^\top\mathbf{h}(\mathbf{x})\right)^2 - \alpha_s(\boldsymbol{\beta}_s^\top\mathbf{x})^{p_s} \right\|_{L_2} \leq 7|\alpha_s|\epsilon \left\| (\boldsymbol{\beta}_s^\top\mathbf{x})^{p_s/2} \right\|_{L_2}.$$

The last step for proving Lemma 8 is to combine $\mathcal{I}_1$ and $\mathcal{I}_2$ together and choose the remaining weight parameters $\mathbf{w}_r^*$ identically $\mathbf{0}$ for $r \geq 2m/3 + 1$. Substituting into the $\mathtt{Quad\text{-}h}$ model, with probability at least $1 - 4\delta$, we deduce

$$
\left\| \frac{1}{2\sqrt{m}} \sum_{r=1}^{m} a_r \mathbb{1}\{\mathbf{w}_{0,r}^\top \mathbf{h}(\mathbf{x}) \geq 0\} \left((\mathbf{w}_r^*)^\top \mathbf{h}(\mathbf{x})\right)^2 - f(\mathbf{x}) \right\|_{L_2}
$$

$$
= \left\| \frac{1}{2\sqrt{m}} \sum_{r \in \mathcal{I}_1 \bigcup \mathcal{I}_2} a_r \mathbb{1}\{\mathbf{w}_{0,r}^\top \mathbf{h}(\mathbf{x}) \geq 0\} \left((\mathbf{w}_r^*)^\top \mathbf{h}(\mathbf{x})\right)^2 - f(\mathbf{x}) \right\|_{L_2}
$$

$$
\leq \sum_{s=1}^{r_\star} \left\| \frac{1}{2\sqrt{m}} \sum_{r \in \mathcal{I}_1 \bigcup \mathcal{I}_2} a_r \mathbb{1}\{\mathbf{w}_{0,r}^\top \mathbf{h}(\mathbf{x}) \geq 0\} \left((\mathbf{w}_r^{s,*})^\top \mathbf{h}(\mathbf{x})\right)^2 - \alpha_s (\boldsymbol{\beta}_s^\top \mathbf{x})^{p_s} \right\|_{L_2}
$$

$$
\leq \sum_{s=1}^{r_\star} \left\| \frac{1}{m_0} \sum_{r=1}^{m_0} 2\alpha_s \mathbb{1}\{\mathbf{w}_{0,r}^\top \mathbf{h}(\mathbf{x}) \geq 0\} \left(\mathbf{a}_s^\top \mathbf{g}_s(\mathbf{x})\right)^2 - \alpha_s (\boldsymbol{\beta}_s^\top \mathbf{x})^{p_s} \right\|_{L_2}
$$

$$
\leq 7\epsilon \sum_{s=1}^{r_\star} |\alpha_s| \left\| (\boldsymbol{\beta}_s^\top \mathbf{x})^{p_s/2} \right\|_{L_2}
$$

$$
\leq 7\epsilon \sqrt{\sum_{s=1}^{r_\star} \alpha_s^2 \sum_{s=1}^{r_\star} \|(\boldsymbol{\beta}_s^\top \mathbf{x})^{p_s}\|_{L_2}}
$$

$$
\leq 7 r_\star \epsilon.
$$

The width $m$ satisfies $m = 3 r_\star m_0 \geq \frac{54 r_\star D (1 + \log \frac{2}{\delta})}{\epsilon^2} \log \frac{1}{\epsilon}$. Replacing $\delta = \delta/4$ completes the proof. $\qquad\square$

**Expressivity with Odd-Degree Polynomials**.

$\mathtt{Quad\text{-}h}$ model can also efficiently express odd-degree polynomials. We rely on the following decomposition trick. Let $k$ be an integer. We rewrite a $(2k+1)$-degree polynomial as

$$
(\boldsymbol{\beta}^\top \mathbf{x})^{2k+1} = \left( \frac{(\boldsymbol{\beta}^\top \mathbf{x})^{k+1} + (\boldsymbol{\beta}^\top \mathbf{x})^k}{2} \right)^2 - \left( \frac{(\boldsymbol{\beta}^\top \mathbf{x})^{k+1} - (\boldsymbol{\beta}^\top \mathbf{x})^k}{2} \right)^2.
$$

Since QuadNTK can naturally implement the quadratic function, we only require that the neural representation $\mathbf{h}(\mathbf{x})$ can approximate $(\boldsymbol{\beta}^\top \mathbf{x})^{k+1} \pm (\boldsymbol{\beta}^\top \mathbf{x})^k$. This is true since random indicator functions can approximate $(\boldsymbol{\beta}^\top \mathbf{x})^{k+1}$ and $(\boldsymbol{\beta}^\top \mathbf{x})^k$ due to Lemma 5. We denote $\mathbf{a}_1^\top \mathbf{g}_1(\mathbf{x}) \approx (\boldsymbol{\beta}^\top \mathbf{x})^{k+1}$ in $L_2$, and $\mathbf{a}_2^\top \mathbf{g}_2(\mathbf{x}) \approx (\boldsymbol{\beta}^\top \mathbf{x})^k$ in $L_2$. Then by stacking $\mathbf{g}_1$ and $\mathbf{g}_2$, we have $[\mathbf{a}_1^\top, \pm \mathbf{a}_2^\top][\mathbf{g}_1^\top, \mathbf{g}_2^\top]^\top \approx (\boldsymbol{\beta}^\top \mathbf{x})^{k+1} \pm (\boldsymbol{\beta}^\top \mathbf{x})^k$ in $L_2$. Therefore, we only need to augment the dimension $D$ of the neural representation to approximate odd-degree polynomials. We concretize this argument in the following lemma.

**Lemma 9.** For a given $f$ in the form of (12), and small constants $\epsilon > 0$ and $\delta > 0$, we choose $D \geq \frac{8 \times 50^2 r_\star^3 \sum_{s=1}^{r_\star} p_s^5 \|\boldsymbol{\beta}_s\|_2^{2\lceil p_s/2 \rceil}}{\epsilon^2 \delta}$, and $m \geq \frac{54 r_\star D (1 + \log \frac{8}{\delta})}{\epsilon^2} \log \frac{1}{\epsilon}$. Let $\mathbf{w}_{0,r} \overset{\text{iid}}{\sim} \mathsf{N}(\mathbf{0}, \mathbf{I}_D)$ and $a_r \overset{\text{iid}}{\sim} \mathrm{Unif}(\{\pm 1\})$ for $r = 1, \ldots, m$, then there exist proper $\{\mathbf{w}_r^*\}$ such that with probability at least $1 - \delta$, we have

$$
\left\| \frac{1}{2\sqrt{m}} \sum_{r=1}^{m} a_r \mathbb{1}\{\mathbf{w}_{0,r}^\top \mathbf{h}(\mathbf{x}) \geq 0\} \left((\mathbf{w}_r^*)^\top \mathbf{h}(\mathbf{x})\right)^2 - f(\mathbf{x}) \right\|_{L_2} \leq 7 r_\star \epsilon.
$$

*Proof.* We can write

$$f(\mathbf{x}) = \sum_{s=1}^{r_\star} \alpha_s (\boldsymbol{\beta}_s^\top \mathbf{x})^{p_s} \mathbb{1}\{p_s \text{ is even}\} + \alpha_s (\boldsymbol{\beta}_s^\top \mathbf{x})^{p_s} \mathbb{1}\{p_s \text{ is odd}\}$$

$$= \sum_{s=1}^{r_\star} \alpha_s \left( (\boldsymbol{\beta}_s^\top \mathbf{x})^{p_s/2} \right)^2 \mathbb{1}\{p_s \text{ is even}\}$$

$$+ \alpha_s \left[ \left( \frac{(\boldsymbol{\beta}_s^\top \mathbf{x})^{\frac{p_s+1}{2}} + (\boldsymbol{\beta}_s^\top \mathbf{x})^{\frac{p_s-1}{2}}}{2} \right)^2 - \left( \frac{(\boldsymbol{\beta}_s^\top \mathbf{x})^{\frac{p_s+1}{2}} - (\boldsymbol{\beta}_s^\top \mathbf{x})^{\frac{p_s-1}{2}}}{2} \right)^2 \right] \mathbb{1}\{p_s \text{ is odd}\}.$$

Applying Lemma 5 once, there exists $\mathbf{a}_s$ such that $\left\| \mathbf{a}_s^\top - (\boldsymbol{\beta}_s^\top \mathbf{x})^{p_s/2} \right\|_{L_2} \le \epsilon/r_\star$, when $p_s$ is even and the corresponding $D_s \ge \frac{4 \times 50^2 r_\star^3 \sum_{s=1}^{r_\star} p_s^5 \|\boldsymbol{\beta}_s\|_2^{2p_s}}{\epsilon^2 \delta}$.

For an odd $p_s$, we apply the technique in Lemma 6. There exist $\mathbf{a}_{s,+}$ and $\mathbf{a}_{s,-}$ with corresponding random indicator features $\mathbf{g}_{s,+}(\mathbf{x})$ and $\mathbf{g}_{s,-}(\mathbf{x})$ such that

$$\left\| [\mathbf{a}_{s,+}^\top, \pm\mathbf{a}_{s,-}^\top][\mathbf{g}_{s,+}^\top(\mathbf{x}), \mathbf{g}_{s,-}^\top(\mathbf{x})]^\top - \left( (\boldsymbol{\beta}_s^\top \mathbf{x})^{\frac{p_s+1}{2}} \pm (\boldsymbol{\beta}_s^\top \mathbf{x})^{\frac{p_s-1}{2}} \right) \right\|_{L_2}$$

$$\le \left\| \mathbf{a}_{s,+}^\top \mathbf{g}_{s,+}(\mathbf{x}) - (\boldsymbol{\beta}_s^\top \mathbf{x})^{p_s/2} \right\|_{L_2} + \left\| \mathbf{a}_{s,-}^\top \mathbf{g}_{s,-}(\mathbf{x}) - (\boldsymbol{\beta}_s^\top \mathbf{x})^{\frac{p_s-1}{2}} \right\|_{L_2}$$

$$\le \epsilon/r_\star.$$

The corresponding neural representation dimension is $D_s \ge 50^2 r_\star^3 \frac{(p_s+1)^5 \|\boldsymbol{\beta}_s\|_2^{p_s+1} + (p_s-1)^5 \|\boldsymbol{\beta}_s\|_2^{p_s-1}}{\epsilon^2 \delta}$. Combining the even and odd degrees together, we can choose

$$D \ge \frac{4 \times 50^2 r_\star^3}{\epsilon^2 \delta} \left( \sum_{s=1}^{r_\star} p_s^5 \|\boldsymbol{\beta}_s\|_2^{p_s} \mathbb{1}\{p_s \text{ is even}\} + 2(p_s+1)^5 \|\boldsymbol{\beta}_s\|_2^{p_s+1} \mathbb{1}\{p_s \text{ is odd}\} \right).$$

Lemma 9 now follows from Lemma 8 by merging $\mathbf{g}_{s,+}, \mathbf{g}_{s,-}$ as a single feature $\mathbf{g}_s$, and $\mathbf{a}_{s,+}, \mathbf{a}_{s,-}$ as a single weight vector $\mathbf{a}_s$ so that $\mathbf{w}_r^{s,*}$ can be chosen accordingly. Unifying the notation for even and odd degree polynomials, we have

$$D \ge \frac{8 \times 50^2 r_\star^3}{\epsilon^2 \delta} \left( \sum_{s=1}^{r_\star} (p_s+1)^5 \|\boldsymbol{\beta}_s\|_2^{2\lceil p_s/2 \rceil} \right).$$

$\square$

## D.3  Generalization of `Quad-h`

Lemma 8 and 9 construct a proper weight matrix $\mathbf{W}^* = [\mathbf{w}_1, \ldots, \mathbf{w}_m]^\top$ such that `Quad-Neural` can well approximates $f_\star$ of form (12) in the $L_2$ sense. Now we show that for sufficiently large $m$, the empirical risk $\widehat{\mathcal{R}}_\ell(f_{\mathbf{W}^*})$ is comparable to that of $f_\star$. By the Lipschitz property of the loss function, we derive

$$\frac{1}{n} \sum_{i=1}^{n} \ell(f_{\mathbf{W}^*}(\mathbf{x}_i), y_i) \le \frac{1}{n} \sum_{i=1}^{n} \ell(f_{\mathbf{W}^*}(\mathbf{x}_i), y_i) - \ell(f_\star(\mathbf{x}_i), y_i) + \ell(f_0(\mathbf{x}_i), y_i)$$

$$\le \frac{1}{n} \sum_{i=1}^{n} |f_{\mathbf{W}^*}(\mathbf{x}_i) - f_\star(\mathbf{x}_i)| + \widehat{\mathcal{R}}(f_\star).$$

For a given $\epsilon_0 > 0$, using Chebyshev's inequality, we have

$$\mathbb{P}\left(|\widehat{\mathcal{R}}(f_\star) - \mathcal{R}(f_\star)| \geq \epsilon_0/2\right) \leq \frac{4\mathbb{E}[(\widehat{\mathcal{R}}(f_\star) - \mathcal{R}(f_\star))^2]}{\epsilon_0^2}$$

$$= \frac{4\mathbb{E}_{(\mathbf{x},y)\sim\mathcal{D}}[\ell(f_\star(\mathbf{x}), y) - \mathbb{E}_{(\mathbf{x},y)\sim\mathcal{D}}[\ell(f_\star(\mathbf{x}), y)]]^2}{n_2\epsilon_0^2}$$

$$\leq \frac{4\mathbb{E}_{(\mathbf{x},y)\sim\mathcal{D}}[\ell(0, y) + |f_\star(\mathbf{x})| - \mathsf{OPT}]^2}{n\epsilon_0^2}$$

$$\leq \frac{8\mathbb{E}_{(\mathbf{x},y)\sim\mathcal{D}}[|f_\star(\mathbf{x})|]^2 + 8(1 + \mathsf{OPT})^2}{n\epsilon_0^2}$$

$$\leq \frac{8(1 + \mathsf{OPT})^2 + 8\mathbb{E}_{\mathbf{x}}\left[\sum_{s=1}^{r_\star}\alpha_s^2 \sum_{s=1}^{r_\star}(\boldsymbol{\beta}_s^\top\mathbf{x})^{2p_s}\right]}{n\epsilon_0^2}$$

$$\leq \frac{8(1 + \mathsf{OPT})^2 + 8r_\star^2}{n\epsilon_0^2}.$$

Choosing $n \geq \frac{8(1+\mathsf{OPT})^2+8r_\star^2}{\delta\epsilon_0^2}$, $\widehat{\mathcal{R}}(f_\star) - \mathcal{R}(f_\star) \leq \epsilon_0/2$ holds with probability at least $1 - \delta$. We further invoke Lemma 8 and Chebyshev's inequality again on $\frac{1}{n}\sum_{i=1}^{n}|f_{\mathbf{W}^*}(\mathbf{x}_i) - f_\star(\mathbf{x}_i)|$:

$$\mathbb{P}_{\mathbf{x}}\left(\frac{1}{n}\sum_{i=1}^{n}|f_{\mathbf{W}^*}(\mathbf{x}_i) - f_\star(\mathbf{x}_i)| \geq \epsilon_0/2\right) \leq \frac{4\mathbb{E}_{\mathbf{x}}\left[\frac{1}{n^2}\left(\sum_{i=1}^{n}|f_{\mathbf{W}^*}(\mathbf{x}_i) - f_\star(\mathbf{x}_i)|\right)^2\right]}{\epsilon_0^2}$$

$$\leq \frac{\frac{4}{n}\sum_{i=1}^{n}\mathbb{E}_{\mathbf{x}}\left[|f_{\mathbf{W}^*}(\mathbf{x}_i) - f_\star(\mathbf{x}_i)|^2\right]}{\epsilon_0^2}$$

$$\leq \frac{196\epsilon^2 r_\star^2}{\epsilon_0^2},$$

where the last inequality holds with probability at least $1 - \delta$. We set $\frac{196r_\star^2\epsilon^2}{\epsilon_0^2} \leq \delta$, which implies $\epsilon^2 \leq \frac{\delta\epsilon_0^2}{196r_\star^2}$. Accordingly, the number of neurons in the top layer needs to be at least

$$m \geq \frac{10584r_\star^3 D\left(1 + \log\frac{8}{\delta}\right)\sum_{s=1}^{r_\star}\left\|(\boldsymbol{\beta}_s^\top\mathbf{x})^{p_s}\right\|_{L_2}}{\delta\epsilon_0^2}\log\frac{7r_\star}{\sqrt{\delta}\epsilon_0},$$

and the dimension of the neural representation is

$$D = \frac{50^2 \times 392r_\star^5}{\delta\epsilon_0^2}\left(\sum_{s=1}^{r_\star}(p_s + 1)^5\|\boldsymbol{\beta}_s\|_2^{2\lceil p_s/2\rceil}\right). \tag{14}$$

This gives us that with probability at least $1 - 3\delta$ over the randomness of data and initialization[7], the empirical risk satisfies

$$\frac{1}{n}\sum_{i=1}^{n}\ell(f_{\mathbf{W}^*}(\mathbf{x}_i), y_i) \leq \mathsf{OPT} + \epsilon_0.$$

Applying Theorem 1 part (2), for any second-order stationary point $\widehat{\mathbf{W}}$ and proper regularization parameter $\lambda$, we have

$$\widehat{\mathcal{R}}_\lambda(f_{\widehat{\mathbf{W}}}^Q) \leq (1 + \tau)(\mathsf{OPT} + \epsilon_0) + \epsilon_0 \leq (1 + \tau_0)\mathsf{OPT} + \epsilon_0.$$

**Bounding $B_{w,\star}$.**

Towards establishing the generalization bound of $f_{\widehat{\mathbf{W}}}^Q$, we first find $B_{w,\star}$:

$$\sum_{r=1}^{m} \|\mathbf{w}_r^*\|_2^4 = \sum_{s=1}^{r_\star} \sum_{r\in\mathcal{I}_1\bigcup\mathcal{I}_2} \|\mathbf{w}_r^{s,*}\|_2^4 = 48r_\star \sum_{s=1}^{r_\star}\sum_{r=1}^{m_0} \alpha_s^2 m_0^{-1} \left\|\widehat{\mathbf{\Sigma}}^{1/2}[\mathbf{0}^\top,\ldots,\mathbf{a}_s^\top,\ldots,\mathbf{0}^\top]^\top\right\|_2^4$$

$$= 48r_\star \sum_{s=1}^{r_\star} \alpha_s^2 \left\|\widehat{\mathbf{\Sigma}}^{1/2}[\mathbf{0}^\top,\ldots,\mathbf{a}_s^\top,\ldots,\mathbf{0}^\top]^\top\right\|_2^4.$$

To bound $\left\|\widehat{\mathbf{\Sigma}}^{1/2}[\mathbf{0}^\top,\ldots,\mathbf{a}_s^\top,\ldots,\mathbf{0}^\top]^\top\right\|_2$, we first replace $\widehat{\mathbf{\Sigma}}$ with $\mathbf{\Sigma}$. We denote $\boldsymbol{\theta}_s = \mathbf{\Sigma}^{1/2}[\mathbf{0}^\top,\ldots,\mathbf{a}_s^\top,\ldots,\mathbf{0}^\top]^\top$, and observe $\boldsymbol{\theta}_s$ is the optimal solution to the following least square problem

$$\boldsymbol{\theta}_s = \underset{\mathbf{u}_1}{\operatorname{argmin}} \left\|F(\mathbf{x}) - \mathbf{u}_1^\top\mathbf{\Sigma}^{-1/2}\mathbf{g}(\mathbf{x})\right\|_{L_2}^2 \quad \text{with} \quad F(\mathbf{x}) = \mathbf{a}_s^\top\mathbf{g}_s^\top(\mathbf{x}).$$

The optimal solution is $\boldsymbol{\theta}_s = \mathbf{u}_1^* = \mathbf{\Sigma}^{-1/2}\mathbb{E}_{\mathbf{x}}[F(\mathbf{x})\mathbf{g}(\mathbf{x})]$. Similarly, the optimal solution to the following least square problem

$$\min_{\mathbf{u}_2} \left\|F(\mathbf{x}) - \mathbf{u}_2^\top\mathbf{g}(\mathbf{x})\right\|_{L_2}^2 \quad \text{with} \quad F(\mathbf{x}) = \mathbf{a}_s^\top\mathbf{g}_s(\mathbf{x})$$

is $\mathbf{u}_2^* = \mathbf{\Sigma}^{-1}\mathbb{E}_{\mathbf{x}}[F(\mathbf{x})g(\mathbf{x})]$. The residual of $\mathbf{u}_2^*$ is

$$\left\|F(\mathbf{x}) - (\mathbf{u}_2^*)^\top\mathbf{g}(\mathbf{x})\right\|_{L_2}^2 = \|F(\mathbf{x})\|_{L_2}^2 - \mathbb{E}_{\mathbf{x}}[F(\mathbf{x})\mathbf{g}(\mathbf{x})^\top]\mathbf{\Sigma}^{-1}\mathbb{E}_{\mathbf{x}}[F(\mathbf{x})\mathbf{g}(\mathbf{x})] \geq 0.$$

This implies

$$\|\boldsymbol{\theta}_s\|_2 = \sqrt{\mathbb{E}_{\mathbf{x}}[F(\mathbf{x})\mathbf{g}(\mathbf{x})^\top]\mathbf{\Sigma}^{-1}\mathbb{E}_{\mathbf{x}}[F(\mathbf{x})\mathbf{g}(\mathbf{x})]} \leq \|F(\mathbf{x})\|_{L_2} \leq \left\|(\boldsymbol{\beta}_s^\top\mathbf{x})^{p_s/2}\right\|_{L_2} + \epsilon,$$

where the last inequality follows from Lemma 5. This gives rise to

$$\sum_{s=1}^{r_\star} \alpha_s^2 \|\boldsymbol{\theta}_s\|_2^4 \leq \sum_{s=1}^{r_\star} \alpha_s^2 \left\|(\boldsymbol{\beta}^\top\mathbf{x})^{p/2}\right\|_{L_2}^4 \leq r_\star.$$

To switch back to $\widehat{\mathbf{\Sigma}}$, we invoke Lemma 3 on the concentration of $\widehat{\mathbf{\Sigma}}$ to $\mathbf{\Sigma}$. Specifically, with probability at least $1-\delta$, choosing $n_0 \geq 4c\delta^{-2}\lambda_{\lceil p/2\rceil}^{-1}D\log D$ for some constant $c$, we have

$$\frac{1}{2}\mathbf{\Sigma} \preceq \widehat{\mathbf{\Sigma}} \preceq \frac{3}{2}\mathbf{\Sigma}.$$

Consequently, by denoting $\widehat{\boldsymbol{\theta}}_s = \widehat{\mathbf{\Sigma}}^{1/2}[\mathbf{0}^\top,\ldots,\mathbf{a}_s^\top,\ldots,\mathbf{0}^\top]^\top$, we have

$$\left\|\widehat{\boldsymbol{\theta}}_s\right\|_2^2 = \left\|\widehat{\mathbf{\Sigma}}^{1/2}\mathbf{\Sigma}^{-1/2}\boldsymbol{\theta}_s\right\|_2^2 = \boldsymbol{\theta}_s^\top\mathbf{\Sigma}^{-1/2}\widehat{\mathbf{\Sigma}}\mathbf{\Sigma}^{-1/2}\boldsymbol{\theta}_s^\top \leq \left\|\mathbf{\Sigma}^{-1/2}\widehat{\mathbf{\Sigma}}\mathbf{\Sigma}^{-1/2}\right\|_{\mathrm{op}}\|\boldsymbol{\theta}_s\|_2^2 \leq \frac{3}{2}\|\boldsymbol{\theta}_s\|_2^2.$$

Plugging into $\sum_{r=1}^{m}\|\mathbf{w}_r^*\|_2^4$, we have

$$\sum_{r=1}^{m}\|\mathbf{w}_r^*\|_2^4 \leq 108r_\star\sum_{s=1}^{r_\star}\alpha_s^2\|\boldsymbol{\theta}_s\|_2^4 \leq 108r_\star^2.$$

Therefore, we can set $B_{w,\star}^4 = 108r_\star^2$. Note that $B_{w,\star}$ is independent of the width $m$.

**Bounding $M_{h,\mathbf{op}}$ and $B_h$.**

The remaining ingredients are $M_{h,\mathrm{op}}$ and $\|\mathbf{h}(\mathbf{x})\|_2$. Conditioned on the event $\frac{1}{2}\mathbf{\Sigma} \leq \widehat{\mathbf{\Sigma}} \leq \frac{3}{2}\mathbf{\Sigma}$, we know $\widehat{\mathbf{\Sigma}}^{-1} \leq 2\mathbf{\Sigma}^{-1}$. Therefore, we have

$$\|\mathbf{h}(\mathbf{x})\|_2^2 = \mathbf{g}(\mathbf{x})^\top\widehat{\mathbf{\Sigma}}^{-1}\mathbf{g}(\mathbf{x}) \leq 2\mathbf{g}(\mathbf{x})^\top\mathbf{\Sigma}^{-1}\mathbf{g}(\mathbf{x}) \leq 2\lambda_{\lceil p/2\rceil}^{-1}D.$$

Note that he norm of $\mathbf{h}(\mathbf{x})$ is in the order of $\sqrt{D}$ according to Assumption 2.

Lastly, we bound $M_{h,\mathrm{op}}$ as

$$
\begin{aligned}
B_h^2 M_{h,\mathrm{op}}^2 &= \mathbb{E}_{\mathbf{x}} \left[ \left\| \frac{1}{n_2} \sum_{i=n_1+1}^{n} \mathbf{h}(\mathbf{x}_i)\mathbf{h}(\mathbf{x}_i)^\top \right\|_{\mathrm{op}} \right] \\
&= \mathbb{E}_{\mathbf{x}} \left[ \left\| \widehat{\boldsymbol{\Sigma}}^{-1/2} \left( \frac{1}{n_2} \sum_{i=n_1+1}^{n} \mathbf{g}(\mathbf{x}_i)\mathbf{g}(\mathbf{x}_i)^\top \right) \widehat{\boldsymbol{\Sigma}}^{-1/2} \right\|_{\mathrm{op}} \right] \\
&\leq \mathbb{E}_{\mathbf{x}} \left[ \left\| \widehat{\boldsymbol{\Sigma}}^{-1/2} \boldsymbol{\Sigma}^{1/2} \boldsymbol{\Sigma}^{-1/2} \left( \frac{1}{n_2} \sum_{i=n_1+1}^{n} \mathbf{g}(\mathbf{x}_i)\mathbf{g}(\mathbf{x}_i)^\top \right) \boldsymbol{\Sigma}^{-1/2} \boldsymbol{\Sigma}^{1/2} \widehat{\boldsymbol{\Sigma}}^{-1/2} \right\|_{\mathrm{op}} \right] \\
&\leq \left\| \widehat{\boldsymbol{\Sigma}}^{-1/2} \boldsymbol{\Sigma}^{1/2} \right\|_{\mathrm{op}} \mathbb{E}_{\mathbf{x}} \left[ \left\| \boldsymbol{\Sigma}^{-1/2} \left( \frac{1}{n_2} \sum_{i=n_1+1}^{n} \mathbf{g}(\mathbf{x}_i)\mathbf{g}(\mathbf{x}_i)^\top \right) \boldsymbol{\Sigma}^{-1/2} \right\|_{\mathrm{op}} \right] \left\| \boldsymbol{\Sigma}^{1/2} \widehat{\boldsymbol{\Sigma}}^{-1/2} \right\|_{\mathrm{op}} \\
&\leq \frac{3}{2} \left\| \widehat{\boldsymbol{\Sigma}}^{-1/2} \boldsymbol{\Sigma}^{1/2} \right\|_{\mathrm{op}}^2 .
\end{aligned}
$$

The last inequality holds, due to Lemma 3 and $\widehat{\boldsymbol{\Sigma}}$ is obtained using independent samples. Conditioned on the same event $\frac{1}{2}\boldsymbol{\Sigma} \leq \widehat{\boldsymbol{\Sigma}} \leq \frac{3}{2}\boldsymbol{\Sigma}$, we have

$$
\left\| \widehat{\boldsymbol{\Sigma}}^{-1/2} \boldsymbol{\Sigma}^{1/2} \right\|_{\mathrm{op}}^2 = \left\| \boldsymbol{\Sigma}^{1/2} \widehat{\boldsymbol{\Sigma}}^{-1} \boldsymbol{\Sigma}^{1/2} \right\|_{\mathrm{op}} \leq 2.
$$

Therefore, $M_{h,\mathrm{op}}^2 \leq 3B_h^{-2}$. Putting all the ingredients together and applying Theorem 1, by choosing

$$
m \geq \max \left\{ \frac{10584 r_\star^3 D \left(1 + \log \frac{8}{\delta}\right) \sum_{s=1}^{r_\star} \left\| (\boldsymbol{\beta}_s^\top \mathbf{x})^{p_s} \right\|_{L_2}}{\delta \epsilon_0^2} \log \frac{7r_\star}{\sqrt{\delta}\epsilon_0}, \ \frac{108 C^2 D^2 r_\star^2}{\epsilon_0 \sqrt{2\lambda_0}} \right\},
$$

we establish for any SOSP $\widehat{\mathbf{W}}$, the generalization error bounded by:

$$
\mathbb{E}_{(\mathbf{x}_i,y_i)} \left[ \left| \mathcal{R}(f_{\widehat{\mathbf{W}}}^Q) - \widehat{\mathcal{R}}(f_{\widehat{\mathbf{W}}}^Q) \right| \right] \leq \widetilde{O} \left( \frac{\|\mathbf{h}(\mathbf{x})\|_2^2 B_{w,\star}^2 M_{h,\mathrm{op}}}{\sqrt{n}} \right) = \widetilde{O} \left( \sqrt{\frac{2\lambda_{\lceil p/2 \rceil}^{-1} D r_\star^2}{n}} \right).
$$

Using Markov's inequality, we have

$$
\begin{aligned}
\mathbb{P} \left( \left| \mathcal{R}(f_{\widehat{\mathbf{W}}}^Q) - \widehat{\mathcal{R}}(f_{\widehat{\mathbf{W}}}^Q) \right| \geq \epsilon_0 \right) &\leq \frac{\mathbb{E}_{(\mathbf{x}_i,y_i)} \left[ \left| \mathcal{R}(f_{\widehat{\mathbf{W}}}^Q) - \widehat{\mathcal{R}}(f_{\widehat{\mathbf{W}}}^Q) \right| \right]}{\epsilon_0} \\
&\leq \widetilde{O} \left( \epsilon_0^{-1} \sqrt{\frac{2\lambda_{\lceil p/2 \rceil}^{-1} D r_\star^2}{n}} \right).
\end{aligned}
$$

We set the above probability upper bounded by $\delta$, which requires

$$
n = \widetilde{O} \left( \frac{\lambda_{\lceil p/2 \rceil}^{-1} r_\star^7}{\epsilon_0^4 \delta^3} \left( \sum_{s=1}^{r_\star} (p_s + 1)^5 \|\boldsymbol{\beta}_s\|_2^{2\lceil p_s/2 \rceil} \right) \right).
$$

We can now bound $\mathcal{R}(f_{\widehat{W}}^Q)$ as

$$
\mathcal{R}(f_{\widehat{\mathbf{W}}}^Q) = \mathcal{R}(f_{\widehat{\mathbf{W}}}^Q) - \widehat{\mathcal{R}}(f_{\widehat{\mathbf{W}}}^Q) + \widehat{\mathcal{R}}(f_{\widehat{\mathbf{W}}}^Q) \leq (1 + \tau_0)\mathsf{OPT} + 2\epsilon_0,
$$

which holds with probability at least $1 - \delta$.

Taking $\|\boldsymbol{\beta}_s\|_2 = \sqrt{d}$, the sample size $n$ grows in the order of $\widetilde{O} \left( \frac{d^{\lceil p/2 \rceil}}{\epsilon_0^4} \frac{\lambda_{\lceil p/2 \rceil}^{-1} r_\star^8 p^5}{\delta^3} \right)$. On the other hand, estimating covariance matrix $\boldsymbol{\Sigma}$ requires $n_0 = \widetilde{O} \left( 4\delta^{-2} \lambda_{\lceil p/2 \rceil}^{-1} D \log D \right)$ samples, which is in the order of $\widetilde{O} \left( \frac{d^{\lceil p/2 \rceil}}{\epsilon_0^2} \frac{\lambda_{\lceil p/2 \rceil}^{-1} r_\star^6 p^5}{\delta^2} \right)$. Adding $n_1, n_2$ together, the sample complexity $n$ grows in the order of $\widetilde{O} \left( \frac{d^{\lceil p/2 \rceil}}{\epsilon_0^4} \mathrm{poly}(r_\star, p, \delta^{-1}) \right)$.

### D.4 Proof of Claim 1

*Proof.* To show $\sup_{\mathbf{x}\in\mathbb{S}^{d-1}} \frac{1}{m_0}\sum_{r=1}^{m_0} 2\mathbb{1}\{\mathbf{w}_{0,r}^\top \mathbf{h}(\mathbf{x}) \geq 0\}$ is well concentrated, we observe that by symmetry, the following holds

$$\sup_{\mathbf{x}\in\mathbb{S}^{d-1}} \frac{1}{m_0}\sum_{r=1}^{m_0} 2\mathbb{1}\{\mathbf{w}_{0,r}^\top \mathbf{h}(\mathbf{x}) \geq 0\} = \sup_{\mathbf{x}\in\mathbb{S}^{d-1}} \frac{1}{m_0}\sum_{r=1}^{m_0} 2\mathbb{1}\{\mathbf{w}_{0,r}^\top \mathbf{h}(\mathbf{x})/\left\|\mathbf{h}(\mathbf{x})\right\|_2 \geq 0\}$$

$$\leq \sup_{\mathbf{y}\in\mathbb{S}^{D-1}} \frac{1}{m_0}\sum_{r=1}^{m_0} 2\mathbb{1}\{\mathbf{w}_{0,r}^\top \mathbf{y} \geq 0\}.$$

For a given $\mathbf{y}$, each $2\,\mathbb{1}\{\mathbf{w}_{0,r}^\top \mathbf{y} \geq 0\}$ is bounded in $[0,2]$, hence it is sub-Gaussian with variance proxy 1. Using the Hoeffding's inequality, for every $\mathbf{y}$, we have

$$\mathbb{P}\left(\left|\frac{1}{m_0}\sum_{r=1}^{m_0} 2\,\mathbb{1}\{\mathbf{w}_{0,r}^\top \mathbf{y} \geq 0\} - 1\right| \geq t\right) \leq 2\exp\left(-m_0 t^2/2\right).$$

To bound the supremum, we discretize the unit sphere. Let $\{\bar{\mathbf{y}}_i\}_{i=1}^{\mathsf{N}(\gamma,\mathbb{S}^{D-1},\|\cdot\|_2)}$ be a $\gamma$-covering of $\mathbb{S}^{D-1}$ with $\gamma < 1$, where $\mathcal{N}(\gamma,\mathbb{S}^{D-1},\|\cdot\|_2)$ denotes the covering number. By the volume ratio argument, we bound $\mathcal{N}(\gamma,\mathbb{S}^{D-1},\|\cdot\|_2) \leq \left(\frac{3}{\gamma}\right)^D$. Applying the union bound, we derive

$$\mathbb{P}\left(\max_{\mathbf{y}\in\{\bar{\mathbf{y}}_i\}_{i=1}^{\mathcal{N}(\gamma,\mathbb{S}^{D-1},\|\cdot\|_2)}} \left|\frac{1}{m_0}\sum_{r=1}^{m_0} 2\,\mathbb{1}\{\mathbf{w}_{0,r}^\top \mathbf{y} \geq 0\} - 1\right| \geq t\right) \leq 2\mathcal{N}(\gamma,\mathbb{S}^{D-1},\|\cdot\|_2)\exp\left(-m_0 t^2/2\right)$$

$$\leq 2\exp\left(-m_0 t^2 + D\log\frac{3}{\gamma}\right).$$

Taking $t = \sqrt{\frac{D\log\frac{3}{\gamma}\left(1+\frac{1}{D}\log\frac{2}{\delta}\right)}{m_0}}$, with probability at least $1-\delta$, we have

$$\max_{\mathbf{y}\in\{\bar{\mathbf{y}}_i\}_{i=1}^{\mathsf{N}(\gamma,\mathbb{S}^{D-1},\|\cdot\|_2)}} \left|\frac{1}{m_0}\sum_{r=1}^{m_0} 2\,\mathbb{1}\{\mathbf{w}_{0,r}^\top \mathbf{y} \geq 0\} - 1\right| \leq \sqrt{\frac{D\log\frac{3}{\gamma}\left(1+\frac{1}{D}\log\frac{2}{\delta}\right)}{m_0}}. \qquad (15)$$

By the definition of $\gamma$-covering, for any given $\mathbf{y}\in\mathbb{S}^{D-1}$, there exists $\bar{\mathbf{y}}$ such that $\|\bar{\mathbf{y}}-\mathbf{y}\|_2 \leq \gamma$. We evaluate how many pairs $\mathbb{1}\{\mathbf{w}_{0,r}^\top \mathbf{y} \geq 0\}, \mathbb{1}\{\mathbf{w}_{0,r}^\top \bar{\mathbf{y}} \geq 0\})$ taking different values, which is equivalent to $(\mathbf{w}_{0,r}^\top \mathbf{y}, \mathbf{w}_{0,r}^\top \bar{\mathbf{y}})$ having opposite signs. Using the Hoeffding's inequality again, with probability at least $1-\delta$, we have

$$\left|\frac{1}{m_0}\sum_{r=1}^{m_0} \mathbb{1}\{\mathbf{w}_{0,r}^\top \mathbf{y}, \mathbf{w}_{0,r}^\top \bar{\mathbf{y}} \text{ having opposite signs}\}\right.$$

$$\left. - \mathbb{E}\left[\frac{1}{m_0}\sum_{r=1}^{m_0} \mathbb{1}\{\mathbf{w}_{0,r}^\top \mathbf{y}, \mathbf{w}_{0,r}^\top \bar{\mathbf{y}} \text{ having opposite signs}\}\right]\right| \leq \sqrt{\frac{\log(2/\delta)}{2m_0}}.$$

To bound the expectation, we observe that $(\mathbf{w}_{0,r}^\top \mathbf{y}, \mathbf{w}_{0,r}^\top \bar{\mathbf{y}})$ is jointly Gaussian with zero mean and the covariance matrix

$$\begin{pmatrix} 1 & \mathbf{y}^\top \bar{\mathbf{y}} \\ \mathbf{y}^\top \bar{\mathbf{y}} & 1 \end{pmatrix}.$$

Therefore, we find the following probability

$$\mathbb{P}\left(\mathbf{w}_{0,r}^\top \mathbf{y}, \mathbf{w}_{0,r}^\top \bar{\mathbf{y}} \text{ opposite signs}\right) = 2\mathbb{P}\left(\mathbf{w}_{0,r}^\top \mathbf{y} \geq 0, \mathbf{w}_{0,r}^\top \bar{\mathbf{y}} \leq 0\right)$$

$$= 2\int_0^\infty \int_{-\infty}^0 \frac{1}{2\pi}\left(1-(\mathbf{y}^\top \bar{\mathbf{y}})^2\right)^{-1/2}\exp\left(-\frac{u^2 - (\mathbf{y}^\top \bar{\mathbf{y}})uv + v^2}{2(1-(\mathbf{y}^\top \bar{\mathbf{y}})^2)}\right)dudv$$

$$\overset{(i)}{\leq} 2\int_0^\infty \int_{-\infty}^0 \frac{1}{2\pi}\left(1-(\mathbf{y}^\top \bar{\mathbf{y}})^2\right)^{-1/2}\exp\left(-\frac{u^2 + v^2}{2(1-(\mathbf{y}^\top \bar{\mathbf{y}})^2)}\right)dudv$$

$$= 2\int_0^\infty \frac{1}{\sqrt{2\pi}}\exp\left(-\frac{v^2}{2(1-(\mathbf{y}^\top \bar{\mathbf{y}})^2)}\right)dv$$

$$= \sqrt{1-(\mathbf{y}^\top \bar{\mathbf{y}})^2},$$

where inequality $(i)$ holds since $uv < 0$. We further bound $1 - (\mathbf{y}^\top \bar{\mathbf{y}})^2 = 1 - (1 + \mathbf{y}^\top (\bar{\mathbf{y}} - \mathbf{y}))^2 \leq 1 - (1 - \gamma)^2 \leq 2\gamma$. Consequently, we deduce $\mathbb{P}\left(\mathbf{w}_{0,r}^\top \mathbf{y}, \mathbf{w}_{0,r}^\top \bar{\mathbf{y}} \text{ having opposite signs}\right) \leq \sqrt{2\gamma}$. Taking $\gamma = m_0^{-1} \log 1/\delta$, we have

$$\mathbb{E}\left[\frac{1}{m_0} \sum_{r=1}^{m_0} \mathbb{1}\{\mathbf{w}_{0,r}^\top \mathbf{y}, \mathbf{w}_{0,r}^\top \bar{\mathbf{y}} \text{ having opposite signs}\}\right] \leq \sqrt{\frac{2\log(2/\delta)}{m_0}}.$$

This implies with probability at least $1 - \delta$,

$$\sqrt{\frac{\log(2/\delta)}{2m_0}} \leq \frac{1}{m_0} \sum_{r=1}^{m_0} \mathbb{1}\{\mathbf{w}_{0,r}^\top \mathbf{y}, \mathbf{w}_{0,r}^\top \bar{\mathbf{y}} \text{ having opposite signs}\} \leq \sqrt{\frac{9\log(2/\delta)}{2m_0}}. \qquad (16)$$

Combining (15) and (16) together, with probability at least $1 - 2\delta$, we deduce

$$\sup_{\mathbf{y} \in \mathbb{S}^{d-1}} \left|\frac{1}{m_0} \sum_{r=1}^{m_0} 2\,\mathbb{1}\{\mathbf{w}_{0,r}^\top \mathbf{y} \geq 0\} - 1\right|$$

$$\leq \sup_{\|\mathbf{y} - \bar{\mathbf{y}}\|_2 \leq m_0^{-1} \log 1/\delta} \left|\frac{1}{m_0} \sum_{r=1}^{m_0} 2\,\mathbb{1}\{\mathbf{w}_{0,r}^\top \mathbf{y}, \mathbf{w}_{0,r}^\top \bar{\mathbf{y}} \text{ having opposite signs}\}\right|$$

$$+ \max_{\mathbf{y} \in \{\bar{\mathbf{y}}_i\}_{i=1}^{N(\gamma, \mathbb{S}^{D-1}, \|\cdot\|_2)}} \left|\frac{1}{m_0} \sum_{r=1}^{m_0} 2\,\mathbb{1}\{\mathbf{w}_{0,r}^\top \mathbf{y} \geq 0\} - 1\right|$$

$$\leq \sqrt{\frac{9\log\frac{2}{\delta}}{2m_0}} + \sqrt{\frac{D\log(3m_0)\left(1 + \frac{1}{D}\log\frac{2}{\delta}\right)}{m_0}}$$

$$\leq 6\sqrt{\frac{D\log(3m_0)\left(1 + \log\frac{2}{\delta}\right)}{m_0}}.$$

As a result, we know

$$\sup_{\mathbf{x} \in \mathbb{S}^{d-1}} \left|\frac{1}{m_0} \sum_{r=1}^{m_0} 2\mathbb{1}\{\mathbf{w}_{0,r}^\top \mathbf{h}(\mathbf{x}) \geq 0\} - 1\right| \leq 6\sqrt{\frac{D\log(3m_0)\left(1 + \log\frac{2}{\delta}\right)}{m_0}}$$

holds with probability at least $1 - 2\delta$. $\qquad\square$

### D.5 Learning in `Quad-Neural` with Data Dependent Regularizer

We consider using data dependent regularizer for learning with unwhitened features $\mathbf{g}(\mathbf{x})$, which also yields improved sample complexity. The full learning algorithm is described Algorithm 2.

Note that `Quad-g` shares the same QuadNTK model as `Quad-h`, and only replaces the neural representation $\mathbf{h}$ with $\mathbf{g}$. The superscript on $\widehat{\mathcal{R}}_\lambda^{\mathrm{dreg}}$ stands for data dependent regularization. We show `Quad-g` enjoys a similarly nice optimization landscape and good generalization properties as `Quad-h`.

**Theorem 1$'$** (Optimization landscape and generalization of `Quad-g`)**.** Suppose Assumption 2 holds.

(1) (Optimization) Given any $\epsilon > 0$ and $\delta > 0$, $\tau = \Theta(1)$, and some radius $B_{w,\star} > 0$, suppose the width $m \geq \widetilde{O}(D^2 B_{w,\star}^4 \epsilon^{-1})$, sample size $n_0 = \widetilde{O}(\delta^{-2} D)$, and we choose a proper regularization coefficient $\lambda > 0$. Then with probability $1 - \delta$ over $\widetilde{S}_{n_0}$, any second-order stationary point $\widehat{\mathbf{W}}$ of the regularized risk $\widehat{\mathcal{R}}_\lambda^{\mathrm{dreg}}(f_{\mathbf{W}}^Q)$ satisfies $\|\widehat{\mathbf{W}}\widehat{\boldsymbol{\Sigma}}^{1/2}\|_{2,4} \leq O(B_{w,\star})$, and achieves

$$\widehat{\mathcal{R}}_\lambda^{\mathrm{dreg}}(f_{\widehat{\mathbf{W}}}^Q) \leq (1 + \tau) \min_{\|\mathbf{W}\widehat{\boldsymbol{\Sigma}}^{1/2}\|_{2,4} \leq B_{w,\star}} \widehat{\mathcal{R}}(f_{\mathbf{W}}^Q) + \epsilon.$$

(2) (Generalization) For any radius $B_w > 0$, we have with high probability (over $(\mathbf{a}, \mathbf{W}_0, \widetilde{S}_{n_0})$) that

$$\mathbb{E}_{(\mathbf{x}_i, y_i)}\left[\sup_{\|\mathbf{W}\widehat{\boldsymbol{\Sigma}}^{1/2}\|_{2,4} \leq B_w} \left|\mathcal{R}(f_{\mathbf{W}}^Q) - \widehat{\mathcal{R}}(f_{\mathbf{W}}^Q)\right|\right] \leq \widetilde{O}\left(\frac{B_g^2 B_w^2 M_{g,\mathrm{op}}}{\sqrt{n}} + \frac{1}{\sqrt{n}}\right),$$

---

**Algorithm 2** Learning with Unwhitened Neural Random Features

**Input**: Labeled data $S_n$, unlabeled data $\widetilde{S}_{n_0}$, initializations $\mathbf{V} \in \mathbb{R}^{D \times d}$, $\mathbf{b} \in \mathbb{R}^D$, $\mathbf{W}_0 \in \mathbb{R}^{m \times D}$, parameters $(\lambda, \epsilon)$.

**Step 1:** 1) Construct model $f_{\mathbf{W}}^Q$ as

$$(\text{Quad-g}) \qquad f_{\mathbf{W}}^Q(\mathbf{x}) = \frac{1}{2\sqrt{m}} \sum_{r=1}^m a_r \phi''(\mathbf{w}_{0,r}^\top \mathbf{g}(\mathbf{x}))(\mathbf{w}_r^\top \mathbf{g}(\mathbf{x}))^2,$$

where $\mathbf{g}(\mathbf{x}) = [\mathbb{1}\{\mathbf{v}_1^\top \mathbf{x} + b_1 \geq 0, \dots, \mathbf{v}_D^\top \mathbf{x} + b_D \geq 0\}]^\top$ is the neural random features.
2) Use $\widetilde{S}_{n_0}$ to estimate the covariance matrix of $\mathbf{g}(\mathbf{x})$, i.e., $\widehat{\boldsymbol{\Sigma}} = \frac{1}{n_0} \sum_{i=1}^{n_0} \mathbf{g}(\mathbf{x}_i)\mathbf{g}(\mathbf{x}_i)^\top$.

**Step 2:** Find a second-order stationary point $\widehat{\mathbf{W}}$ of the data dependent regularized empirical risk (on the data $S_n$):

$$\widehat{\mathcal{R}}_\lambda^{\text{dreg}}(f_{\mathbf{W}}^Q) = \frac{1}{n} \sum_{i=1}^n \ell(f_{\mathbf{W}}^Q(\mathbf{x}_i), y_i) + \lambda \left\| \mathbf{W}\widehat{\boldsymbol{\Sigma}}^{1/2} \right\|_{2,4}^4.$$

---

where $M_{g,\text{op}}^2 = B_g^{-2} \mathbb{E}_\mathbf{x} \left[ \left\| \frac{1}{n} \sum_{i=1}^n \mathbf{h}(\mathbf{x}_i)\mathbf{h}(\mathbf{x}_i)^\top \right\|_{\text{op}} \right]$.

*Proof of Theorem 1′, Optimization Part.* We recall the second-order directional derivative of $\widehat{\mathcal{R}}(f_{\mathbf{W}}^Q)$ satisfies

$$\nabla_\mathbf{W}^2 \widehat{\mathcal{R}}(f_{\mathbf{W}}^Q)[\mathbf{W}_\star, \mathbf{W}_\star] \leq \left\langle \nabla\widehat{\mathcal{R}}(f_{\mathbf{W}}^Q), \mathbf{W} \right\rangle - 2(\widehat{\mathcal{R}}(f_{\mathbf{W}}^Q) - \widehat{\mathcal{R}}(f_{\mathbf{W}^*}^Q)) + m^{-1} B_g^4 \|\mathbf{W}\|_{2,4}^2 \|\mathbf{W}_\star\|_{2,4}^2,$$

which is established in (9) and $\mathbf{W}^*$ is any given matrix. Note we have replaced $B_h$ with $B_g = \|\mathbf{g}(\mathbf{x})\|_2$, and $B_g$ is upper bounded by $\sqrt{D}$. Similar to the proof A.1, we specialize $\mathbf{W}^*$ to be the optimizer $\mathbf{W}^* = \text{argmin}_{\|\mathbf{W}\widehat{\boldsymbol{\Sigma}}^{1/2}\|_{2,4} \leq B_{w,\star}} \widehat{\mathcal{R}}(f_{\mathbf{W}}^Q)$ and denote its risk $\widehat{\mathcal{R}}(f_{\mathbf{W}^*}^Q) = M$. We choose the regularization coefficient as

$$\lambda = \lambda_0 B_{w,\star}^{-4},$$

where $\lambda_0$ is to be determined. We argue that any second-order stationary point $\widehat{\mathbf{W}}$ has to satisfy $\left\| \widehat{\mathbf{W}}\widehat{\boldsymbol{\Sigma}}^{1/2} \right\|_{2,4} = O(B_{w,\star})$. We already know from proof A.1 that for any $\mathbf{W}$, $\left\langle \nabla\widehat{\mathcal{R}}(f_{\mathbf{W}}^Q), \mathbf{W} \right\rangle \geq -2$ holds.

Combining with the fact

$$\left\langle \nabla_\mathbf{W} \left( \left\| \mathbf{W}\widehat{\boldsymbol{\Sigma}}^{1/2} \right\|_{2,4}^4 \right), \mathbf{W} \right\rangle = 4 \left\| \mathbf{W}\widehat{\boldsymbol{\Sigma}}^{1/2} \right\|_{2,4}^4,$$

we have simultaneously for all $\mathbf{W}$ that

$$\left\langle \nabla\widehat{\mathcal{R}}_\lambda^{\text{data}}(f_{\mathbf{W}}^Q), \mathbf{W} \right\rangle \geq \left\langle \nabla_\mathbf{W}(\lambda \left\| \mathbf{W}\widehat{\boldsymbol{\Sigma}}^{1/2} \right\|_{2,4}^4), \mathbf{W} \right\rangle + \left\langle \nabla_\mathbf{W}\widehat{\mathcal{R}}(f_{\mathbf{W}}^Q), \mathbf{W} \right\rangle$$

$$\geq 4\lambda \left\| \mathbf{W}\widehat{\boldsymbol{\Sigma}}^{1/2} \right\|_{2,4}^4 - 2.$$

Therefore we see that any stationary point $\mathbf{W}$ has to satisfy

$$\left\| \mathbf{W}\widehat{\boldsymbol{\Sigma}}^{1/2} \right\|_{2,4} \leq (2\lambda)^{-1/4}.$$

Choosing

$$\lambda_0 = \frac{1}{36}(2\tau M + \epsilon),$$

we get $36\lambda B_{w,\star}^4 = 2\tau M + \epsilon$. The second-order directional derivative of $\widehat{\mathcal{R}}_\lambda^{\mathrm{data}}(f_{\mathbf{W}}^Q)$ along direction $\mathbf{W}_\star$ is upper bounded by

$$
\begin{aligned}
\nabla_{\mathbf{W}}^2 \widehat{\mathcal{R}}_\lambda^{\mathrm{data}}(f_{\mathbf{W}}^Q)[\mathbf{W}_\star, \mathbf{W}_\star] &= \nabla_{\mathbf{W}}^2 \widehat{\mathcal{R}}^{\mathrm{data}}(f_{\mathbf{W}}^Q)[\mathbf{W}_\star, \mathbf{W}_\star] + \lambda \nabla_{\mathbf{W}}^2 \left\| \mathbf{W}\widehat{\mathbf{\Sigma}}^{1/2} \right\|_{2,4}^4 [\mathbf{W}_\star, \mathbf{W}_\star] \\
&= \nabla_{\mathbf{W}}^2 \widehat{\mathcal{R}}^{\mathrm{data}}(f_{\mathbf{W}}^Q)[\mathbf{W}_\star, \mathbf{W}_\star] + 4\lambda \sum_{r \leq m} \left( \mathbf{w}_{\star,r} \widehat{\mathbf{\Sigma}} \mathbf{w}_{\star,r} \right) \left( \mathbf{w}_r \widehat{\mathbf{\Sigma}} \mathbf{w}_r \right) \\
&\quad + 8\lambda \sum_{r \leq m} \left\langle \mathbf{w}_r \widehat{\mathbf{\Sigma}}^{1/2}, \mathbf{w}_{\star,r} \widehat{\mathbf{\Sigma}}^{1/2} \right\rangle^2 \\
&\leq \left\langle \nabla \widehat{\mathcal{R}}(f_{\mathbf{W}}^Q), \mathbf{W} \right\rangle - 2(\widehat{\mathcal{R}}(f_{\mathbf{W}}^Q) - M) + m^{-1} B_g^4 \|\mathbf{W}\|_{2,4}^2 \|\mathbf{W}_\star\|_{2,4}^2 \\
&\quad + 12\lambda \left\| \mathbf{W}\widehat{\mathbf{\Sigma}}^{1/2} \right\|_{2,4}^2 \left\| \mathbf{W}_\star \widehat{\mathbf{\Sigma}}^{1/2} \right\|_{2,4}^2 \\
&\overset{(i)}{\leq} \left\langle \nabla \widehat{\mathcal{R}}(f_{\mathbf{W}}^Q), \mathbf{W} \right\rangle - 2(\widehat{\mathcal{R}}(f_{\mathbf{W}}^Q) - M) + m^{-1} B_g^4 \|\mathbf{W}\|_{2,4}^2 \|\mathbf{W}_\star\|_{2,4}^2 \\
&\quad + \lambda \left\| \mathbf{W}\widehat{\mathbf{\Sigma}}^{1/2} \right\|_{2,4}^4 + 36\lambda \left\| \mathbf{W}_\star \widehat{\mathbf{\Sigma}}^{1/2} \right\|_{2,4}^4 \\
&\leq \left\langle \nabla \widehat{\mathcal{R}}_\lambda^{\mathrm{data}}(f_{\mathbf{W}}^Q), \mathbf{W} \right\rangle - 2(\widehat{\mathcal{R}}_\lambda^{\mathrm{data}}(f_{\mathbf{W}}^Q) - M) \\
&\quad + m^{-1} B_g^4 \|\mathbf{W}\|_{2,4}^2 \|\mathbf{W}_\star\|_{2,4}^2 - \lambda \left\| \mathbf{W}\widehat{\mathbf{\Sigma}}^{1/2} \right\|_{2,4}^4 + 36\lambda \|\mathbf{W}_\star\|_{2,4}^4.
\end{aligned}
$$

We used the fact $12ab \leq a^2 + 36b^2$. For a second order-stationary point $\widehat{\mathbf{W}}$ of $\widehat{\mathcal{R}}_\lambda(f_{\mathbf{W}}^Q)$, its gradient vanishes and the Hessian is positive definite. Therefore, we have

$$
0 \leq -2(\widehat{\mathcal{R}}_\lambda^{\mathrm{data}}(f_{\widehat{\mathbf{W}}}^Q) - M) + m^{-1} B_g^4 \left\| \widehat{\mathbf{W}} \right\|_{2,4}^2 \|\mathbf{W}_\star\|_{2,4}^2 - \lambda \left\| \widehat{\mathbf{W}} \widehat{\mathbf{\Sigma}}^{1/2} \right\|_{2,4}^4 + 36\lambda \left\| \mathbf{W}_\star \widehat{\mathbf{\Sigma}}^{1/2} \right\|_{2,4}^4.
$$

By Assumption 2, we have $\lambda_{\min}(\mathbf{\Sigma}) \geq \lambda_k$. Moreover, by Lemma 3, when $n_0 = O\left( \delta^{-2} D \log D \right)$, with probability at least $1 - \delta$, we have the following relative concentration of $\widehat{\mathbf{\Sigma}}$:

$$
\frac{1}{2}\mathbf{\Sigma} \preceq \widehat{\mathbf{\Sigma}} \preceq \frac{3}{2}\mathbf{\Sigma}.
$$

Combining these two ingredients together, we deduce

$$
\begin{aligned}
\|\mathbf{W}\|_{2,4}^4 = \sum_{r=1}^m \left\| \widehat{\mathbf{\Sigma}}^{-1/2} \widehat{\mathbf{\Sigma}}^{1/2} \mathbf{w}_r \right\|_2^4 &\leq \sum_{r=1}^m \left\| \widehat{\mathbf{\Sigma}}^{-1/2} \right\|_{\mathrm{op}}^4 \left\| \widehat{\mathbf{\Sigma}}^{1/2} \mathbf{w}_r \right\|^4 \\
&\leq \sum_{r=1}^m 4\lambda_k^{-2} \left\| \widehat{\mathbf{\Sigma}}^{1/2} \mathbf{w}_r \right\|^4 = 4\lambda_k^{-2} \left\| \mathbf{W}\widehat{\mathbf{\Sigma}} \right\|_{2,4}^4.
\end{aligned}
$$

Exactly the same argument yields $\|\mathbf{W}_\star\|_{2,4}^4 \leq 4\lambda_k^{-2} \left\| \mathbf{W}_\star \widehat{\mathbf{\Sigma}} \right\|_{2,4}^4$. Therefore, we choose $m = 4\epsilon^{-1} \lambda_k^{-2} (2\lambda_0)^{-1/2} C^2 B_g^4 B_{w,\star}^4 \geq \epsilon^{-1} C^2 B_g^4 \|\widehat{\mathbf{W}}\|_{2,4}^2 \|\mathbf{W}_\star\|_{2,4}^2$ and the above inequality implies

$$
\begin{aligned}
&2(\widehat{\mathcal{R}}_\lambda^{\mathrm{data}}(f_{\widehat{\mathbf{W}}}^Q) - M) \leq 2\tau M + \epsilon + \epsilon \\
&\implies \widehat{\mathcal{R}}_\lambda^{\mathrm{data}}(f_{\widehat{\mathbf{W}}}^Q) \leq (1 + \tau)M + \epsilon.
\end{aligned}
$$

Plugging in the naive upper bound $\|\mathbf{g}(\mathbf{x})\|_2 \leq \sqrt{D}$ in $B_g$, the proof is complete. $\qquad\square$

*Proof of Theorem 1', Generalization Part.* Built upon the proof A.2, we have

$$
\mathbb{E}_{(\mathbf{x}_i, y_i)} \left[ \sup_{\left\| \mathbf{W}\widehat{\mathbf{\Sigma}}^{1/2} \right\|_{2,4} \leq B_w} \left| \mathcal{R}(f_{\mathbf{W}}^Q) - \widehat{\mathcal{R}}(f_{\mathbf{W}}^Q) \right| \right] \leq 2\mathbb{E}_{(\mathbf{x}_i, y_i), \boldsymbol{\xi}} \left[ \sup_{\left\| \mathbf{W}\widehat{\mathbf{\Sigma}}^{1/2} \right\|_{2,4} \leq B_w} \left| \frac{1}{n} \sum_{i=1}^n \xi_i \ell(f_{\mathbf{W}}^Q(\mathbf{x}_i), y_i) \right| \right],
$$

where $\xi$ is i.i.d. Rademacher random variables. Recall that the whitened feature is $\mathbf{h}(\mathbf{x}) = \widehat{\boldsymbol{\Sigma}}^{-1/2}\mathbf{g}(\mathbf{x})$. We further have

$$
\mathbb{E}_{(\mathbf{x}_i, y_i), \boldsymbol{\xi}} \left[ \left| \left| \sup_{\|\mathbf{W}\widehat{\boldsymbol{\Sigma}}^{1/2}\|_{2,4} \leq B_w} \frac{1}{n} \sum_{i=1}^{n} \xi_i \ell(y_i, f^Q_{\mathbf{W}}(\mathbf{x}_i)) \right| \right| \right]
$$

$$
\leq 4\mathbb{E}_{\mathbf{x}_i, \boldsymbol{\xi}} \left[ \sup_{\|\mathbf{W}\widehat{\boldsymbol{\Sigma}}^{1/2}\|_{2,4} \leq B_w} \frac{1}{\sqrt{m}} \sum_{r \leq m} \left\langle \frac{1}{n} \sum_{i=1}^{n} \xi_i a_r \phi''(\mathbf{w}_{0,r}^\top \mathbf{g}(\mathbf{x}_i)) \mathbf{h}(\mathbf{x}_i)\mathbf{h}(\mathbf{x}_i)^\top, \widehat{\boldsymbol{\Sigma}}^{1/2}\mathbf{w}_r\mathbf{w}_r^\top\widehat{\boldsymbol{\Sigma}}^{1/2} \right\rangle \right]
$$

$$
+ \frac{2}{\sqrt{n}}
$$

$$
\leq 4\mathbb{E}_{\mathbf{x}_i, \boldsymbol{\xi}} \left[ \sup_{\|\mathbf{W}\widehat{\boldsymbol{\Sigma}}^{1/2}\|_{2,4} \leq B_w} \max_{r \in [m]} \left| \left| \frac{1}{n} \sum_{i=1}^{n} \xi_i \phi''(\mathbf{w}_{0,r}^\top \mathbf{h}(\mathbf{x}_i)) \mathbf{h}(\mathbf{x}_i)\mathbf{h}(\mathbf{x}_i)^\top \right| \right|_{\mathrm{op}} \right.
$$

$$
\times \left. \frac{1}{\sqrt{m}} \sum_{r \leq m} \left\| \widehat{\boldsymbol{\Sigma}}^{1/2}\mathbf{w}_r\mathbf{w}_r^\top\widehat{\boldsymbol{\Sigma}}^{1/2} \right\|_* \right] + \frac{2}{\sqrt{n}}
$$

$$
\leq 4\mathbb{E}_{\mathbf{x}_i, \boldsymbol{\xi}} \left[ \max_{r \in [m]} \left| \left| \frac{1}{n} \sum_{i=1}^{n} \xi_i \phi''(\mathbf{w}_{0,r}^\top \mathbf{h}(\mathbf{x})_i) \mathbf{h}(\mathbf{x}_i)\mathbf{h}(\mathbf{x}_i^\top) \right| \right|_{\mathrm{op}} \right]
$$

$$
\times \underbrace{\sup_{\|\mathbf{W}\widehat{\boldsymbol{\Sigma}}^{1/2}\|_{2,4} \leq B_w} \frac{1}{\sqrt{m}} \sum_{r \leq m} \left\| \widehat{\boldsymbol{\Sigma}}^{1/2}\mathbf{w}_r \right\|_2^2}_{\leq B_w^2} + \frac{2}{\sqrt{n}},
$$

Consequently, the generalization error is still bounded by

$$
\mathbb{E}_{(\mathbf{x}_i, y_i)} \left[ \sup_{\|\mathbf{W}\widehat{\boldsymbol{\Sigma}}^{1/2}\|_{2,4} \leq B_w} \left| \mathcal{R}(f^Q_{\mathbf{W}}) - \widehat{\mathcal{R}}(f^Q_{\mathbf{W}}) \right| \right] \leq \widetilde{O}\left( \frac{B_h^2 B_w^2 M_{h,\mathrm{op}}}{\sqrt{n}}\sqrt{\log(Dm)} + \frac{1}{\sqrt{n}} \right).
$$

$\square$

When using $\texttt{Quad-g}$ to learn low-rank polynomials in the form of

$$
f_\star(\mathbf{x}) = \sum_{s=1}^{r_\star} \alpha_s (\boldsymbol{\beta}_s^\top \mathbf{x})^{p_s} \quad \text{defined in (5)},
$$

we derive the following sample complexity bound.

**Theorem 2$'$** (Sample complexity of $\texttt{Quad-g}$)**.** Suppose Assumption 2 holds, and there exists some $f_\star$ that achieves low risk: $\mathcal{R}(f_\star) \leq \mathsf{OPT}$. Then for any $\epsilon, \delta \in (0, 1)$ and $\tau = \Theta(1)$, choosing

$$
D = \Theta\left( \mathrm{poly}(r_\star, p) \sum_s \|\boldsymbol{\beta}_s\|_2^{2\lceil p_s/2 \rceil} \epsilon^{-2}\delta^{-1} \right), \quad m \geq \widetilde{O}\left(\mathrm{poly}(r_\star, D)\epsilon^{-2}\delta^{-1}\right), \quad (17)
$$

$n_0 = \widetilde{O}(D\delta^{-2})$, and a proper $\lambda > 0$, Algorithm 2 achieves the following guarantee: with probability at least $1 - \delta$ over the randomness of data and initialization, any second-order stationary point $\widehat{\mathbf{W}}$ of $\widehat{\mathcal{R}}_\lambda^{\mathrm{dreg}}(f^Q_{\mathbf{W}})$ satisfies

$$
\mathcal{R}(f^Q_{\widehat{\mathbf{W}}}) \leq (1 + \tau)\mathsf{OPT} + \epsilon + \widetilde{O}\left( \sqrt{\frac{\mathrm{poly}(r_\star, p, \delta^{-1})\lambda_{\lceil p/2 \rceil}^{-1}\epsilon^{-2}\sum_{s=1}^{r_\star}\|\boldsymbol{\beta}_s\|_2^{2\lceil p_s/2 \rceil}}{n}} \right).
$$

In particular, for any $\epsilon > 0$, we can achieve $\mathcal{R}(f^Q_{\widehat{\mathbf{W}}}) \leq (1 + \tau)\mathsf{OPT} + 2\epsilon$ with sample complexity

$$
n_0 + n \leq \widetilde{O}\left( \mathrm{poly}(r_\star, p, \lambda_{\lceil p/2 \rceil}^{-1}, \epsilon^{-1}, \delta^{-1}) \sum_{s=1}^{r_\star} \|\boldsymbol{\beta}_s\|_2^{2\lceil p_s/2 \rceil} \right). \quad (18)
$$

*Proof.* The proof reproduces that for Quad-**h** in Sections D.1, D.2, and D.3. Specifically, following the same argument in Lemma 9, we can establish the expressivity of Quad-**h**, where for $r = 1, \ldots, m_0$, we only need to choose

$$\mathbf{w}_r^{s,*} = \begin{cases} 2\sqrt{\alpha_s}(3r_\star)^{1/4}m_0^{-1/4}[\mathbf{0}^\top, \ldots, \mathbf{a}_s^\top, \ldots, \mathbf{0}^\top]^\top, & \text{for the } s\text{-th group in } \mathcal{I}_1 \text{ with } \alpha_s > 0 \\ 2\sqrt{|\alpha_s|}(3r_\star)^{1/4}m_0^{-1/4}[\mathbf{0}^\top, \ldots, \mathbf{a}_s^\top, \ldots, \mathbf{0}^\top]^\top, & \text{for the } s\text{-th group in } \mathcal{I}_2 \text{ with } \alpha_s < 0 \end{cases}.$$

Remember $\mathcal{I}_1 = \{1, \ldots, m/3\}$ where $a_r = 1$ for $r \in \mathcal{I}_1$ and $\mathcal{I}_2 = \{m/3 + 1, 2m/3\}$ with $a_r = -1$. Compared to using whitened representation $\mathbf{h}(\mathbf{x})$, we remove the multiplicative factor $\widehat{\mathbf{\Sigma}}^{1/2}$ in $\mathbf{w}_r^*$ (see Lemma 8). The corresponding representation dimension $D = \frac{8 \times 50^2 r_\star^3 \sum_{s=1}^{r_\star} p_s^5 \|\boldsymbol{\beta}_s\|_2^{2\lceil p_s/2 \rceil}}{\epsilon^2 \delta}$ and the width $m \geq \frac{54 r_\star D (1 + \log \frac{8}{\delta})}{\epsilon^2} \log \frac{1}{\epsilon}$ remain unchanged. Then with probability $1 - \delta$, we have

$$\left\| \frac{1}{2\sqrt{m}} \sum_{r=1}^m a_r \mathbb{1}\{\mathbf{w}_{0,r}^\top \mathbf{g}(\mathbf{x}) \geq 0\} \left((\mathbf{w}_r^*)^\top \mathbf{g}(\mathbf{x})\right)^2 - f(\mathbf{x}) \right\|_{L_2} \leq 7 r_\star \epsilon.$$

The rest of the proof follows Section D.3, where we need to upper bound $M_{g,\mathrm{op}}$, $B_{w,\star}$, and $B_g$, respectively. We use the naive upper bound on $B_g \leq \sqrt{D}$, since each entry of $\mathbf{g}(\mathbf{x})$ is bounded by 1. By definition, we have

$$B_g^2 M_{g,\mathrm{op}}^2 = B_h^2 M_{h,\mathrm{op}}^2 \leq 3.$$

Lastly, observe $B_{w,\star}^4 = \left\| \mathbf{W}^* \widehat{\mathbf{\Sigma}}^{1/2} \right\|_{2,4}^4 = \sum_{r=1}^m \left\| \widehat{\mathbf{\Sigma}}^{1/2} \mathbf{w}_r^* \right\|_2^4$. An upper bound has been already derived in Section D.3, which is $108 r_\star^2$. As can be seen, quantities $M_{g,\mathrm{op}}$, $B_{w,\star}$, and $B_g$ all retain the same order as using the whitened neural representation $\mathbf{h}(\mathbf{x})$ (with possibly different absolute constants). Therefore, in order to achieve

$$\mathcal{R}(f_{\widehat{\mathbf{W}}}^Q) = \mathcal{R}(f_{\widehat{\mathbf{W}}}^Q) - \widehat{\mathcal{R}}(f_{\widehat{\mathbf{W}}}^Q) + \widehat{\mathcal{R}}(f_{\widehat{\mathbf{W}}}^Q) \leq (1 + \tau)\mathsf{OPT} + 2\epsilon,$$

the sample complexity needs to satisfy

$$n = \widetilde{O}\left( \frac{\lambda_{\lceil p/2 \rceil}^{-1} r_\star^7}{\epsilon_0^4 \delta^3} \left( \sum_{s=1}^{r_\star} (p_s + 1)^5 \|\boldsymbol{\beta}_s\|_2^{2\lceil p_s/2 \rceil} \right) \right),$$

and $n_0$ stays the same for the covariance estimation. This yields the same sample complexity (again with a potentially different absolute constant) as using the whitened representation $\mathbf{h}(\mathbf{x})$. $\qquad\square$