[Reviews · NeurIPS 2020]

Review 1

Summary and Contributions: Theoretical analysis aimed at showing that neural network representations (a combination of an affine and indicator transformations), can be beneficial for supervised learning. The authors postulate that sufficiently complex models (such as a quadratic Taylor model) can capitalise on the neural representations. To support this claim, they provide sample complexity analysis for fitting low-degree polynomials (up to 4th order); the savings obtained with the quadratic Taylor model are contrasted with the lack thereof when feeding the neural representations to the linear (Neural Tangent Kernel) model. Moreover, the authors extend the work of Bay and Lee (2020) on the generalisation landscape of the quadratic models: generalisation bounds are provided for a more general case of using representation h(x), rather than x as the network input.

Strengths: The work is well written and theoretically sound. The extended material includes well described proofs for all theorems. NeurIPS community should benefit from the theoretical intuition provided, particularly as it addresses the benefits of increasing network depth, a go-to method for any ML practitioner.

Weaknesses: The authors did a great job describing the problem setup and summarising the conclusions. Understanding all the assumptions and theorems is more of a challenge, especially that not all symbols are explained (how does B_w,star differ from B_w? What is the role of the epsilon in Algorithm 1? Is lambda_min(Sigma) the smallest eigenvalue? What is q(z) in line 174? Please, describe.) I would suggest adding a Supplementary table with all the symbols explained. One could see the assumptions of the analysis as another limitation, such as whitening the neural representation (albeit the authors suggest this might be replaced by a data-dependent online normalisation), bounding loss to be not greater than 1, the L2 norm of the polynomial, etc, etc. This “limitation” is however typical for theoretical approaches and as such should not impact the paper’s acceptance.

Correctness: The claims and methods appear correct, I did not review all the proofs in detail. Theorem 1 (Generalization) appears to be missing the sqrt(log(Dm)) factor, but it seems to be a typo (it’s referred to later in the text).

Clarity: Yes, but see above about the absolute clarity of the mathematical notation.

Relation to Prior Work: Yes

Reproducibility: Yes

Additional Feedback: Please, see above. ---- POST FEEDBACK ---- Thank you for your response and addressing all the points above. I increased my confidence score as I trust the final submission will be exceptionally clearly presented.


Review 2

Summary and Contributions: This is a theoretical paper on supervised learning. It takes off from many works which show that learning dynamics linearizes in some regimes, most notably the wide network limit which leads to the neural tangent kernel (NTK). The authors study simple limits introduced in ref. [8] which expand the activation function around the initial condition and keep the degree k term. The k=1 case is closely related to the NTK. Here they look at a 3-layer network in which the first layer is a fixed random feature model. The main claim is that a k=2 model can fit a degree p polynomial in d dimensions with d^(p/2) samples instead of d^{p-1} for k=1 or the NTK, in both cases with the random feature layer. It is suggested that this illustrates a reason that real nonlinear NN models can perform better than linearized models and the NTK.

Strengths: The basic claim is clearly stated and I think it is new. It is supported with rigorous results whose proofs are spelled out in great detail in the supplement.

Weaknesses: It would be helpful to say more about what is responsible for the difference between the quadratic model and the linear models. The lower bound for the linear model is just quoted from [27], which is not so helpful. Also, how much does the main result say about the advantages of a nonlinear NN over a linearized NN or a kernel model. It seems plausible that as training proceeds, the higher order terms become more important, but I'm not sure this paper gives evidence for that. Note added: the authors' feedback on this point was helpful.

Correctness: Looks correct.

Clarity: Yes.

Relation to Prior Work: Yes.

Reproducibility: Yes

Additional Feedback:


Review 3

Summary and Contributions: This paper looks at the benefits of intermediate representations of deep neural networks. For doing so, it compares linearized and quadratic approximations to one- and two-layer neural networks in terms of their ability to learn low-rank polynomials. The paper establishes that while in linearized models, depth does not influence this ability to learn, in quadratic models, the two-layer networks have provably lower sample complexity than one-layer networks.

Strengths: This paper makes very sound claims, is thorough, clean and most of all, things are explained well. This goes as far as annotating different parts of a bound with explanations of what conditions give rise to it, it really helps when reading. I am not an expert in this particular sub-field, but I'm convinced that the claims made hold and that the work is significant, relevant and novel. Further, the results are interpreted well and it feels like a coherent story that is built around the central claims.

Weaknesses: Again, I am not an expert, so my questions are conceptual, and my score should be interpreted as "undecided", but if I'm convinced by your answers or the other reviewers that my concerns are invalid, which is very probable, I'll recommend accepting. - in 4.1, you whiten the intermediate neural representation. Since your paper explicitly wants to investigate the effect of having the neural representation, the additional whitening seems like it adds a second effect into the mix. If you need to do this because of the proofs, then at least it seems like you should also whiten the "raw" representations in section 3. Would the bounds for these change if you did? How much of the proofs rely on this whitening? - You keep the neural representation at its random initialization. How much is it really still a neural representation and not a simple random up-project of the data? When we think of neural intermediate representations, they arise because all the layers are learned, which here is not the case. I understand that you have "training the neural representation" in your future work section, but my criticism isn't that you haven't done it, my criticism is as to in what respect your results are still telling us anything about neural representations. What properties of your random representations actually make the difference for the sample complexity here? The nonlinearities? The fact that you have higher dimension? The whitening you do after? The randomness in initialization? - More a comment: I found the "Algorithm 1" box to be a bit superfluous. The paper is written very well, such that the content of the box is entirely obvious and just repetition at the point where the box appears. But if you have the space, I guess it can't hurt also.

Correctness: Yes

Clarity: Yes, very well.

Relation to Prior Work: Yes, at length.

Reproducibility: Yes

Additional Feedback: POST-FEEDBACK UPDATE: Thank you for answering my questions, it was very helpful. I've updated my score leaning towards accept. The reason it's not higher is that I am not an expert in this particular field.


Review 4

Summary and Contributions: The authors provide some theoretical results regarding why hierarchical representations in neural networks are beneficial for achieving low training error and generalization in Quadratic Taylor models. They focus on a specialized setting in which the first layer is a randomly initialized and fixed layer, and show improved generalization (in terms of sample complexity) with the additional randomly initialized layer compared to just feeding in the raw input. ======================================================= The authors addressed many of my questions. I don't think it warrants adjusting my score.

Strengths: - Address gap in the theoretical understanding of neural networks - particularly that "hierarchical learning" is extremely beneficial empirically, but there is little theory explaining why exactly hierarchical learning is so beneficial. - Demonstrate a setting in which putting the input through a randomly initialized neural network layer improves sample complexity compared to just feeding in the raw input. - Prove a lower bound result that hierarchical representations are not beneficial for neural tangent kernels.

Weaknesses: - Examine a very specialized setting in which the network is first passed through a randomly initialized network and then into a trainable network. This is non-standard for many practical neural network problems. Moreover the networks is that of a Quadratic Taylor model, which is not commonly used in practice. - Demonstrate that adding an additional layer outperforms a low-rank polynomial, which does not seem like the strongest baseline.

Correctness: The methods and claims appear to be correct.

Clarity: The paper is well-written overall.

Relation to Prior Work: It is pretty clear how the paper differs from other works. - One area you could discuss more about is how your work is related is that on overparameterized neural networks. This includes work on interpolation and generalization, as well as neural networks that are not very overparameterized (e.g. "Mildly Overparametrized Neural Nets can Memorize Training Data Efficiently" and their citations). Given your assumptions on the rate of growth of "m" the width of the hidden layer, are your models overparameterized? It would be interesting to know how your results change with smaller "m". - Additionally, since you are looking at neural networks in which the first layer is randomly initialized and not updated, it would be nice for you to discuss how your work relates to other work that construct random features (e.g. Random Features for Large-Scale Kernel Machines).

Reproducibility: Yes

Additional Feedback: - Since the quadratic Taylor model may be unfamiliar to readers, I think it would be nice to have more in the lines 59-68 section discussing the motivations for looking at this particular model. Do you have any further arguments as to why the performance of this model may inform that of other more standard neural network architectures? - a_r in equation (1) near line 43 is not defined until line 99. Would be nice to move this definition near the first time a_r is mentioned. - In Theorem 1, your results hold for "proper choice of lambda". It seems that in your proof that the proper choice of lambda depends on the empirical risk of the optimal W* (M), which is unknown. This proper choice of lambda thus seems unrealistic to ever expect in practice. Can you speak to this condition and it's reasonableness? - Your results hold for sufficiently large width of layers (m), but do not say anything about when m is small. It would nice to have more discussion on how your work relates to overparmeterization / interpolation and work on mildly overparameterized neural networks.

[Author Response · NeurIPS 2020]

We appreciate all the reviewers' valuable comments. Here is our response to the major questions raised by the reviewers.

**Reviewer #1**:

**Q**: Regarding the notation $B_{w,\star}, B_w, \lambda_{\min}$, and $q(z)$. Missing $\sqrt{\log(Dm)}$ factor in Theorem 1.

**A**: We used $B_{w,\star}$ to highlight it's the radius of the ball that contains $\mathbf{W}_\star$, and $B_w$ to denote an arbitrary radius (later we
apply this result with $B_w$ larger than $B_{w,\star}$). $\lambda_{\min}$ denotes the minimum eigenvalue. $q(z)$ in Line 174 is the dummy
variable (under the $\min$) for a degree-$k$ polynomial. $\epsilon$ in Algorithm 1 determines the choice of $D$ (as in Theorem 2). We
appreciate these suggestions and will make a supplementary table for all the symbols in our next version. Our Theorem
1 does contain a $\sqrt{\log(Dm)}$ factor (complete version is provided in Appendix Line 464 with $\sqrt{\log(Dm)}$ in the first
term). We used $\widetilde{O}(\cdot)$ to hide dependency on any log factor in the theorem statement.

**Reviewer #2**:

**Q**: What is responsible for the difference between the quadratic model and the linear models. How much does the main
result say about the advantages of a nonlinear NN over a linearized NN or a kernel model?

**A**: Our main results show that the quadratic model achieves $\widetilde{O}(d^{\lceil p/2 \rceil})$ sample complexity with neural representations,
while the linearized model / kernel suffers from at least $\Omega(d^p)$. The key thing behind is that the generalization
performance of the quadratic model depends on (and can benefit from) the conditioning of the covariance of the
input (Line 143-147). This enables the sample complexity to be reduced when we feed it with an expressive and
isotropic feature map. In contrast, linearized models/neural tangent kernels cannot benefit from feature isotropicity, and
generalizes at most as well as a kernel, as stated in the lower bound.

**Reviewer #3**:

**Q**: How does whitening affect the proof in Section 4.1? Why not also whiten the raw input?

**A**: The only effect of whitening is to make the features isotropic, which does not change the expressivity of the features
(since it is only a linear transformation) but is beneficial to generalization, as discussed in Line 208-217. We also
showed that whitening is not the only option — using unwhitened features $\mathbf{g}$ along with a proper data dependent
regularizer on $\mathbf{W}$ gives us exactly the same result as whitening the features (Appendix C.5). On the other hand, existing
results on raw representations already assumed $\mathbf{x}$ is exactly or nearly isotropic (Line 244 for `NTK-Raw`; Line 196-198
for `Quad-Raw`). Those bounds won't be improved if we further whiten $\mathbf{x}$ to be exactly isotropic.

**Q**: What properties of your random representations actually make the difference for the sample complexity here?

**A**: The key thing that allowed a fixed neural representation to be helpful is that the nonlinearities (along with the width)
give us strong expressive power. Linear combinations of these fixed neurons can already express high-complexity
nonlinear functions, and such expressivity can be used by the top trainable model to reduce the complexity of the
function it has to learn itself. In comparison, when we use the raw input, linear combinations of the input is only a
linear function, thus all the "heavylifting" is on the top trainable model, causing the sample complexity to be higher.
Therefore, our theory shows that lower-layer representations can ease the burden of learning in the upper layers, which
we suspect is also the case in practice even when the representation function is trainable as well.

**Reviewer #4**:

**Q**: Are our models overparameterized? How would our results change with smaller $m$.

**A**: Our model is overparameterized (we chose $D, m$ to be large) for the purpose of approximating the ground truth
function and making the optimization landscape nice. However, our choice of $D, m$ does not explicitly depend on $n$
(Line 181), making our model not necessarily wide enough to memorize the training data. In this sense we are not as
overparametrized as the memorization regime.

**Q**: It would be nice for you to discuss how your work relates to other works that construct random features.

**A**: Prior work e.g. of Rahimi and Recht considered training only the output layer of the network ($a_r$ in our notation),
which is effectively a linear model/kernel method. In constrast our model is non-linear (quadratic) in the trainable
parameter $\mathbf{W}$ and has different optimization/generalization behaviors from kernel methods.

**Q**: Further motivations for considering the quadratic Taylor model.

**A**: As one example apart from the theoretical benefits shown in this paper, empirically the (full) quadratic Taylor model
also approximates the training trajectories of standard neural networks better than the linearized model, as shown in Bai
et al. 2020 "Taylorized Training".

We appreciate all the above questions and will incorporate these discussions in our next version.

[Meta-Review · NeurIPS 2020]

The reviewers consider the results clearly stated and novel.